

# Robust semi-Dirac points and unconventional topological phase transitions in doped superconducting Sr2IrO4 tunnel coupled to t2g electron systems

**Mats Horsdal[1]⋆ and Timo Hyart[2]**

**1** Department of Physics, University of Oslo, P. O. Box 1048 Blindern, N-0316 Oslo, Norway
**2** Department of Physics and Nanoscience Center, University of Jyväskylä, P.O. Box 35 (YFL), FI-40014 University of Jyväskylä, Finland

⋆ mats.horsdal@fys.uio.no

## Abstract

Semi-Dirac fermions are known to exist at the critical points of topological phase transitions requiring fine-tuning of the parameters. We show that robust semi-Dirac points can appear in a heterostructure consisting of superconducting Sr2IrO4 and a t2g electron system (t2g-ES) without fine-tuning. They are topologically stable in the presence of the symmetries of the model, metallic t2g-ES and a single active band in Sr2IrO4. If the t2g metal is coupled to two different layers of Sr2IrO4 (effectively a multiband superconductor) in a three-layer-structure the semi-Dirac points can split into two stable Dirac points with opposite chiralities. A similar transition can be achieved if the t2g-ES supports intrinsic triplet superconductivity. By considering Sr2RuO4 as an example of a t2g-ES we predict a rich topological phase diagram as a function of various parameters.

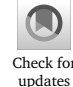

## 1 Introduction

Native triplet superconductivity is typically fragile and appears only at very low temperatures [1–4]. Therefore, driven by the desire to realize exotic topological phases and Majorana zero modes [4–8], a great deal of research has been invested in different ways of engineering materials so that their low-energy theory is described by effective triplet pairing correlations [2, 9–16]. Recently, the idea to utilize strong *intraionic spin-orbit coupling* in a middle layer to convert singlet Cooper pairs into triplet ones in a three layer heterostructure was proposed [17], and the growth technology for the realization of such kind of heterostructures is under development [18]. The advantage of this idea is that the conversion from singlet to triplet Cooper pairs can take place in a single atomic layer, so that the induced superconducting order parameter is determined by microscopic energy scales given by tunneling amplitudes between the layers and the superconducting gap in the singlet superconductor.

In this manuscript we explore the possibility and consequences of triplet pairing correlations in a heterostructure where doped superconducting $Sr_2IrO_4$ with strong intraionic spin orbit coupling is tunnel coupled to a $t_{2g}$ electron system ($t_{2g}$-ES). $Sr_2IrO_4$ is a layered $5d^5$ transition metal oxide (TMO) where the strong spin-orbit coupling mixes the $t_{2g}$ orbitals ($|yz\rangle$, $|zx\rangle$ and $|xy\rangle$) [19–21] so that there exists only one active band described by the hybridized $j_{\text{eff}} = 1/2$ states labelled by the *pseudospin*

$$|f, \Uparrow\rangle = \frac{1}{\sqrt{3}}[|xy, \uparrow\rangle + |yz, \downarrow\rangle + i|xz, \downarrow\rangle],$$

$$|f, \Downarrow\rangle = \frac{1}{\sqrt{3}}[|xy, \downarrow\rangle - |yz, \uparrow\rangle + i|xz, \uparrow\rangle]. \tag{1}$$

Due to strong correlation effects $Sr_2IrO_4$ is a Mott insulator at half-filling [22–24] and it is expected to become a high-temperature superconductor upon doping [21, 24, 25]. It may be considered as the best studied member of the family of the iridate compounds which are anticipated to support a zoo of topological spin liquid and superconducting phases due to cooperative action of spin-orbit coupling and Coulomb interactions. These topological phases include the Kitaev spin liquid phase [19, 26], different types of three dimensional spin liquid phases [27–30], the chiral $d$-wave superconductor phase [31, 32], $p$-wave superconductors with helical, chiral and flat Majorana edge modes [32–35] and three-dimensional nodal superconducting [36] phases. In contrast to these more complicated compounds, $Sr_2IrO_4$ has a square lattice and upon electron doping it is expected to support a $d$-wave superconducting

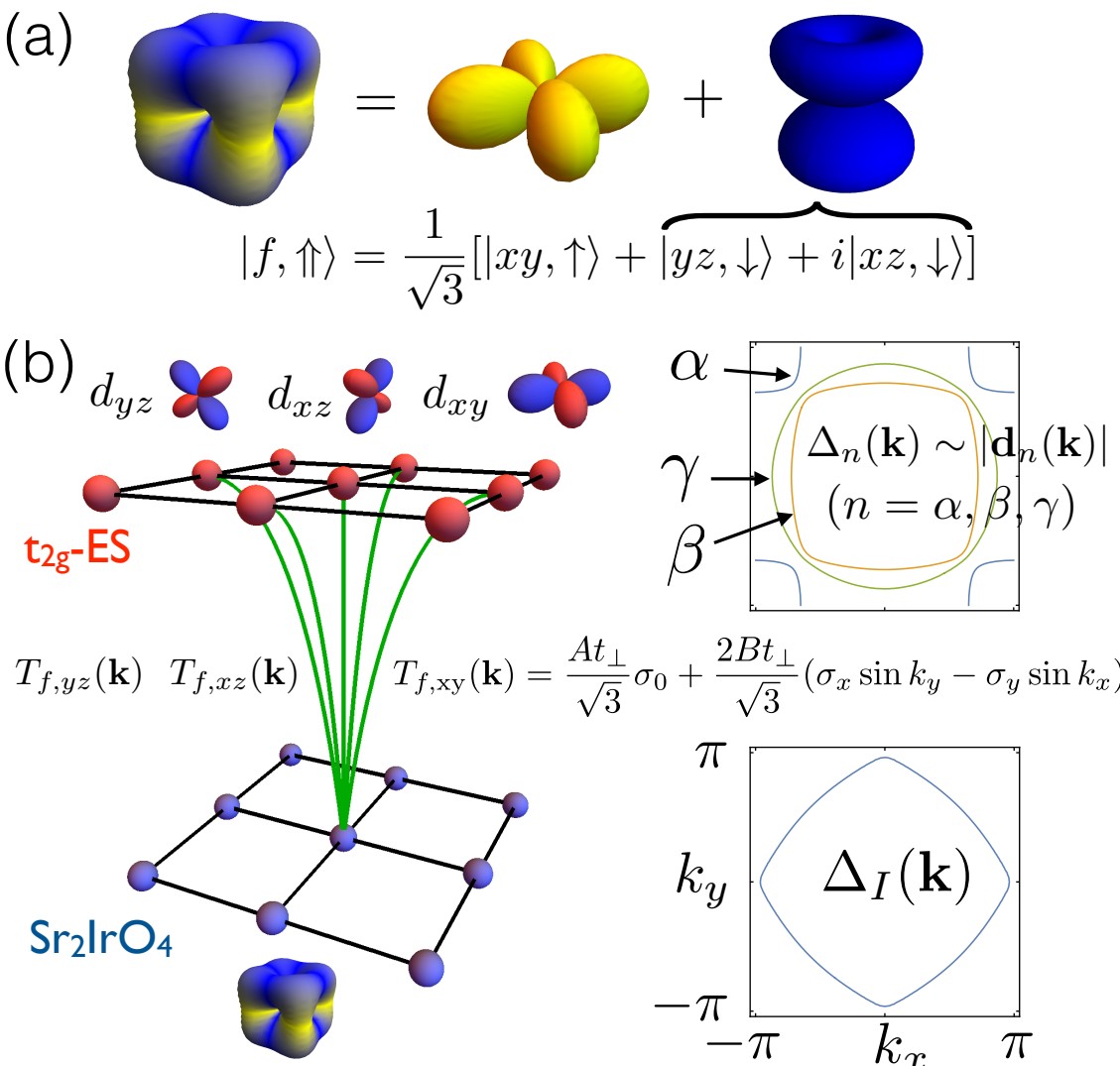

Figure 1: (a) Due to the strong spin-orbit coupling the active pseudospin orbitals in $Sr_2IrO_4$ are described by strongly entangled spin and orbital degrees of freedom. (b) When $Sr_2IrO_4$ is tunnel coupled to a $t_{2g}$ electron system, the tunneling of the pseudospin singlet Cooper pairs [described by the order parameter $\Delta_I(\mathbf{k})$] into the $t_{2g}$-ES leads to an apperance of singlet $\Delta_n(\mathbf{k})$ and triplet $\mathbf{d}_n(\mathbf{k})$ components of the induced order parameter in each band $n = \alpha, \beta, \gamma$ of the $t_{2g}$-ES. These order parameters can have similar magnitudes $\Delta_n(\mathbf{k}) \sim |\mathbf{d}_n(\mathbf{k})|$ leading to a possibility of a robust semi-Dirac phase.

phase analogously to the cuprates [21, 24, 25]. Moreover, the first experimental signatures of the $d$-wave superconductivity have already been observed in doped $Sr_2IrO_4$ [37,38]. However, in contrast to cuprates, the strong spin-orbit coupling in $Sr_2IrO_4$ causes the Cooper pairs to be formed as pseudospin singlets, where the pseudospin [Eq. (1)] describes entangled orbital and spin degrees of freedom (see Fig. 1). This has no consequences in the earlier studies where the isolated $Sr_2IrO_4$ was studied. However, we show that the differences to cuprate superconductors become evident when $Sr_2IrO_4$ is tunnel coupled to another system in a heterostructure.

In this paper we consider a heterostructure consisting of a doped superconducting $Sr_2IrO_4$ tunnel coupled to a reasonably thin layer of a $t_{2g}$-ES. Suitable candidates for the $t_{2g}$-ES are the extensively studied 4d TMOs $Sr_2RuO_4$ and $Sr_2RhO_4$ because they have similar crystal structures as $Sr_2IrO_4$ [3, 39–42]. $Sr_2RhO_4$ is observed to be metallic down to the lowest experi-

mentally accessible temperatures and $Sr_2RuO_4$ supports an interesting superconducting phase at low temperatures, where the order parameter is of multi-orbital nature and not yet fully understood [43–46]. We show that the tunneling of the pseudospin singlet Cooper pairs from $Sr_2IrO_4$ into the $t_{2g}$-ES naturally leads to an apperance of both triplet and singlet Cooper pairs in the $t_{2g}$-ES, so that the triplet and singlet components of the induced order parameter are of similar magnitude (see Fig. 1).

We find that in the case of a metallic $t_{2g}$-ES the Bogoliubov-de Gennes (BdG) Hamiltonian generically supports robust semi-Dirac points. Moreover, we generalize this result for a class of superconductor-metal heterostructures. The semi-Dirac points can be described with an effective low-energy Hamiltonian

$$H_{SD} = \hbar v q_x \sigma_x + \mathscr{C}_1 (\mathscr{C} q_x + \mathscr{D} q_y)^2 \sigma_y, \tag{2}$$

where $q_x$ and $q_y$ are the deviations of the momentum from the two-fold degenerate nodal point in the quasiparticle spectrum perpendicular and parallel to the Fermi line of the $t_{2g}$-ES metal, respectively. The descriptive picture of the semi-Dirac Hamiltonian (2) is that the quasiparticles are (massless) relativistic particles along the $q_x$-direction with velocity $v$ but they are nonrelativistic in the $q_y$-direction with an effective mass $\frac{\hbar^2}{2m} = \mathscr{C}_1 \mathscr{D}^2$ [47–54]. For generality we have allowed two dimensionless constants $\mathscr{C}$ and $\mathscr{D}$ in addition to $v$ and $\mathscr{C}_1$, but these parameters are not important for the qualitative low-energy properties as long as $v \neq 0$ and $\mathscr{C}_1 \mathscr{D}^2 \neq 0$.

Semi-Dirac nodal points are previously known to exist in different systems as *critical points* of a topological phase transition where as a function of some parameter $\mathscr{M}$ two Dirac points with opposite chiralities will meet and merge in the momentum space [47–50, 54]. In the presence of chiral symmetry this transition can be described with an effective Hamiltonian of the form

$$H = h_x(\mathbf{q})\sigma_x + h_y(\mathbf{q})\sigma_y, \tag{3}$$
$$h_x(\mathbf{q}) = \hbar v q_x, \ h_y(\mathbf{q}) = \mathscr{C}_1(\mathscr{C} q_x + \mathscr{D} q_y)^2 - (\mathscr{M} - \mathscr{M}_c),$$

where $\vec{h}(\mathbf{q})$ describes an effective momentum-dependent pseudomagnetic field in the vicinity of the merging point and $\mathscr{M}$ is a parameter which drives the quantum phase transition at $\mathscr{M} = \mathscr{M}_c$. For simplicity we assume $\frac{\hbar^2}{2m} = \mathscr{C}_1 \mathscr{D}^2 > 0$, $\mathscr{M}_c > 0$ and $\mathscr{C} = 0$, but these assumptions are not important as long as $v, \mathscr{C}_1, \mathscr{D} \neq 0$. The spectrum of this Hamiltonian is then given by

$$E_\pm(\mathbf{q}) = \pm E(\mathbf{q}) = \pm \sqrt{\hbar^2 v^2 q_x^2 + \left[\frac{\hbar^2 q_y^2}{2m} - (\mathscr{M} - \mathscr{M}_c)\right]^2}. \tag{4}$$

For $\mathscr{M} > \mathscr{M}_c$ this Hamiltonian describes two Dirac points located at $q_x = 0$ and $q_y = \pm\sqrt{2m(\mathscr{M} - \mathscr{M}_c)}/\hbar$ (see Fig. 2). These two Dirac points are described by low-energy Hamiltonians

$$H_D(q_x, \delta q_y) = \hbar v q_x \sigma_x \pm \hbar v_y \delta q_y \sigma_y, \tag{5}$$

where the velocity in $y$-direction is $v_y = \sqrt{2(\mathscr{M} - \mathscr{M}_c)/m}$, and $\delta q_y = q_y \mp \sqrt{2m(\mathscr{M} - \mathscr{M}_c)}/\hbar$ describes the deviation of the momentum from the Dirac point. The pseudomagnetic field $\vec{h}(\mathbf{q})$ forms vortices around the Dirac points [Fig. 2(a), (b)], and based on the direction of the winding of $\vec{h}(\mathbf{q})$ around them it is possible to define topological charges $Q^D = \pm 1$ for the Dirac points. When $\mathscr{M}$ approaches $\mathscr{M}_c$ from above the two Dirac points with opposite topological charges approach each other in the momentum space and they meet at $\mathscr{M} = \mathscr{M}_c$ forming a semi-Dirac point described by Hamiltonian (2). For $\mathscr{M} < \mathscr{M}_c$ the vortices are annihilated and the spectrum $E(\mathbf{q})$ is fully gapped. Although this type of merging transitions have been experimentally observed in different systems [55–59], it is difficult to study the phenomenology

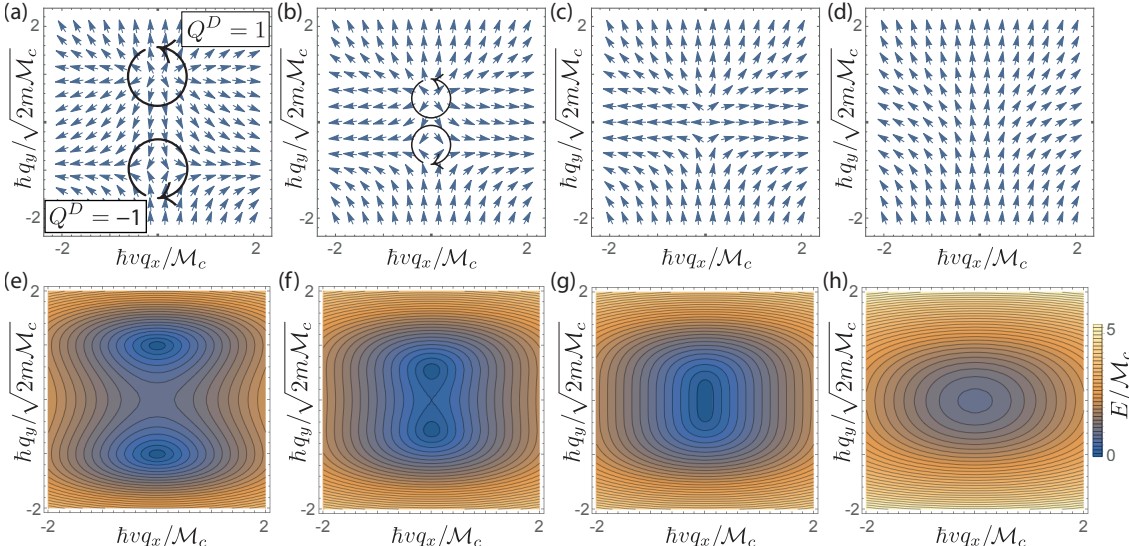

Figure 2: Illustration of the appearance of a semi-Dirac point as a critical point of a topological phase transition described by Hamiltonian (3). Figures (a)-(d) show the direction of the pseudomagnetic field $\vec{h}(\mathbf{q})/|\vec{h}(\mathbf{q})|$ [Eq. (3)] and figures (e)-(h) the energy spectrum $E(\mathbf{q})$ [Eq. (4)] as a function of momentum $\mathbf{q}$ for different values of $\mathcal{M}$: (a),(e) $\mathcal{M} = 2\mathcal{M}_c$, (b),(f) $\mathcal{M} = 1.3\mathcal{M}_c$, (c),(g) $\mathcal{M} = \mathcal{M}_c$ and (d),(h) $\mathcal{M} = 0$. For $\mathcal{M} > \mathcal{M}_c$ this Hamiltonian describes two Dirac points located at $q_x = 0$ and $q_y = \pm\sqrt{2m(\mathcal{M} - \mathcal{M}_c)}/\hbar$ [(e), (f)]. The pseudomagnetic field $\vec{h}(\mathbf{q})$ forms vortices around these points with opposite chiralities (i.e. a vortex-antivortex pair) [(a), (b)]. Therefore, it is possible to define topological charges $Q^D = \pm 1$ for the Dirac points based on the direction of the winding of $\vec{h}(\mathbf{q})$ around them [(a), (b)]. When $\mathcal{M}$ approaches $\mathcal{M}_c$ from above the two Dirac points with opposite topological charges approach each other in the momentum space and they meet at $\mathcal{M} = \mathcal{M}_c$ forming a semi-Dirac point described by Hamiltonian (2) [(c), (g)]. For $\mathcal{M} < \mathcal{M}_c$ the vortex-antivortex pair is annihilated and the spectrum $E(\mathbf{q})$ of the system is fully gapped [(d), (h)]. We have chosen $\frac{\hbar^2}{2m} = \mathcal{C}_1\mathcal{D}^2 > 0$, $\mathcal{M}_c > 0$ and $\mathcal{C} = 0$.

of the semi-Dirac points in these systems, because the semi-Dirac point appears only at the critical point at $\mathcal{M} = \mathcal{M}_c$.

In the presence of additional symmetries and constraints the semi-Dirac points may however become stable against small perturbations of the parameters of the model. Such kind of situation has been predicted to occur in a specific model [52], where there exists two overlapping bands which are not coupled directly but only virtually via a third band, and there exists a specific symmetry (mirror symmetry) which forbids this coupling to the third band within a particular high-symmetry line (mirror line). In this kind of situation the semi-Dirac points are stable and they always appear at the high-symmetry line. The robust semi-Dirac points of the Bogoliubov quasiparticles discussed in this manuscript have a very different origin. They do not require the existence of a high-symmetry line, which means that they can appear anywhere in the momentum space. Moreover, their robustness is of topological nature so that they carry topological charges $Q^{SD}$ (definition will be given below). This is a surprising result because the semi-Dirac points are not associated with Berry phases and therefore one might expect them to be unstable towards gapping or splitting into Dirac points. We also show that semi-Dirac points with opposite charges are always nucleated/annihilated in a pairwise manner as a function of the parameters of the model. The merging transitions of the semi-Dirac points

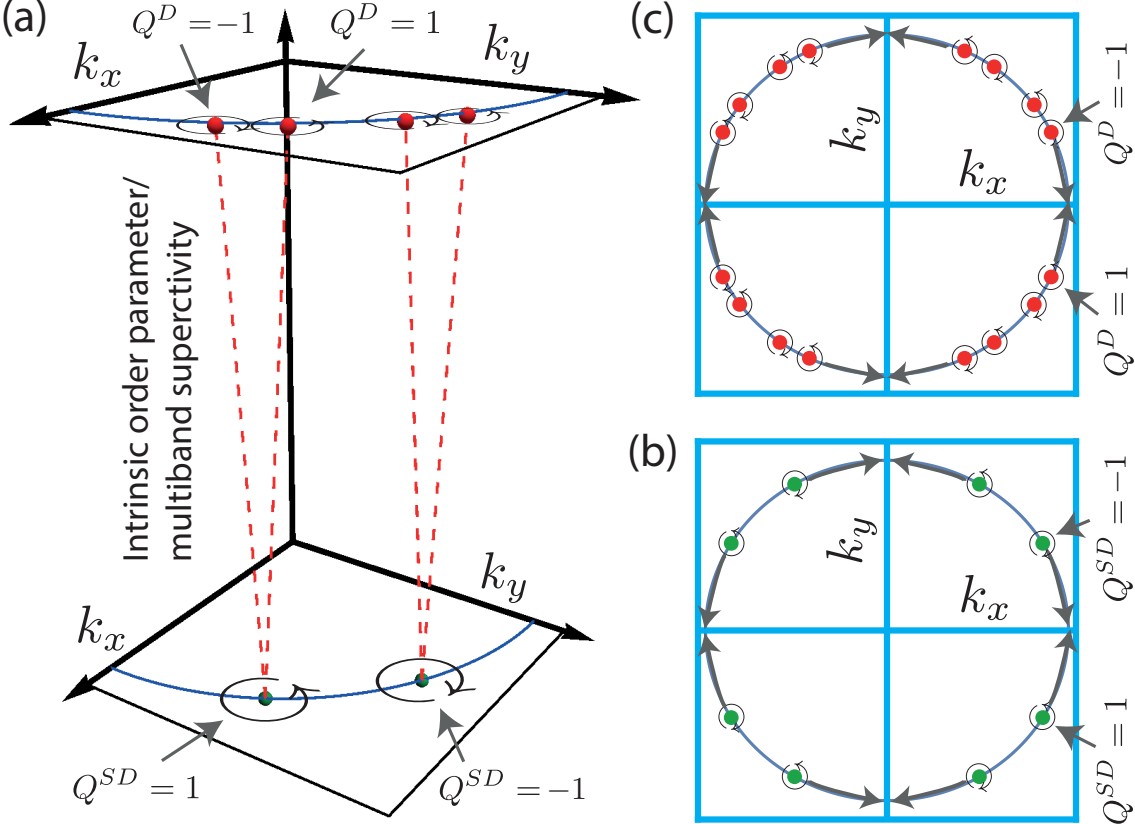

Figure 3: Different types of topological phase transitions occurring in heterostructures where $Sr_2IrO_4$ is tunnel coupled to a $t_{2g}$-ES. (a) Lower plane: a pair of semi-Dirac points carrying topological charges $Q^{SD} = \pm 1$. They are protected against small perturbations in the presence of symmetries of the model, metallic $t_{2g}$-ES and a single active band in $Sr_2IrO_4$. Upper plane: Each semi-Dirac point can become gapped or split into two Dirac points with opposite topological charges $Q^D = \pm 1$ if these conditions are intentionally broken so that the $t_{2g}$ metal is coupled to two different layers of $Sr_2IrO_4$ (effectively a multiband superconductor) or the $t_{2g}$-ES supports an intrinsic superconducting order parameter. These transitions are described by Hamiltonian (3). (b) In the constrained parameter space, where the semi-Dirac points are stable against small perturbations, two semi-Dirac points with opposite topological charges $Q^{SD}$ can be nucleated or annihilated in a pairwise manner. These merging transitions are desribed by Hamiltonian (6). (c) In the unconstrained parameter space, where a semi-Dirac point can split into two Dirac points, the Dirac points can move in the momentum space and merge with Dirac points (carrying opposite $Q^D$) that have emerged from other semi-Dirac points. When this merging occurs at the mirror lines (thick blue lines) these transitions lead to topological mirror superconductivity.

can be described with a low-energy Hamiltonian

$$H = \hbar v q_x \sigma_x + \mathscr{C}_1 \left[ \mathscr{C} q_x + \tilde{\mathscr{D}} q_y^2 - (\mathscr{M}^{SD} - \mathscr{M}_c^{SD}) \right]^2 \sigma_y. \tag{6}$$

Interestingly, at $\mathscr{M}^{SD} = \mathscr{M}_c^{SD}$ the dispersion is linear along the $q_x$ direction and quartic along the $q_y$ direction. The merging point of two semi-Dirac points may be considered as a simultaneous merging of four Dirac points. Such kind of transitions can usually only exist if several different parameters are simultaneously fine-tuned to particular values [60]. In the heterostructures studied in this manuscript they appear as critical points between the semi-Dirac phase and trivial phase (or two topologically distinct semi-Dirac phases as discussed

| | Model | Constr. parameter space | Topologically distinct phases |
|---|---|---|---|
| Minimal Model | Bi-layer: SC / metal | Yes | Stable semi-Dirac points |
| | Tri-layer: SC / metal / SC | No | Semi-Dirac points can be gapped or splitted into Dirac-points. Topological mirror superconductivity. |
| Microscopic Model | Bi-layer: $Sr_2IrO_4$ / metallic $t_{2g}$-ES | Yes | Stable semi-Dirac points |
| | Tri-layer: $Sr_2IrO_4$ / metallic $t_{2g}$-ES / $Sr_2IrO_4$ | No | Semi-Dirac points can be gapped or splitted into Dirac-points. Topological mirror superconductivity. |
| | Bi-layer: $Sr_2IrO_4$ / superconducting $t_{2g}$-ES | No | Semi-Dirac points can be gapped or splitted into Dirac-points. Topological mirror superconductivity. |

Table 1: Summary of the models discussed in the paper. For each model it is specified whether it is in the constrained parameter space, and we have also summarized the topologically distinct phases it can support. The minimal models are discussed in Section 2 and the microscopic models in Sections 3-8.

below) and therefore they can be realized as a function of any single parameter which can be used to drive a quantum phase transition between these topologically distinct phases.

We find that the semi-Dirac points in the superconductor-metal heterostructures are topologically stable in the presence of the symmetries of the model (time-reversal symmetry, two-fold rotational symmetry and inversion symmetries within the layers), metallic $t_{2g}$-ES and a single active band in the superconductor. This finding opens a path for breaking the protection of the semi-Dirac points intentionally in a controlled manner. Namely, each semi-Dirac point can become gapped or split into two Dirac points with opposite topological charges $Q^D = \pm 1$ if the $t_{2g}$ metal is coupled to two different layers of $Sr_2IrO_4$ (effectively a multiband superconductor) or the $t_{2g}$-ES supports intrinsic triplet superconductivity. These transitions are described by Hamiltonian (3). Moreover, arbitrary weak perturbations breaking this protection will lead to these transitions demonstrating the nature of the semi-Dirac points as critical points of topological phase transitions in the *unconstrained parameter space*. In addition to the splitting-merging transition of Dirac points described by Hamiltonian (3) and the merging of semi-Dirac points (6), we find that systems supporting an additional mirror symmetry can support a third type of topological phase transition. Namely, in the unconstrained parameter space, where a semi-Dirac point can split into two Dirac points, the Dirac points can move in the momentum space and merge with Dirac points (carrying opposite $Q^D$) that have emerged from other semi-Dirac points. When this merging occurs at the mirror lines [as illustrated in Fig. 3] these transitions lead to topological mirror superconductivity [61–63]. All these different types of topological phase transitions are summarized in Fig. 3. By considering $Sr_2RuO_4$ as a specific example of $t_{2g}$-ES we predict a rich topological phase diagram as a function of various parameters. Moreover, we discuss the properties of the surface states and the other experimental signatures of the different phases and phase transitions. In Table 1 we have summarized the conclusions for the different systems studied in this work. In the next section we study two simplified models: a bi-layer superconductor/metal heterostructure and a tri-layer superconductor/metal/superconductor heterostructure. The simplified models allow to analytically demonstrate the main result discussed above. From Section 3 onwards we show that similar results are obtained by studying more realistic microscopic models: a bi-layer $Sr_2IrO_4$/metallic $t_{2g}$-ES heterostructure, a tri-layer $Sr_2IrO_4$/metallic $t_{2g}$-ES/$Sr_2IrO_4$ heterostructure and a bi-layer $Sr_2IrO_4$/superconducting $t_{2g}$-ES heterostructure.

## 2 Minimal model for robust semi-Dirac points and unconventional phase transitions

The full model for $Sr_2IrO_4$ tunnel coupled to $t_{2g}$-ES is a reasonably complex system so that we need to partially rely on numerical calculations. Therefore, it is useful to first illustrate the basic ideas using the simplest possible model that already gives rise to the robust semi-Dirac points and unconventional topological phase transitions illustrated in Fig. 3. We stress that the model used in this section is constructed mainly for illustration purposes, and we are not aware of suitable materials and conditions under which this model would be realized. However, given the simplicity of this model it seems plausible that in the future it will be possible to identify physical systems where this model is faithfully realized.

Namely, in this section we first consider a single-band $s$-wave superconductor tunnel coupled to a single-band metal. The Hamiltonian for the $s$-wave superconductor can be written in the Nambu basis $\Psi_{\mathbf{k}}^{\dagger} = (c_{\mathbf{k}\Uparrow}^{\dagger}, c_{\mathbf{k}\Downarrow}^{\dagger}, c_{-\mathbf{k}\Uparrow}, c_{-\mathbf{k}\Downarrow})$ as

$$H_{SC}(\mathbf{k}) = \begin{pmatrix} \xi_{SC}(\mathbf{k})\sigma_0 & i\Delta_{SC}\sigma_y \\ -i\Delta_{SC}\sigma_y & -\xi_{SC}(\mathbf{k})\sigma_0 \end{pmatrix}, \tag{7}$$

where $\sigma_i$ are the Pauli spin matrices. We have assumed that the superconductor obeys time-reversal and inversion symmetries so that there exists degenerate Kramer's partners $|\mathbf{k}\Uparrow\rangle$ and $|\mathbf{k}\Downarrow\rangle$ at each momentum, and the singlet pairing takes place within this internal degree of freedom. It turns out that the dispersion of the superconductor $\xi_{SC}(\mathbf{k})$ is unimportant (see sections below) so in this section we neglect it by assuming $\xi_{SC}(\mathbf{k}) = 0$. We fix the gauge so that the superconducting order parameter satisfies $\Delta_{SC} > 0$. In this section we also neglect the possible momentum dependence of the singlet order parameter $\Delta_{SC}$, because it is not important for the appearance of the semi-Dirac points and unconventional topological phase transitions (see sections below). We assume that the metal obeys time-reversal and inversion symmetries so that there also exists degenerate Kramer's partners at each momentum in the metal. In general these internal degrees of freedom in the metal and superconductor are different from each other, and this is important in the following. The Hamiltonian of the metal in the basis of the eigenfunctions of the metal (in Nambu space) can then be written as

$$H_M(\mathbf{k}) = \begin{pmatrix} \xi_M(\mathbf{k})\sigma_0 & 0 \\ 0 & -\xi_M(\mathbf{k})\sigma_0 \end{pmatrix}. \tag{8}$$

For simplicity, we assume that the metal has a spherical Fermi surface with dispersion

$$\xi_M(\mathbf{k}) = \hbar v(|\mathbf{k}| - k_F). \tag{9}$$

The metal and the superconductor are tunnel coupled via a tunneling matrix $T(\mathbf{k})$, so that the Hamiltonian for the full system (in the Nambu space) is

$$H = \begin{pmatrix} H_M(\mathbf{k}) & H_T(\mathbf{k}) \\ H_T^{\dagger}(\mathbf{k}) & H_{SC}(\mathbf{k}) \end{pmatrix}, H_T(\mathbf{k}) = \begin{pmatrix} T(\mathbf{k}) & 0 \\ 0 & -T^*(-\mathbf{k}) \end{pmatrix}. \tag{10}$$

We assume that the tunneling matrix obeys time-reversal

$$\sigma_y T^*(-\mathbf{k})\sigma_y = T(\mathbf{k}), \tag{11}$$

and two-fold rotational symmetries

$$\sigma_z T(-\mathbf{k})\sigma_z = T(\mathbf{k}). \tag{12}$$

In the following the essential requirement for the tunneling matrix is that it contains both diagonal and off-diagonal elements (momentum even and momentum odd components) of similar magnitude so that it mixes the internal degrees of freedom in the metal and superconductor. Such kind of tunneling matrices have been rarely considered in the literature because it is not obvious how they can be realized in physical systems. However, this kind of tunneling matrix is naturally realized when one of the layers (metal or superconductor) has a very large intraionic spin-orbit coupling so that the internal degree of freedom in that layer is the pseudospin discussed in Section 1. In the following sections we consider situations where the very strong intraionic spin-orbit coupling appears in the superconducting layer, but we point out that similar tunneling matrices are obtained also if the metallic layer has a strong intraionic spin-orbit coupling instead of the superconductor. To be explicit we assume

$$T(\mathbf{k}) = \frac{At_\perp}{\sqrt{3}}\sigma_0 + \frac{2Bt_\perp}{\sqrt{3}}(\sigma_x \sin k_y - \sigma_y \sin k_x), \tag{13}$$

where $t_\perp$ describes the overall magnitude of the tunneling, and $A$ and $B$ are dimensionless constants describing the relative magnitudes of momentum even and momentum odd components of the tunneling matrix. We discuss one possible realization of the tunneling matrix (13) below, but in this section this can be considered just as a simple model giving rise to semi-Dirac points.

The superconductor is fully gapped, so we can concentrate on the metallic layer. The low-energy BdG Hamiltonian for the metal can be expressed as

$$H(\mathbf{k}) = \begin{pmatrix} \xi_M(\mathbf{k})\sigma_0 & \Delta_{\text{ind}}(\mathbf{k}) \\ \Delta_{\text{ind}}^\dagger(\mathbf{k}) & -\xi_M(\mathbf{k})\sigma_0 \end{pmatrix}, \tag{14}$$

where $\Delta_{\text{ind}}(\mathbf{k})$ is the induced order parameter i.e. the anomalous part of the self-energy evaluated at the Fermi energy. It is given by

$$\Delta_{\text{ind}}(\mathbf{k}) \equiv i[\Delta(\mathbf{k})\sigma_0 + \mathbf{d}(\mathbf{k}) \cdot \vec{\sigma}]\sigma_y = i\left[\frac{T(\mathbf{k})T^\dagger(\mathbf{k})}{\Delta_{SC}}\right]\sigma_y, \tag{15}$$

where $\Delta(\mathbf{k})$ and $\mathbf{d}(\mathbf{k})$ are the induced singlet and triplet superconducting order parameters. To arrive to this expression we have utilized the time-reversal symmetry of the tunneling matrix [Eq. (11)]. The induced order parameter $\Delta_{\text{ind}}(\mathbf{k})$ satisfies $\Delta_{\text{ind}}^T(-\mathbf{k}) = -\Delta_{\text{ind}}(\mathbf{k})$, which means that $\Delta(-\mathbf{k}) = \Delta(\mathbf{k})$ and $\mathbf{d}(-\mathbf{k}) = -\mathbf{d}(\mathbf{k})$. Moreover, $d_z(\mathbf{k}) = 0$ due to symmetries. For the explicit form of the tunneling matrix (13) we obtain

$$d_x(\mathbf{k}) = \frac{4t_\perp^2}{3\Delta_{SC}}AB \sin k_y, \, d_y(\mathbf{k}) = -\frac{4t_\perp^2}{3\Delta_{SC}}AB \sin k_x,$$
$$\Delta(\mathbf{k}) = \frac{t_\perp^2}{3\Delta_{SC}}\left[A^2 + 4B^2(\sin^2 k_y + \sin^2 k_x)\right]. \tag{16}$$

By diagonalizing the Hamiltonian (14) we find that the quasiparticle energies are $E(\mathbf{k}) = \pm E_\pm(\mathbf{k})$, where

$$E_\pm(\mathbf{k}) = \sqrt{\xi_M^2(\mathbf{k}) + \left[\Delta(\mathbf{k}) \pm |\mathbf{d}(\mathbf{k})|\right]^2}. \tag{17}$$

The BdG Hamiltonian (14) contains a particle-hole redundancy, which gives rise to a Majorana constraint $\Gamma_E^\dagger(\mathbf{k}) = \Gamma_{-E}(-\mathbf{k})$ for the creation and annihilation operators obtained as solutions of the BdG equation. Therefore, the positive and negative energy solutions do not describe independent degrees of freedom and the quasiparticles can be considered as their own antiparticles i.e. they are Majorana fermions [64–67]. This is a generic property of all Bogoliubov quasiparticles, which means that all the quasiparticles considered in this paper have this Majorana character.

It is easy to see from Eq. (17) that there are nodes in the quasiparticle spectrum at the momenta where $\xi_M(\mathbf{k}) = 0$ and $\Delta(\mathbf{k}) = |\mathbf{d}(\mathbf{k})|$. Importantly, it follows from the symmetries (11) and (12) that the matrix elements of $T(\mathbf{k})$ satisfy $T_{11}(\mathbf{k}) = T_{22}^*(\mathbf{k})$ and $T_{12}(\mathbf{k}) = T_{21}^*(\mathbf{k})$ so that $\det[T(\mathbf{k})] \in \mathbb{R}$. Moreover, it is possible to show that

$$\Delta(\mathbf{k}) - |\mathbf{d}(\mathbf{k})| = \frac{1}{\Delta_{SC}} \frac{\det^2[T(\mathbf{k})]}{(|T_{11}(\mathbf{k})| + |T_{12}(\mathbf{k})|)^2}. \tag{18}$$

Thus the induced singlet order parameter is always larger or equal to the induced triplet order parameter $\Delta(\mathbf{k}) \geq |\mathbf{d}(\mathbf{k})|$ and they are equal if and only if $\det[T(\mathbf{k})] = 0$. The conditions for the appearance of the nodes in the quasiparticle spectrum can therefore be expressed as

$$E(\mathbf{k}) = 0 \iff \det[T(\mathbf{k})] = 0 \text{ and } \xi_M(\mathbf{k}) = 0. \tag{19}$$

The appearance of the nodes when these conditions are satisfied can be verified also using the Hamiltonian (10).

At first sight it seems that the conditions given in Eq. (19) are difficult to satisfy simultaneously, since in general $\det[T(\mathbf{k})] \in \mathbb{C}$, and therefore three equations would need to be satisfied by varying two variables $k_x$ and $k_y$. However, as discussed above, in the presence of the symmetries of the model $\det[T(\mathbf{k})] \in \mathbb{R}$, and therefore these conditions can be satisfied in a robust manner if $\det[T(\mathbf{k})]$ changes sign along the Fermi line where $\xi_M(\mathbf{k}) = 0$. We will now illustrate the appearance of these robust nodal points in the case of the specific tunneling matrix (13). In this case

$$\Delta(\mathbf{k}) - |\mathbf{d}(\mathbf{k})| = \frac{t_\perp^2}{3\Delta_{SC}} \left[ |A| - 2|B| \sqrt{\sin^2 k_y + \sin^2 k_x} \right]^2,$$

$$\det[T(\mathbf{k})] = \frac{t_\perp^2}{3} \left[ A^2 - 4B^2(\sin^2 k_y + \sin^2 k_x) \right]. \tag{20}$$

Therefore, it is possible to realize the situation illustrated in Fig. 4 where the regions of $\xi_M(\mathbf{k}) < 0$ and $\det[T(\mathbf{k})] < 0$ partially overlap, and due to continuity of these functions there must exist values of $k_x$ and $k_y$, where $\xi_M(\mathbf{k}) = 0$ and $\det[T(\mathbf{k})] = 0$ are simultaneously satisfied. Furthermore, these nodal points are robust against small variations of parameters because the only way to remove them is to deform the regions of $\xi_n(\mathbf{k}) < 0$ and $\det[T(\mathbf{k})] < 0$ in such a way that their boundaries no longer cross each other in the momentum space. Thus the nodal points exist in a full phase in the parameter space and they are always nucleated/annihilated in a pairwise manner (see Fig. 4). Using Eqs. (9) and (20) and assuming $k_F < \pi$, we find that the condition for the existence of nodes is

$$\exists \mathbf{k} \text{ s.t.} E(\mathbf{k}) = 0 \iff 2\sin k_F \leq \left| \frac{A}{B} \right| \leq 2\sqrt{2} \sin\left( \frac{k_F}{\sqrt{2}} \right) \tag{21}$$

and the system is fully gapped otherwise.

We can formalize the topological protection of these nodal points by defining a topological charge $Q_m^{SD}$ for the $m$th node at $\mathbf{k} = \mathbf{k}_m$ as

$$Q_m^{SD} = -\frac{i}{2\pi} \oint_{\mathbf{k}_m} d\mathbf{k} \cdot \frac{1}{Z^{SD}(\mathbf{k})} \nabla_\mathbf{k} Z^{SD}(\mathbf{k}),$$

$$Z^{SD}(\mathbf{k}) = \xi_M(\mathbf{k})/(\hbar v k_F) + i \det\left[ T(\mathbf{k})/t_\perp \right], \tag{22}$$

where the integral is calculated around a path enclosing the nodal point at $\mathbf{k} = \mathbf{k}_m$. $Q_m^{SD}$ are always integers, and the nodal points can be considered as vortices in the $\mathbf{k}$-space formed in

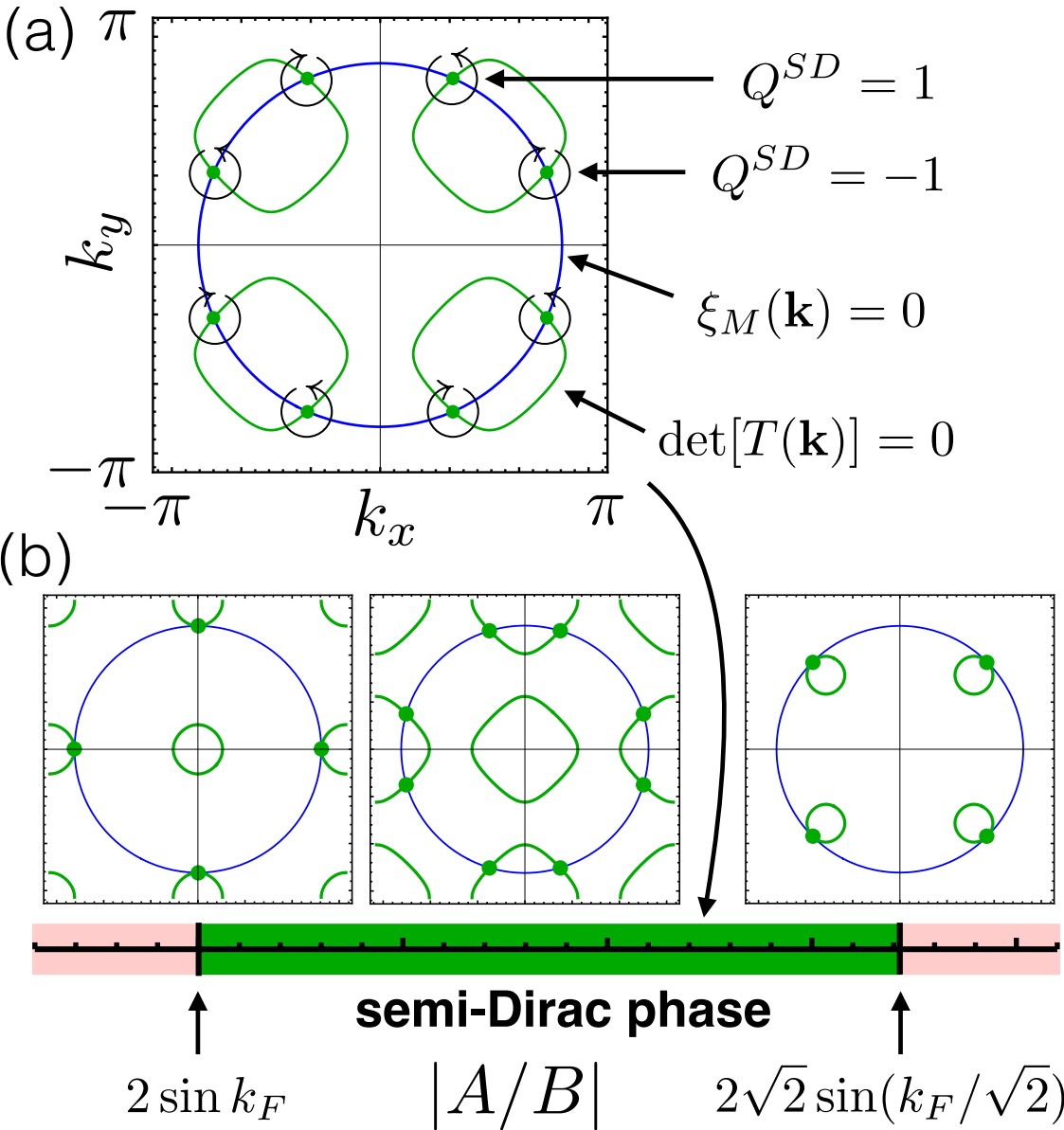

Figure 4: Illustration of the apperance of robust semi-Dirac points. (a) In the semi-Dirac phase the regions of $\xi_M(\mathbf{k}) < 0$ and $\det[T(\mathbf{k})] < 0$ partially overlap, so that there exists robust nodal points at the intersection points of the lines $\xi_M(\mathbf{k}) = 0$ and $\det[T(\mathbf{k})] = 0$. These nodal points are therefore stable against small perturbations of the parameters of the model and they carry topological charges $Q^{SD} = \pm 1$ [Eq. (22)]. The low-energy theory in the vicinity of these nodal points is described by the semi-Dirac Hamiltonian [Eq. (2)] i.e. the dispersion is linear in the direction perpendicular to the Fermi line, and quadratic in the direction along the Fermi line. (b) Phase diagram for the model described by the Hamiltonian (10) with tunneling matrix (13). The semi-Dirac phase appears for $2\sin k_F < |A/B| < 2\sqrt{2}\sin(k_F/\sqrt{2})$. Outside this region of parameters the system is in a trivial fully gapped phase. The semi-Dirac points with opposite topological charges are nucleated/annihilated in a pairwise manner, so that in the vicinity of the transition points between the semi-Dirac phase and the trivial phase the system is described by the Hamiltonian (6). At $|A/B| = 2\sin k_F$ the merging of the semi-Dirac points occurs at $(k_x, k_y) =: (\pm k_F, 0), (0, \pm k_F)$. At $|A/B| = 2\sqrt{2}\sin(k_F/\sqrt{2})$ the merging occurs at $(k_x, k_y) =: (\pm k_F, \pm k_F)/\sqrt{2}$. In the figures we have chosen $k_F = 5\pi/6$.

the field defined by $Z^{SD}(\mathbf{k})$. The pairs of the nodal points which are nucleated/annihilated together carry opposite topological charges.

By projecting the Hamiltonian (14) to the basis determined by the eigenvectors at the nodal point where $\xi_M(\mathbf{k}) = 0$ and $\det[T(\mathbf{k})] = 0$ we find that the generic low-energy theory for each band can be expressed as

$$H^{\mathrm{eff}}(\mathbf{k}) = \xi_M(\mathbf{k})\sigma_x + [\Delta(\mathbf{k}) - |\mathbf{d}(\mathbf{k})|]\sigma_y. \qquad (23)$$

Because $[\Delta(\mathbf{k}) - |\mathbf{d}(\mathbf{k})|] \propto \det^2[T(\mathbf{k})]$, we can usually expand the Hamiltonian (23) around the nodal point in a form described by the semi-Dirac Hamiltonian [Eq. (2)]. Here we have used the polar coordinates $q_x = |\mathbf{k}| - k_F$ and $q_y = k_F(\varphi - \varphi_m)$, where $\varphi$ is the polar angle and $\mathbf{k}_m = k_F(\cos\varphi_m, \sin\varphi_m)$. Therefore, the nodal points discussed above are semi-Dirac points and the corresponding phase in parameter space, where the semi-Dirac points are present, can be called semi-Dirac phase.

The phase diagram for the model described by the Hamiltonian (10) with tunneling matrix (13) is shown in Fig. 4(b). The semi-Dirac phase appears if the conditions given in Eq. (21) are satisfied. Outside this region of parameters the system is in a trivial fully gapped phase. The semi-Dirac points with opposite topological charges are nucleated/annihilated in a pairwise manner, so that in the vicinity of the transition points between the semi-Dirac phase and the trivial phase the system is described by the Hamiltonian (6). At $\left|A/B\right| = 2\sin k_F$ the merging of the semi-Dirac points occurs at $(k_x, k_y) =: (\pm k_F, 0), (0, \pm k_F)$, whereas at $\left|A/B\right| = 2\sqrt{2}\sin\left(k_F/\sqrt{2}\right)$ the merging occurs at $(k_x, k_y) =: (\pm k_F, \pm k_F)/\sqrt{2}$.

As discussed in Sec. 1, the semi-Dirac points, described by the low-energy Hamiltonian (2), are known to exist in different systems as critical points of a topological phase transition where two Dirac points meet and merge in the momentum space. In contrast to these previous studies, the semi-Dirac points in the kind of superconductor-metal heterostructures are stable against small perturbations so that they exist within a full phase in the parameter space. The reason for this stability is that the induced order parameters satisfy $[\Delta(\mathbf{k}) - |\mathbf{d}(\mathbf{k})|] \propto \det^2[T(\mathbf{k})]$ and this quantity determines one of the components of the pseudomagnetic field $h_y(\mathbf{k}) = \Delta(\mathbf{k}) - |\mathbf{d}(\mathbf{k})|$ in the low-energy theory of the system (23). On one hand, it follows from these relations that the induced singlet order parameter is always larger or equal to the induced triplet order parameter $|\Delta(\mathbf{k})| \geq |\mathbf{d}(\mathbf{k})|$. This immediately results in no-go theorems as it prevents the possibility of topologically nontrivial fully gapped triplet dominating superconducting phase in this kind of systems in agreement with previous findings [68,69]. Moreover, it also prevents the possibility of a gapless nodal superconducting phase with a Dirac Hamiltonian (5) around the nodal point because $h_y(\mathbf{k})$ does not change sign at the nodal point.[1] On the other hand, the fact that $h_y(\mathbf{k}) \propto \det^2[T(\mathbf{k})]$ means that it is possible to define a generalized square root of $h_y(\mathbf{k})$ in such a way that it is a polynomial function of the parameters of the Hamiltonian. This generalized square root is essentially $\det[T(\mathbf{k})]$ which then enters into the definition of the topological charge for the semi-Dirac point [Eq. (22)]. The generalized roots which are polynomial functions of the Hamiltonian parameters are in general a resource of topological invariants because they can change sign when the energy gap closes as a function of various parameters [70]. However, to our knowledge the topological charge for semi-Dirac points [Eq. (22)] has not been previously proposed in the literature. The significance of this result is that for the realization of the semi-Dirac phase one only needs to find superconductor-metal heterostructures supporting *topologically*

---

[1] This statement is true as long as we are considering time-reversal invariant single-band singlet superconductors where $\Delta_{SC}(\mathbf{k})$ does not change sign. In the case of more general singlet order parameters (e.g. $d$-wave superconductors) Dirac points can appear. However, they are still constrained to exist within particular lines in the momentum space where $\Delta_{SC}(\mathbf{k})$ changes sign.

*nontrivial tunneling matrices $T(\mathbf{k})$ such that* $\det[T(\mathbf{k})]$ *changes sign along the Fermi line of the metal* $\xi_M(\mathbf{k}) = 0$.

In the following sections we identify more carefully the conditions under which the semi-Dirac points are stable. We find that they are topologically stable in the presence of time-reversal symmetry, two-fold rotational symmetry, inversion symmetries within the layers and a single active band in the superconductor. If parameters of the model are changed within the constrained parameter space where these conditions are satisfied the semi-Dirac points are stable against small perturbations and they are always annihilated/nucleated in a pairwise manner. However, we can also break the protection of the semi-Dirac points intentionally in a controlled manner. Namely, each semi-Dirac point can become gapped or split into two Dirac points if the metallic layer is tunnel coupled to two separate superconducting layers in a three-layer-structure. In this case there are two distinct superconducting bands so that the system is effectively coupled to a multiband superconductor. Alternatively, this kind of splitting can occur if the metallic layer is replaced with a superconductor supporting an intrinsic superconducting order parameter. Moreover, arbitrary weak perturbations breaking this protection will lead to these transitions demonstrating the nature of the semi-Dirac points as critical points of topological phase transitions in the unconstrained parameter space.

We can demonstrate the different types of possible transitions in the unconstrained parameter space using a simple modification of our minimal model

$$H(\mathbf{k}) = \begin{pmatrix} \xi_M(\mathbf{k})\sigma_0 & \Delta_{\text{ind}}(\mathbf{k}) + \tilde{\Delta}(\mathbf{k}) \\ \Delta_{\text{ind}}^\dagger(\mathbf{k}) + \tilde{\Delta}^\dagger(\mathbf{k}) & -\xi_M(\mathbf{k})\sigma_0 \end{pmatrix}, \tag{24}$$

where

$$\tilde{\Delta}(\mathbf{k}) = i[\bar{\Delta}(\mathbf{k})\sigma_0 + \bar{\mathbf{d}}(\mathbf{k}) \cdot \vec{\sigma}]\sigma_y \tag{25}$$

is a perturbation in the order parameter, which can originate either from an intrinsic order parameter or it can be an induced order parameter from another superconducting band or another independent superconducting layer. In this section we choose a specific form for $\tilde{\Delta}(\mathbf{k})$

$$\bar{\Delta}(\mathbf{k}) = -(\tilde{t}_\perp/t_\perp)^2 \Delta(\mathbf{k}), \quad \bar{\mathbf{d}}(\mathbf{k}) = (\tilde{t}_\perp/t_\perp)^2 \mathbf{d}(\mathbf{k}). \tag{26}$$

In Sec. 6 we show that this can be realized in a three-layer heterostructure where the metallic layer is tunnel coupled to two different superconductors with relative phase difference of the order parameters $\varphi = \pi$. In this setup $\tilde{t}_\perp$ describes the overall magnitude of the tunneling to the second superconducting layer. However, in this section $|\tilde{t}_\perp/t_\perp|$ can simply be considered as a dimensionless parameter, and we can study the behavior of the system when it is varied. Without loss of generality we assume $|\tilde{t}_\perp/t_\perp| < 1$.

The Hamiltonian (24) satisfies particle-hole symmetry

$$\tau_x\sigma_0 H^T(-\mathbf{k})\tau_x\sigma_0 = -H(\mathbf{k}) \tag{27}$$

and time-reversal symmetry

$$\tau_0\sigma_y H^T(-\mathbf{k})\tau_0\sigma_y = H(\mathbf{k}), \tag{28}$$

where the Pauli matrices $\tau_i$ and $\sigma_i$ act in the particle-hole and the "spin" (Kramer's partners in the normal state Hamiltonian of the metal) spaces, respectively. Together these two symmetries give rise to a chiral symmetry

$$CH(\mathbf{k})C = -H(\mathbf{k}), \quad C = \tau_x\sigma_y. \tag{29}$$

Therefore the Hamiltonian (24) can be block-off-diagonalized into a form

$$V^\dagger H(\mathbf{k})V = \begin{pmatrix} 0 & D(\mathbf{k}) \\ D^\dagger(\mathbf{k}) & 0 \end{pmatrix}, \tag{30}$$

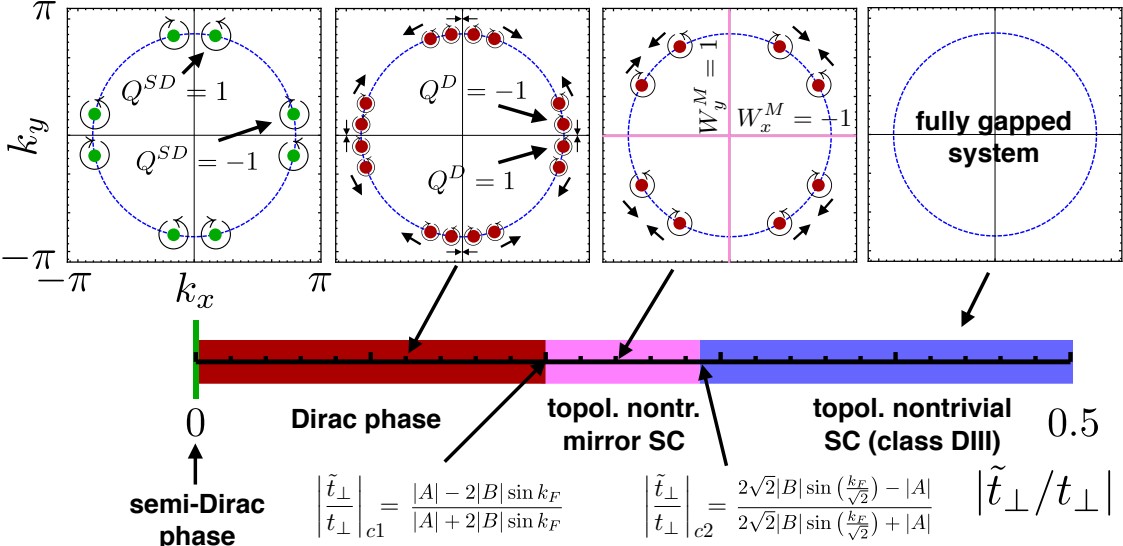

Figure 5: Phase diagram for the system described by the Hamiltonian (24) as a function of the parameter $|\tilde{t}_\perp/t_\perp|$ describing the breaking of the protection of the semi-Dirac points. The parameters $|A/B| = 1.5$ and $k_F = 5\pi/6$ are chosen so that for $|\tilde{t}_\perp/t_\perp| = 0$ the system is in the semi-Dirac phase [Eq. (21)]. For $|\tilde{t}_\perp/t_\perp| = 0$ (within the constrained parameter space) the system supports 8 semi-Dirac points carrying topological charges $Q^{SD} = \pm 1$ [Eq. (22)]. By increasing $|\tilde{t}_\perp/t_\perp|$ (entering the unconstrained parameter space) each semi-Dirac point splits into two Dirac points [Eq. (3)], so that for $0 < |\tilde{t}_\perp/t_\perp| < |\tilde{t}_\perp/t_\perp|_{c1}$ [Eq. (35)] the system is in a Dirac phase supporting 16 Dirac points. Each Dirac point carries a topological charge $Q^D = \pm 1$ [Eq. (34)]. At $|\tilde{t}_\perp/t_\perp| = |\tilde{t}_\perp/t_\perp|_{c1}$ there occurs merging transitions of Dirac points at $(k_x, k_y) =: (\pm k_F, 0), (0, \pm k_F)$ [Eq. (3)]. These Dirac points were nucleated from two different semi-Dirac points and due to this merging transition the topological mirror invariants $W_x^{M,0}$ and $W_y^{M,0}$ [Eq. (46)] change from zero to $-1$ and 1. Therefore, for $|\tilde{t}_\perp/t_\perp|_{c1} < |\tilde{t}_\perp/t_\perp| < |\tilde{t}_\perp/t_\perp|_{c2}$ [Eq. (47)] the system is in a topologically nontrivial mirror superconducting phase, and additionally there exists also 8 Dirac points. At $|\tilde{t}_\perp/t_\perp| = |\tilde{t}_\perp/t_\perp|_{c2}$ there occurs another merging of Dirac points at $(k_x, k_y) =: (\pm k_F, \pm k_F)/\sqrt{2}$ [Eq. (3)]. For $|\tilde{t}_\perp/t_\perp| > |\tilde{t}_\perp/t_\perp|_{c2}$ the system is in a fully gapped topologically nontrivial phase in class DIII, which is topologically equivalent to a helical $p$-wave superconductor.

where $D(\mathbf{k}) = \xi_M(\mathbf{k})\sigma_y + \Delta_{\text{ind}}(\mathbf{k}) + \tilde{\Delta}(\mathbf{k})$ and $V = (\tau_0\sigma_0 + i\tau_y\sigma_y)/\sqrt{2}$. We find that the conditions for the existence of nodal points in the quasiparticle spectrum are now

$$E(\mathbf{k}) = 0 \iff \det[\Delta_{\text{ind}}(\mathbf{k}) + \tilde{\Delta}(\mathbf{k})] = 0 \text{ and } \xi_M(\mathbf{k}) = 0. \tag{31}$$

Using the explicit form of the order parameters (16) corresponding to the tunneling matrix (13), the conditions for the existence of nodes can be written as

$$\xi_M(\mathbf{k}) = 0 \text{ and } \left|\frac{\tilde{t}_\perp}{t_\perp}\right| = \frac{\left||A| - 2|B|\sqrt{\sin^2 k_x + \sin^2 k_y}\right|}{|A| + 2|B|\sqrt{\sin^2 k_x + \sin^2 k_y}}. \tag{32}$$

In the following we choose $|A/B| = 1.5$ and $k_F = 5\pi/6$ so that for $|\tilde{t}_\perp/t_\perp| = 0$ the system is in the semi-Dirac phase [Eq. (21)]. This means that for $|\tilde{t}_\perp/t_\perp| = 0$ (within the constrained parameter space) there exists 8 semi-Dirac points located at the momenta where conditions

$$\xi_M(\mathbf{k}) = 0 \text{ and } |A| = 2|B|\sqrt{\sin^2 k_x + \sin^2 k_y} \tag{33}$$

are satisfied (see Fig. 5). In the unconstrained parameter space the system is fine tuned to criticality when $|\tilde{t}_\perp/t_\perp| = 0$. Therefore, by increasing $|\tilde{t}_\perp/t_\perp|$ we find using Eqs. (32) that each semi-Dirac point splits into two nodal points (Fig. 5). (With a different type of perturbation in the unconstrained parameter space the semi-Dirac points can also become gapped as discussed in sections 6 and 8.) After this splitting the dispersions around the nodal points obtained from Eqs. (32) are linear in both directions i.e. they are described by a massless Dirac cone,[2] and these splitting transitions where each semi-Dirac point splits into two Dirac points are described by the Hamiltonian (3). The systems satisfying a chiral symmetry are known to support gapless topological phases and topologically protected flat bands at the surfaces [35, 71–76]. Namely, it is possible to define a topological charge $Q_n^D$ for each Dirac nodal point $\mathbf{k} = \mathbf{k}_n$ in the quasiparticle spectrum

$$Q_n^D = -\frac{i}{2\pi} \oint_{\mathbf{k}_n} d\mathbf{k} \cdot \frac{1}{Z(\mathbf{k})} \nabla_{\mathbf{k}} Z(\mathbf{k}), \tag{34}$$

where $Z(\mathbf{k}) = \det[D(\mathbf{k})]/|\det[D(\mathbf{k})]|$, $D(\mathbf{k})$ is the block off-diagonal part of the Hamiltonian in Eq. (30), and the integral is calculated around a path enclosing the nodal point $\mathbf{k}_n$ [8, 73, 75]. These topological charges are always integers, and from the viewpoint of topological defects the nodal points can be considered as vortices in the momentum space with winding numbers $Q_n^D$. According to the bulk-boundary correspondence of topological media, $Q_n^D$ determine the momentum space structure of the topologically protected flat bands at the surface. We discuss this correspondence in detail in Sec. 4. Because each semi-Dirac point splits into two Dirac points the system supports 16 Dirac points if

$$\text{Dirac phase} \quad 0 < \left|\frac{\tilde{t}_\perp}{t_\perp}\right| < \left|\frac{\tilde{t}_\perp}{t_\perp}\right|_{c1} \equiv \frac{|A| - 2|B|\sin k_F}{|A| + 2|B|\sin k_F}. \tag{35}$$

At $|\tilde{t}_\perp/t_\perp| = |\tilde{t}_\perp/t_\perp|_{c1}$ there occurs merging transitions of Dirac points at $(k_x, k_y) =: (\pm k_F, 0), (0, \pm k_F)$ [Eq. (3)]. These Dirac points were nucleated from two different semi-Dirac points and this merging transition leads to a change of topological mirror invariants. Namely, due to the fact that $T(k_x, k_y)$ obeys mirror symmetries

$$\sigma_y T(k_x, k_y)\sigma_y = T(k_x, -k_y),$$
$$\sigma_x T(k_x, k_y)\sigma_x = T(-k_x, k_y), \tag{36}$$

the BdG Hamiltonian (24) obeys mirror symmetries

$$M_x^\dagger H(k_x, k_y)M_x = H(k_x, -k_y), \ M_x = -i\tau_0\sigma_y,$$
$$M_y^\dagger H(k_x, k_y)M_y = H(-k_x, k_y), \ M_y = -i\tau_z\sigma_x. \tag{37}$$

The mirror symmetry operators $M_i$ $(i = x, y)$ can be diagonalized with transformations

$$U_x = \frac{1}{\sqrt{2}}\begin{pmatrix} 0 & -i & 0 & i \\ 0 & 1 & 0 & 1 \\ -i & 0 & i & 0 \\ 1 & 0 & 1 & 0 \end{pmatrix}, \ U_y = \frac{1}{\sqrt{2}}\begin{pmatrix} 0 & 1 & 0 & -1 \\ 0 & 1 & 0 & 1 \\ -1 & 0 & 1 & 0 \\ 1 & 0 & 1 & 0 \end{pmatrix}. \tag{38}$$

---

[2]In three dimensional semimetals the four-fold degenerate nodal points are called Dirac points and the two-fold degenerate nodal points are called Weyl points. In the two dimensional case the terminology is not well established: In some references both two- and four-fold degenerate nodal points are called Dirac points, whereas in other references two-fold degenerate nodal points are called Weyl points. Depending on the choice of terminology the two-fold degenerate Majorana nodal points in two dimensional superconductors can either be called Majorana-Dirac points or Majorana-Weyl points. For brevity, we often call these nodal points just Dirac points.

At the mirror lines $k_y = 0$ and $k_y = \pi$ the $y$-component of $\mathbf{k}$ satisfies $k_y = -k_y$, and therefore the mirror symmetry $M_x$ at these lines allows to block-diagonalize the Hamiltonian. Similarly at the mirror lines $k_x = 0$ and $k_x = \pi$ the mirror symmetry $M_y$ allows to block-diagonalize the Hamiltonian. Therefore the Hamiltonians at the mirror lines can be expressed as $[H_x^{0,\pi}(k_x) = H(k_x, k_y = \{0,\pi\}), H_y^{0,\pi}(k_y) = H(k_x = \{0,\pi\}, k_y)]$

$$U_x^\dagger H_x^{0,\pi}(k_x) U_x = \begin{pmatrix} H_{x,+}^{0,\pi}(k_x) & 0 \\ 0 & H_{x,-}^{0,\pi}(k_x) \end{pmatrix} \tag{39}$$

and

$$U_y^\dagger H_y^{0,\pi}(k_x) U_y = \begin{pmatrix} H_{y,+}^{0,\pi}(k_x) & 0 \\ 0 & H_{y,-}^{0,\pi}(k_x) \end{pmatrix}. \tag{40}$$

Because the chiral symmetry operator $C$ [Eq. (29)] commutes with the mirror symmetry operators $M_x$ and $M_y$ [Eqs. (37)] the transformations (38) also block-diagonalize the chiral symmetry operator

$$U_{x(y)}^\dagger C U_{x(y)} = \begin{pmatrix} C_{x(y),+} & 0 \\ 0 & C_{x(y),-} \end{pmatrix}. \tag{41}$$

Therefore, each block of the Hamiltonian now obeys a chiral symmetry

$$C_{x(y),\pm} H_{x(y),\pm}^{0,\pi}(k_{x(y)}) C_{x(y),\pm} = -H_{x(y),\pm}^{0,\pi}(k_{x(y)}). \tag{42}$$

In the formulation of a mirror invariant we can concentrate only on one of the blocks $H_{x(y),+}^{0,\pi}$ in each mirror line. Due to the presence of chiral symmetries $C_{x(y),+}$ we can block-off diagonalize the Hamiltonians with the transformations

$$\tilde{U}_x^\dagger H_{x,+}^{0,\pi}(k_x) \tilde{U}_x = \begin{pmatrix} 0 & D_x^{0,\pi}(k_x) \\ [D_x^{0,\pi}(k_x)]^\dagger & 0 \end{pmatrix},$$

$$\tilde{U}_y^\dagger H_{y,+}^{0,\pi}(k_y) \tilde{U}_y = \begin{pmatrix} 0 & D_y^{0,\pi}(k_y) \\ [D_y^{0,\pi}(k_y)]^\dagger & 0 \end{pmatrix}, \tag{43}$$

where

$$\tilde{U}_x = \frac{1}{\sqrt{2}} \begin{pmatrix} 1 & -1 \\ 1 & 1 \end{pmatrix}, \quad \tilde{U}_y = \frac{1}{\sqrt{2}} \begin{pmatrix} i & -i \\ 1 & 1 \end{pmatrix}, \tag{44}$$

and

$$D_x^{0,\pi}(k_x) = \left\{ \xi(k_x, k_y) - i\left(1 - |\tilde{t}_\perp/t_\perp|^2\right)\Delta(k_x, k_y) - i\left(1 + |\tilde{t}_\perp/t_\perp|^2\right)d_y(k_x, k_y) \right\}\Big|_{k_y = 0,\pi}$$

$$D_y^{0,\pi}(k_y) = \left\{ \xi(k_x, k_y) - i\left(1 - |\tilde{t}_\perp/t_\perp|^2\right)\Delta(k_x, k_y) - i\left(1 + |\tilde{t}_\perp/t_\perp|^2\right)d_x(k_x, k_y) \right\}\Big|_{k_x = 0,\pi} \tag{45}$$

The four different topological mirror invariants $W_{x(y)}^{M,\{0,\pi\}}$ for each mirror line can be defined as

$$W_{x(y)}^{M,\{0,\pi\}} = \frac{-i}{2\pi} \int_{-\pi}^{\pi} dk_{x(y)} \frac{1}{Z_{x(y)}^{0,\pi}(k_{x(y)})} \frac{dZ_{x(y)}^{0,\pi}(k_{x(y)})}{dk_{x(y)}},$$

$$Z_{x(y)}^{0,\pi}(k_{x(y)}) = \frac{\det[D_{x(y)}^{0,\pi}(\mathbf{k}_{x(y)})]}{|\det[D_{x(y)}^{0,\pi}(\mathbf{k}_{x(y)})]|}. \tag{46}$$

From these equations we find that the mirror invariants $W_x^{M,0}$ and $W_y^{M,0}$ change from zero to $-1$ and $1$, respectively, at $|\tilde{t}_\perp/t_\perp| = |\tilde{t}_\perp/t_\perp|_{c1}$, so that the system is in a topologically nontrivial mirror superconducting phase (TMS) [61–63] for

$$\text{TMS} \quad \left|\frac{\tilde{t}_\perp}{t_\perp}\right|_{c1} < \left|\frac{\tilde{t}_\perp}{t_\perp}\right| < \left|\frac{\tilde{t}_\perp}{t_\perp}\right|_{c2} \equiv \frac{2\sqrt{2}|B|\sin\left(\frac{k_F}{\sqrt{2}}\right) - |A|}{2\sqrt{2}|B|\sin\left(\frac{k_F}{\sqrt{2}}\right) + |A|}, \tag{47}$$

where there exists additionally also 8 Dirac points. The Dirac points give rise to Majorana flat bands at the edge of the system, where the momentum space structure depends on the direction of the edge in a similar way as discussed in Sec. 4. Due to the mirror superconductivity there exists also helical Majorana edge modes if the direction of the edge is such that the edge remains invariant in the mirror symmetry operation [61–63].

For $|\tilde{t}_\perp/t_\perp| > |\tilde{t}_\perp/t_\perp|_{c2}$ the system is in a fully gapped topologically nontrivial phase in class DIII, which is topologically equivalent to a helical $p$-wave superconductor [77]. It supports helical Majorana edge modes independently on the direction of the edge.

# 3 Model

The Hamiltonian for the system $\hat{H} = \hat{H}_I + \hat{H}_{t2g} + \hat{H}_{I,t2g}$ consists of the Hamiltonians $\hat{H}_I$ and $\hat{H}_{t2g}$ for the doped superconducting iridate and the thin layer of the $t_{2g}$-ES, respectively, and the Hamiltonian $\hat{H}_{I,t2g}$ describing the tunneling between them.

$Sr_2IrO_4$ is a layered material so that it can be described with a standard two-dimensional single band Hubbard model on a square lattice where the spin is replaced by the pseudospin degree of freedom [Eq. (1)][3] [20, 21]. It supports a Mott insulator phase at half-filling [22–24], and therefore according to the resonating valence bond theory of high-$T_c$ superconductivity [78–81] one expect that the electron doped $Sr_2IrO_4$ will support a high-$T_c$ $d$-wave superconducting phase where the usual spin-singlet Cooper pairs are now just replaced by pseudospin singlets [21, 24, 25]. The BdG Hamiltonian for the doped superconducting $Sr_2IrO_4$ in the Nambu-pseudospin basis $\Psi_{\mathbf{k}}^\dagger = (f_{\mathbf{k}\Uparrow}^\dagger, f_{\mathbf{k}\Downarrow}^\dagger, f_{-\mathbf{k}\Uparrow}, f_{-\mathbf{k}\Downarrow})$ can then be written as

$$H_I(\mathbf{k}) = \begin{pmatrix} \xi_I(\mathbf{k})\sigma_0 & i\Delta_I(\mathbf{k})\sigma_y \\ -i\Delta_I(\mathbf{k})\sigma_y & -\xi_I(\mathbf{k})\sigma_0 \end{pmatrix}, \tag{48}$$

where $\xi_I(\mathbf{k}) = -2t_I(\cos k_x + \cos k_y) - 4t_I' \cos k_x \cos k_y - \mu_I$ and $\Delta_I(\mathbf{k}) = \Delta_0(\cos k_x - \cos k_y)$ describe the single particle dispersion and the momentum dependence of the superconducting order parameter in the iridate layer, respectively. We assume that the tight-binding parameters satisfy $t_I' = 0.23t_I$ [21], which produces Fermi surfaces with a similar shape as observed in experiments [82]. The hopping $t_I$ is renormalized by the strong correlations and it depends on the doping level. We assume that $\mu_I = 0.9t_I$, which corresponds to the electron doping chosen close to the optimal doping for superconductivity, which according to Ref. [25] is around 20%. Based on the bare value of the hopping amplitude [21] and the crudest approximation for the renormalization of the hopping amplitude [80], we estimate $t_I \sim 0.05$ eV. We assume $\Delta_0 = 0.2t_I$, which is consistent with the experimentally observed $d$-wave gap [38] and theoretical estimates based on the resonating valence bond theory [80].

---

[3]We do not take into account the lattice distortion caused by the rotation of the oxygen octahedra around the Iridium sites by $\pm 11^o$. It is known that this kind of structural distortion can cause non-perturbative effects in $Sr_2RhO_4$ because it couples the $|xy\rangle$ and $|x^2 - y^2\rangle$ bands, which both exist in the vicinity of Fermi level in this material [40]. However, in $Sr_2IrO_4$ the $|x^2 - y^2\rangle$ band does not exist in the vicinity of Fermi level and therefore we expect that this structural distortion can be treated perturbatively, and it will not change our results qualitatively.

For the $t_{2g}$-ES we first consider the 4d TMOs $Sr_2RhO_4$ and $Sr_2RuO_4$ in their normal states. In these systems the intraionic spin-orbit coupling is significantly smaller than in $Sr_2IrO_4$, and as a consequence also the correlation effects are expected to be much weaker. While the $Sr_2RhO_4$ is observed to be metallic down to the lowest experimentally accessible temperatures, $Sr_2RuO_4$ supports an interesting superconducting phase at low temperatures, where the order parameter is of multi-orbital nature and not yet fully understood [43–46]. These intrinsic superconducting correlations can be neglected as a first approximation if the induced order parameter from the $Sr_2IrO_4$ is much larger than the intrinsic order parameter of $Sr_2RuO_4$ or the temperature is much larger than the critical temperature of $Sr_2RuO_4$. Because the critical temperature and superconducting order parameter are much larger for $Sr_2IrO_4$ than for $Sr_2RuO_4$, there exists a large parameter regime where these conditions are satisfied. The advantages of $Sr_2RuO_4$ in comparison to $Sr_2RhO_4$ are the very pure crystal structure with long mean free path and the fact that in $Sr_2RhO_4$ a structural distortion causes mixing of $|xy\rangle$ and $|x^2-y^2\rangle$ orbitals so that the $|xy\rangle$ band is completely below the fermi level [40,41]. Therefore, in the following we mainly use the bulk tight-binding parameters for $Sr_2RuO_4$ although we expect that our results are also qualitatively applicable to $Sr_2RhO_4$ especially if it is slightly hole doped so that the $|xy\rangle$ band becomes active. Via epitaxial stabilization it is also possible to realize thin films of $Ba_2RuO_4$ which are isostructural and isoelectronic to $Sr_2RuO_4$ [83], so that this material is also a suitable candidate for metallic $t_{2g}$-ES. Moreover, it is possible to control the lattice constants and the electronic structures of $Sr_2RuO_4$ and $Ba_2RuO_4$ in a disorder-free manner by growing thin films of these materials on lattice mismatched substrates [83].

The use of the bulk tight-binding parameters for $Sr_2RuO_4$ is a simplification because the Fermi surfaces have a weak dependence on the out-of-plane momentum [84] and in thin layers the $\gamma$ band can be closer to a Lifshitz transition than in the bulk system [83]. However, these effects and the number of atomic layers in the $t_{2g}$-ES are not important for our qualitative conclusions, because the semi-Dirac points are robust as long as the symmetries are present and the tunneling matrices are topologically nontrivial as discussed in the previous section. The modifications of the tight-binding parameters result in a renormalizion of the critical points of phase transitions in the phase diagrams discussed below. Thus, the possibility to control the tight-binding parameters for example by chemical doping, gates and strain engineering is interesting because it may allow to drive the system through topological phase transitions in a controlled way.

The Hamiltonian for the metallic $t_{2g}$-ES in the basis $c_{\mathbf{k}}^{\dagger} = (c_{\mathbf{k}yz\uparrow}^{\dagger}, c_{\mathbf{k}yz\downarrow}^{\dagger}, c_{\mathbf{k}xz\uparrow}^{\dagger}, c_{\mathbf{k}xz\downarrow}^{\dagger}, c_{\mathbf{k}xy\uparrow}^{\dagger}, c_{\mathbf{k}xy\downarrow}^{\dagger})$ [here $c_{\mathbf{k}yz\sigma}^{\dagger}$, $c_{\mathbf{k}xz\sigma}^{\dagger}$ and $c_{\mathbf{k}xy\sigma}^{\dagger}$ are the creation operators for $|yz\rangle$, $|zx\rangle$ and $|xy\rangle$ bands, respectively] can be written as

$$\hat{H}_{t2g} = \sum_{\mathbf{k}} c_{\mathbf{k}}^{\dagger} h_0(\mathbf{k}) c_{\mathbf{k}}, \tag{49}$$

where

$$h_0(\mathbf{k}) = \begin{pmatrix} \xi_{yz}(\mathbf{k})\sigma_0 & \xi_D(\mathbf{k})\sigma_0 + \frac{i\lambda}{2}\sigma_z & -\frac{i\lambda}{2}\sigma_y \\ \xi_D(\mathbf{k})\sigma_0 - \frac{i\lambda}{2}\sigma_z & \xi_{xz}(\mathbf{k})\sigma_0 & i\frac{\lambda}{2}\sigma_x \\ \frac{i\lambda}{2}\sigma_y & -i\frac{\lambda}{2}\sigma_x & \xi_{xy}(\mathbf{k})\sigma_0 \end{pmatrix}.$$

Here

$$\xi_{yz/xz}(\mathbf{k}) = -2t_L \cos k_{y/x} - 2t_S \cos k_{x/y} - \mu,$$
$$\xi_{xy}(\mathbf{k}) = -2t_L(\cos k_x + \cos k_y) - 4t_L' \cos k_x \cos k_y - \Delta_E - \mu$$

and

$$\xi_D(\mathbf{k}) = -4t_D \sin k_x \sin k_y.$$

We assume the relative strengths of the tight-binding parameters from Ref. [45]. They can be understood in the following way. The $|xy\rangle$ orbitals are in plane producing equivalent tight-binding hopping parameters $t_L$ and $t'_L = 0.4 t_L$ in all directions. The two other orbitals have lobes both in plane and out of plane giving rise to one large $t_L$ and one small $t_S = 0.1 t_L$ hopping elements. Additionally the $|yz\rangle$ and $|xz\rangle$ orbitals are hybridized by a diagonal hopping $t_D = 0.1 t_L$. Due to the layered structure the $|xy\rangle$ band is lowered in energy by $\Delta_E = 0.2 t_L$ with respect to the other bands. The chemical potential is chosen so that the bands have correct fillings, and it is approximately $\mu = 1.1 t_L$. By comparing the experimentally measured value of the spin-orbit coupling to the width of the energy bands [42,43] we estimate that the strength of the spin-orbit coupling in $Sr_2RuO_4$ is $\lambda = 0.17 t_L$. The Fermi surfaces for $\alpha$, $\beta$ and $\gamma$ bands obtained by diagonalizing the Hamiltonian (49) and using these tight-binding parameters are shown in Fig. 1(b), and they are in good agreement with experimentally measured Fermi surfaces in $Sr_2RuO_4$ [42]. The band width in $Sr_2RuO_4$ is significantly larger than the estimated renormalized hopping amplitude in $Sr_2IrO_4$, so that we choose $t_I = 0.1 t_L$. In the following we will also consider a $t_{2g}$-ES with various values of $\lambda$ but if not otherwise stated we use the value $\lambda = 0.17 t_L$.

We will also consider situations where the $t_{2g}$-ES supports intrinsic superconductivity in sections 7 and 8. When this kind of intrinsic order parameter is present the BdG Hamiltonian for the $t_{2g}$-ES in the Nambu space $[\Phi^\dagger_{\mathbf{k}} = (c^\dagger_{\mathbf{k}}, c_{\mathbf{k}})]$ becomes

$$H^{sc}_{t2g} = \begin{pmatrix} h_0(\mathbf{k}) & \Delta_{t2g}(\mathbf{k}) \\ \Delta^\dagger_{t2g}(\mathbf{k}) & -h^T_0(-\mathbf{k}) \end{pmatrix}, \tag{50}$$

where the intrinsic order parameter $\Delta_{t2g}(\mathbf{k})$ should be solved self-consistently taking into account also the induced order parameter from the $Sr_2IrO_4$. We postpone the discussion of the self-consistently solved intrinsic order parameter $\Delta_{t2g}(\mathbf{k})$ to Section 7.

We now turn to the description of the tunneling Hamiltonian $\hat{H}_{IM}$ when a heterostructure consisting of the $Sr_2IrO_4$ and $t_{2g}$-ES is formed.[4] This can be obtained by identifying the tunneling paths between the layers allowed by the symmetries and projecting the Hamiltonian obtained this way to the pseudospin orbitals [Eq. (1)]. By taking into account only the dominant tunneling paths we obtain after a lengthy calculation[5]

$$\hat{H}_{IM} = \sum_{\mathbf{k}, \theta = yz, xz, xy} (f^\dagger_{\mathbf{k}\Uparrow}, f^\dagger_{\mathbf{k}\Downarrow}) T_{f,\theta}(\mathbf{k}) \begin{pmatrix} c_{\mathbf{k}\theta\uparrow} \\ c_{\mathbf{k}\theta\downarrow} \end{pmatrix} + h.c., \tag{51}$$

where the tunneling matrices are

$$T_{f,xy}(\mathbf{k}) = \frac{A t_\perp}{\sqrt{3}} \sigma_0 + \frac{2B t_\perp}{\sqrt{3}} (\sigma_x \sin k_y - \sigma_y \sin k_x),$$

$$T_{f,xz}(\mathbf{k}) = \frac{2iB t_\perp}{\sqrt{3}} \sigma_0 \sin k_y + \frac{i t_\perp}{\sqrt{3}} \sigma_x,$$

$$T_{f,yz}(\mathbf{k}) = \frac{2iB t_\perp}{\sqrt{3}} \sigma_0 \sin k_x - \frac{i t_\perp}{\sqrt{3}} \sigma_y. \tag{52}$$

Here $t_\perp$ describes the interlayer hopping parameter for a process where a tunneling occurs between two orbitals directly on top of each other which have lobes out of plane.[6] Therefore,

---

[4]We do not expect the interface effects to be dramatic because we consider materials with similar crystal structures. Nevertheless, when a heterostructure of two different materials is constructed the layers will always undergo electronic reconstruction where electrons will be transferred from one layer to the other. As the lowest order approximation this will only result in renormalization of the tight-binding parameters, and small renormalization of the parameters will not influence our results qualitatively.

[5]See the Appendices for more details.

[6]See footnote 5.

it is expected to be the largest interlayer hopping amplitude. The dimensionless parameter $A$ describes the reduction of the tunneling amplitude when the tunneling occurs between two orbitals which have lobes only in the plane and the dimensionless parameter $B$ describes the reduction when the tunneling occurs between two orbitals which are not directly on top of each other but have lobes out of plane.[7] Therefore we expect $|A|, |B| < 1$, but their signs and relative magnitudes are not known. Due to the layered structure we expect that $t_\perp \ll t_L$, and in the following we use $t_\perp = 0.04 t_L$. This tunneling Hamiltonian (51) is consistent with the symmetries of the system.[8]

## 4  Topological properties of the superconducting $Sr_2IrO_4$

As discussed above $Sr_2IrO_4$ supports a $d$-wave superconducting phase so that the only difference to the cuprates is that the Cooper pairs are formed as pseudospin singlets, where the pseudospin describes entangled orbital and spin degrees of freedom. This has no consequences if $Sr_2IrO_4$ is isolated and therefore the topological properties of isolated $Sr_2IrO_4$ are similar to the other $d$-wave superconductors studied earlier [71, 72]. In this section we will briefly review these properties so that they can be compared to the topological properties of the heterostructures in the following sections.

The Hamiltonian (48) satisfies particle-hole [Eq. (27)], time-reversal [Eq. (28)] and chiral [Eq. (29)] symmetries, where the Pauli matrices $\tau_i$ and $\sigma_i$ act in the particle-hole and the pseudospin spaces, respectively. Because of the chiral symmetry, the Hamiltonian (48) can be block-off-diagonalized into a form

$$V^\dagger H_I(\mathbf{k})V = \begin{pmatrix} 0 & D_I(\mathbf{k}) \\ D_I^\dagger(\mathbf{k}) & 0 \end{pmatrix}, \tag{53}$$

where $D_I(\mathbf{k}) = [\xi_I(\mathbf{k}) + i\Delta_I(\mathbf{k})]\sigma_y$. It is easy to see that there exists nodes in the quasiparticle spectrum if $\xi_I(\mathbf{k}) = 0$ and $\Delta_I(\mathbf{k}) = 0$. Because $\Delta_I(\mathbf{k}) = 0$ along the lines at $k_x = \pm k_y$, nodes are found at the four momenta $\mathbf{k}_n =: (\pm k_0, \pm k_0)$ where also $\xi_I(\mathbf{k}_n) = 0$ [see Fig. 6(a)]. The low-energy theory around the node at $\mathbf{k} = \mathbf{k}_n$ consist of two copies of a massless Dirac Hamiltonian on top of each other

$$H = (\mathbf{k} - \mathbf{k}_n) \cdot \left[ \nabla_{\mathbf{k}} \xi_I(\mathbf{k})\big|_{\mathbf{k}=\mathbf{k}_n} \sigma_x + \nabla_{\mathbf{k}} \Delta_I(\mathbf{k})\big|_{\mathbf{k}=\mathbf{k}_n} \sigma_y \right], \tag{54}$$

so that the velocity of the massless particle in the direction perpendicular to the Fermi line is determined by $\nabla_{\mathbf{k}} \xi(\mathbf{k})\big|_{\mathbf{k}=\mathbf{k}_n}$ and along the Fermi line by $\nabla_{\mathbf{k}} \Delta(\mathbf{k})\big|_{\mathbf{k}=\mathbf{k}_n}$. The BdG Hamiltonian (48) contains a particle-hole redundancy, which gives rise to a Majorana constraint. Therefore in the description of the theory where the nodal points are four-fold degenerate the quasiparticles should be considered as Majorana fermions. In the case of a singlet superconductor with $SU(2)$-symmetry, such as Hamiltonian (48), it is possible to describe the quasiparticles with usual fermion operators without the Majorana constraint and in this kind of description the nodal points are just two-fold degenerate. However, the description with the Majorana constraint is necessary when the system is tunnel coupled to $t_{2g}$-ES, and therefore we will consistently use the Majorana fermion basis everywhere.

The systems satisfying a chiral symmetry are known to support gapless topological phases and topologically protected flat bands at the surfaces [35, 71–76]. To understand these flat bands, we utilize the topological charges $Q_n^D$ for each nodal point $\mathbf{k}_n$, which can be calculated

---

[7]See the Appendices for more details.
[8]See footnote 7.

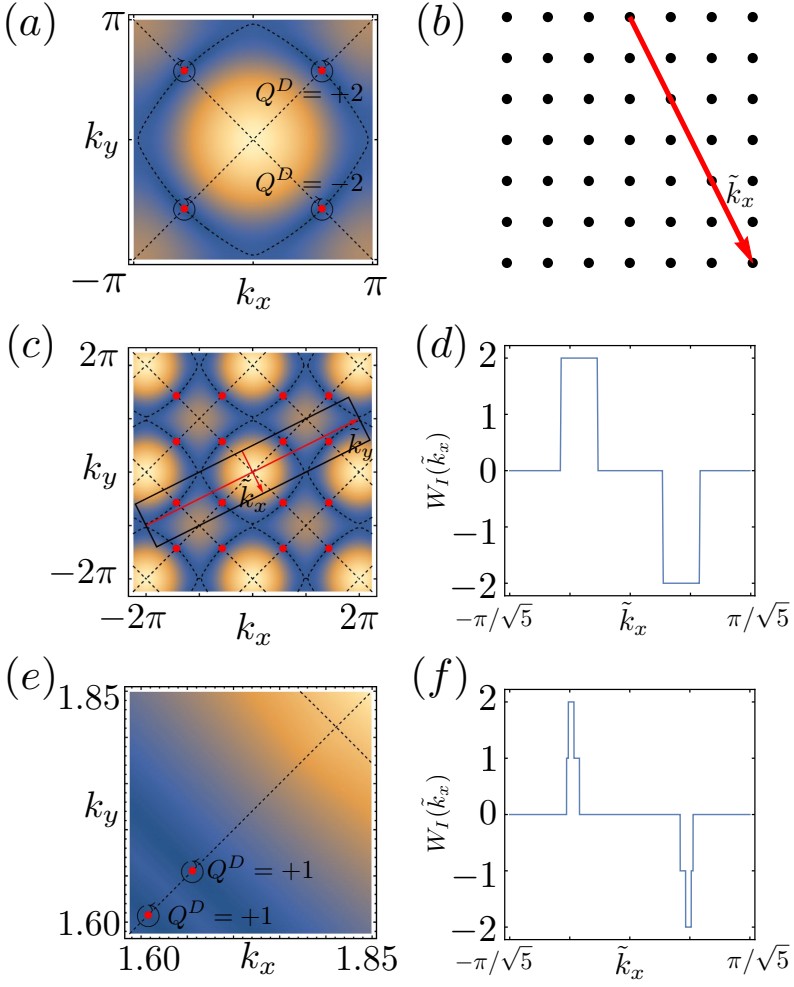

Figure 6: (a) Quasiparticle dispersion $E(\mathbf{k}) = \sqrt{\xi_I^2(\mathbf{k}) + \Delta_I^2(\mathbf{k})}$ for the Hamiltonian (48) describing an isolated $d$-wave superconductor. There exists four nodes at momenta $\mathbf{k}_n =: (\pm k_0, \pm k_0)$ where $\xi_I(\mathbf{k}_n) = 0$. The topological charges for these Dirac nodes are $Q^D = \pm 2$. (b) The edge state spectrum depends on the direction of the edge. We illustrate the appearance of the flat bands at the edge for the specific direction shown in the figure. (c) The period $d$ along the edge is in general different from the lattice constant. Therefore the momentum along the edge is restricted to the reduced Brillouin zone $-\pi/d \leq \tilde{k}_x \leq \pi/d$. In order to cover the full Brillouin zone we need to consider an extended Brillouin zone in the perpendicular direction $-\pi d \leq \tilde{k}_y \leq \pi d$. (d) The topological invariant $W_I(\tilde{k}_x)$ can be defined as a function of the momentum along the edge and it determines the number of Majorana zero energy states at that momentum. The topological invariant $W_I(\tilde{k}_x)$ changes only at the values of $\tilde{k}_x$ corresponding to the projected nodal points $\mathbf{k}_n$, and the corresponding jumps in $W_I(\tilde{k}_x)$ are determined by the topological charges $Q^D$. (e) If the $Sr_2IrO_4$ layer is tunnel coupled to a $t_{2g}$-ES the four-fold degenerate Majorana nodal points are split into two-fold degenerate nodes. Each of these nodes carries a topological charge $Q^D = \pm 1$. (f) The splitting of the four-fold degenerate nodal points leads to the appearance of odd number of Majorana flat bands at edge in certain intervals of $\tilde{k}_x$. The value of $t_\perp$ has been increased fourfold compared to the value in the text to make the changes clearly visible. The tight-binding parameters are described in the text. For the tunneling amplitudes we have used values $A = -0.2$ and $B = -0.62$.

as discussed in Sec. 2 [Eq. (34)]. The topological charges $Q_n^D = \pm 2$ obtained for the nodal points of the $d$-wave superconductor are shown in Fig. 6(a).

According to the bulk-boundary correspondence of topological media, these topological charges determine the momentum space structure of the topologically protected flat bands at the surface. Namely, we can consider an arbitrary direction of the edge of the sample as illustrated in Fig. 6(b). There exists a translational symmetry along the direction of the edge, which means that the momentum $\tilde{k}_x$ along the edge is a good quantum number. The period of the system $\tilde{d}$ along this direction is in general different from the lattice constant (normalized to 1), and it depends on the chosen direction. Therefore, we can restrict $\tilde{k}_x$ to the reduced Brillouin zone $|\tilde{k}_x| \leq \pi/\tilde{d}$. Moreover, we can perform a coordinate transformation of Hamiltonian (53) $\mathbf{k} = \mathbf{k}(\tilde{k}_x, \tilde{k}_y)$ so that we can identify a matrix $D_I(\tilde{k}_x, \tilde{k}_y)$ and correspondingly $Z_I(\tilde{k}_x, \tilde{k}_y)$. These quantities are periodic as a function of the momentum component perpendicular to the edge $\tilde{k}_y$ with a period $2\pi\tilde{d}$ defining an extended Brillouin zone in this direction $|\tilde{k}_y| \leq \pi\tilde{d}$ [see Fig. 6(c)]. Therefore we can define a topological invariant (winding number)

$$W_I(\tilde{k}_x) = -\frac{i}{2\pi} \int_{-\pi\tilde{d}}^{\pi\tilde{d}} d\tilde{k}_y \frac{1}{Z_I(\tilde{k}_x, \tilde{k}_y)} \frac{dZ_I(\tilde{k}_x, \tilde{k}_y)}{d\tilde{k}_y}, \tag{55}$$

for all values of $\tilde{k}_x$ where there is no gap closing as a function of $\tilde{k}_y$, and this invariant is an integer which determines the number of Majorana zero energy edge states for each $\tilde{k}_x$. Moreover, $W_I(\tilde{k}_x)$ can change only at the momenta $\tilde{k}_x$ where there is a gap closing as a function of $\tilde{k}_y$. Therefore, a fixed number of zero energy Majorana edge states exist in the interval of $\tilde{k}_x$ in between the projected nodal points, forming Majorana flat bands at the edge. The jumps in $W_I(\tilde{k}_x)$ occurring at the edge momenta $\tilde{k}_x$ corresponding to the projected nodal points $\mathbf{k}_n$ are given by the topological charges $Q_n^D$. The number of flat bands $W_I(\tilde{k}_x)$ as a function of $\tilde{k}_x$ for an isolated $d$-wave superconductor and a particular direction of the edge is illustrated in Fig. 6(d).

The linearly dispersing Majorana-Dirac points in the $Sr_2IrO_4$ layer discussed in this section are present also when this system is coupled to the $t_{2g}$-ES. The only possible effect of the tunnel coupling to the $t_{2g}$-ES is a small splitting of the four-fold degenerate Majorana nodal points into two-fold degenerate nodes with linear dispersions around each of them. To illustrate how this splitting occurs we first find that the low-energy effective Hamiltonian for the $Sr_2IrO_4$ layer in presence of the tunnel coupling to the $t_{2g}$-ES can be expressed as

$$H_I^{\text{eff}}(\mathbf{k}) = \begin{pmatrix} \xi_I(\mathbf{k})\sigma_0 + \delta H_I(\mathbf{k}) & i\Delta_I(\mathbf{k})\sigma_y \\ -i\Delta_I(\mathbf{k})\sigma_y & -\xi_I(\mathbf{k})\sigma_0 - \delta H_I^T(-\mathbf{k}) \end{pmatrix}, \tag{56}$$

where $\delta H_I(\mathbf{k})$ is the self-energy induced by the coupling to the $t_{2g}$-ES. In the vicinity of the nodal points $\mathbf{k} = \mathbf{k}_n$ the self-energy can be evaluated at the zero energy, and it can be expressed as

$$\delta H_I(\mathbf{k}) = -\sum_n \frac{1}{\xi_n(\mathbf{k})} T_n^\dagger(\mathbf{k}) T_n(\mathbf{k}), \tag{57}$$

where $T_n(\mathbf{k})$ are the tunneling matrices from the iridate to different bands ($n = \alpha, \beta, \gamma$) in the $t_{2g}$-ES and $\xi_n(\mathbf{k})$ are the dispersions of these bands. Furthermore, by utilizing the time-reversal symmetry of the tunneling matrix, which in a suitable basis[9] can be written as $\sigma_y T_n^*(-\mathbf{k})\sigma_y = T_n(\mathbf{k})$, and the inversion symmetry $\xi_n(\mathbf{k}) = \xi_n(-\mathbf{k})$, we

---

[9]In particular, the time-reversal symmetry will take this form when we fix the overall phase of the single particle eigenstates of Hamiltonian (49) in such a way that the eigenvector components corresponding to the $|xy\rangle$ orbitals are real and positive. We will use this gauge choice in all calculations presented in this manuscript. If $\lambda = 0$ some of the eigenstates will have zero weight in the $|xy\rangle$ orbitals, but this situation can be considered as the limit $\lambda \to 0$. See the Appendices for more details.

find that the Hamiltonian (56) satisfies particle-hole, time-reversal and chiral symmetries. Therefore, it can be block-off-diagonalized, and the block-off-diagonal matrix is now $D_I^{\text{eff}}(\mathbf{k}) = [\xi_I(\mathbf{k})\sigma_0 + \delta H_I(\mathbf{k})]\sigma_y + i\Delta_I(\mathbf{k})\sigma_y$. This way we can find the nodal points for Hamiltonian (56) and compute the topological charges for them. We generically find that each of the four-fold degenerate Majorana nodal points at $\mathbf{k}_n$ with topological charge $Q_n^D = \pm 2$ splits into two two-fold degenerate Majorana nodal points at $\mathbf{k}_{nA}$ and $\mathbf{k}_{nB}$ each carrying a topological charge $Q_{nA}^D = Q_{nB}^D = \pm 1$. This leads to an appearance of odd number of Majorana flat bands at the edge in the interval of edge momenta between the projected nodal points $\tilde{k}_{x,nA}$ and $\tilde{k}_{x,nB}$. The splitting of the nodal points and the Majorana flat band spectrum at the edge are illustrated in Fig. 6(e) and (f) for particular values of the tunneling amplitudes.

In the following sections we do not concentrate on these nodes appearing in the $Sr_2IrO_4$ layer since they are present in all regimes of the parameters considered in this manuscript.

# 5 Induced order parameters in the $t_{2g}$-ES and the appearance of robust semi-Dirac points

The main goal of this section is to show that robust Majorana semi-Dirac points appear in the heterostructure consisting of $Sr_2IrO_4$ tunnel coupled to a $t_{2g}$-ES layer as a result of the mixture of the induced singlet and triplet order parameters. We also generalize the results of Sec. 2 to more complicated situations and we identify the conditions under which the semi-Dirac points are stable in this kind of heterostructures. Moreover, we will compute the phase-diagrams to demonstrate that the stable semi-Dirac points appear in a large portion of the parameter space of the model.

The momentum dependent energy gap for the full model is shown in Fig. 7 for a specific choice of the microscopic parameters illustrating the type of nodal points generically present in the spectrum in one quarter of the Brillouin zone (the other quarters are related to each other via the mirror symmetries). There exists 8 Dirac points (two for each band of the model) at the diagonal lines where $\Delta_I(\mathbf{k}) = 0$. These Dirac points are present independently of the choice of model parameters, and they are similar to the Dirac points discussed in Sec. 4. Therefore, in the following we focus on the additional nodal points appearing outside these high-symmetry lines, and we show that they are semi-Dirac points localized in the $t_{2g}$-ES layer similarly as in Sec. 2.

In order to study the nature of the nodal points localized in the $t_{2g}$-ES layer we first notice that if the $t_{2g}$-ES, described by the Hamiltonian (49) is isolated, there exist both inversion symmetry and time-reversal symmetry, and therefore each band is doubly degenerate with dispersions $\xi_n(\mathbf{k}) = \xi_n(-\mathbf{k})$ ($n = \alpha, \beta, \gamma$), i.e. due to this combination of symmetries there exists Kramer's partners at each momentum separately. Thus, when we couple the $t_{2g}$-ES to the iridate, the low-energy BdG Hamiltonians for each band can be expressed as

$$H_n(\mathbf{k}) = \begin{pmatrix} \xi_n(\mathbf{k})\sigma_0 + \delta H_n(\mathbf{k}) & \Delta_{\text{ind},n}(\mathbf{k}) \\ \Delta_{\text{ind,n}}^{\dagger}(\mathbf{k}) & -\xi_n(\mathbf{k})\sigma_0 - \delta H_n^T(-\mathbf{k}) \end{pmatrix}, \tag{58}$$

where $\delta H_n(\mathbf{k})$ and $\Delta_{\text{ind},n}(\mathbf{k})$ are given by the normal and anomalous (induced superconductivity) part of the self-energy evaluated at the Fermi energy. They can be obtained from the expressions

$$\delta H_n(\mathbf{k}) = h_{n0}(\mathbf{k})\sigma_0 + \mathbf{h}_n(\mathbf{k}) \cdot \vec{\sigma}$$
$$\Delta_{\text{ind},n}(\mathbf{k}) = i[\Delta_n(\mathbf{k})\sigma_0 + \mathbf{d}_n(\mathbf{k}) \cdot \vec{\sigma}]\sigma_y \tag{59}$$

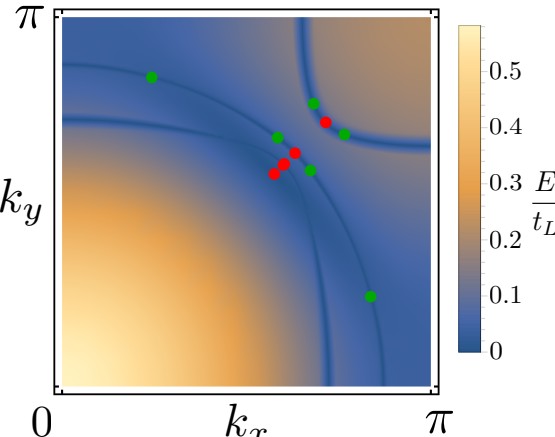

Figure 7: The momentum dependent energy gap $E(\mathbf{q})$ for the full model. The semi-Dirac nodes (green) and the Dirac nodes (red) are also shown. The splitting of the Dirac nodes along $k_x = k_y$ is too small to be visible in the plot. The tight-binding parameters are described in the text. For the tunneling amplitudes we have used values $A = 0.68$ and $B = 0.44$.

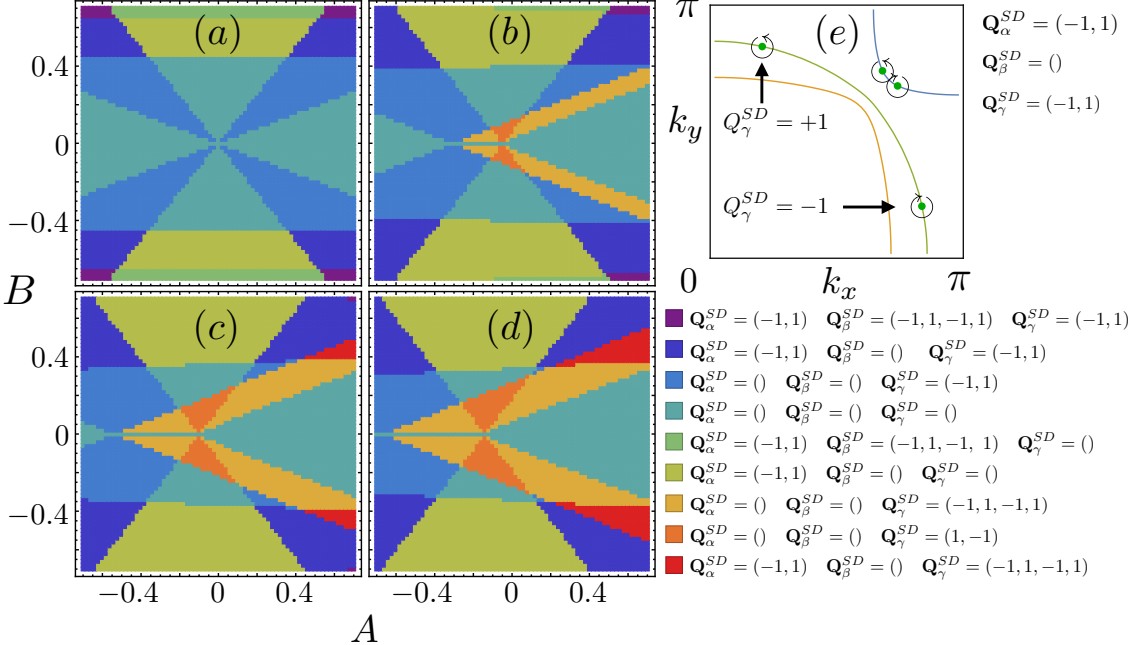

Figure 8: (a)-(d) Phase diagrams for the system as a function of the interlayer coupling strengths $A$ and $B$ for different spin orbit coupling strengths $\lambda =: 0, 0.06t_L, 0.12t_L, 0.17t_L$, respectively. Each phase is characterized by three vectors $Q_n^{SD}$ ($n = \alpha, \beta, \gamma$) labeling the charges of the semi-Dirac nodes as illustrated in (e). The semi-Dirac points shown in (e) are realised for $\lambda = 0.17t_L$ and $A = -0.68$ and $B = -0.34$. The tight-binding parameters are described in the text.

where

$$h_{n0}(\mathbf{k})\sigma_0 + \mathbf{h}_n(\mathbf{k}) \cdot \vec{\sigma} = -\frac{\xi_I(\mathbf{k})}{\xi_I^2(\mathbf{k}) + \Delta_I^2(\mathbf{k})} T_n(\mathbf{k}) T_n^\dagger(\mathbf{k})$$

$$\Delta_n(\mathbf{k})\sigma_0 + \mathbf{d}_n(\mathbf{k}) \cdot \vec{\sigma} = \frac{\Delta_I(\mathbf{k})}{\xi_I^2(\mathbf{k}) + \Delta_I^2(\mathbf{k})} T_n(\mathbf{k}) T_n^\dagger(\mathbf{k}). \tag{60}$$

Here $h_{n0}(\mathbf{k})$ and $\mathbf{h}_n(\mathbf{k})$ describe the renormalization of the dispersion and the induced spin-orbit coupling in the $t_{2g}$-ES due to the coupling to the iridate, $\Delta_n(\mathbf{k})$ and $\mathbf{d}_n(\mathbf{k})$ are the induced singlet and triplet superconducting order parameters, and $T_n(\mathbf{k})$ are the tunneling matrices from the iridate to different bands ($n = \alpha, \beta, \gamma$) in the $t_{2g}$-ES.

The semi-Dirac nodes outside the high-symmetry lines appear if the conditions $\xi_n(\mathbf{k}) = 0$ and $\det[T(\mathbf{k})] = 0$ are satisfied, and they can be understood in the same way as in Sec. 2. Namely, the tunneling matrices $T_n(\mathbf{k})$ satisfy the time-reversal (11) and two-fold rotational (12) symmetries so that $\det[T_n(\mathbf{k})] \in \mathbb{R}$. Therefore, the semi-Dirac points carry topological charges $Q^{SD} = \pm 1$ [Eq. (22)], they are robust against small perturbations, and if one considers large perturbations semi-Dirac points with opposite topological charges are always nucleated/annihilated in a pairwise manner. The low-energy theory can be expressed as

$$
\begin{aligned}
H_n^{\mathrm{eff}}(\mathbf{k}) &= h_x(\mathbf{k})\sigma_x + h_y(\mathbf{k})\sigma_y, \\
h_x(\mathbf{k}) &= \xi_n(\mathbf{k}) + h_{n0}(\mathbf{k}) - \mathrm{sgn}[\Delta_n(\mathbf{k})]\frac{\mathbf{h}_n(\mathbf{k})\cdot\mathbf{d}_n(\mathbf{k})}{|\mathbf{d}_n(\mathbf{k})|} \\
&= \xi_n(\mathbf{k}) - \frac{\xi_I(\mathbf{k})}{\xi_I^2(\mathbf{k}) + \Delta_I^2(\mathbf{k})}\frac{\det^2[T_n(\mathbf{k})]}{(|T_{11}(\mathbf{k})| + |T_{12}(\mathbf{k})|)^2}, \\
h_y(\mathbf{k}) &= \Delta_n(\mathbf{k}) - \mathrm{sgn}[\Delta_n(\mathbf{k})]|\mathbf{d}_n(\mathbf{k})| \\
&= \frac{\Delta_I(\mathbf{k})}{\xi_I^2(\mathbf{k}) + \Delta_I^2(\mathbf{k})}\frac{\det^2[T_n(\mathbf{k})]}{(|T_{11}(\mathbf{k})| + |T_{12}(\mathbf{k})|)^2}.
\end{aligned}
\tag{61}
$$

Therefore, we can expand the Hamiltonian (61) around the semi-Dirac points as

$$
H^{\mathrm{sD}} = \left[\hbar v q_x + \mathscr{C}_2(\mathscr{C}q_x + \mathscr{D}q_y)^2\right]\sigma_x + \mathscr{C}_1(\mathscr{C}q_x + \mathscr{D}q_y)^2\sigma_y.
\tag{62}
$$

Here we have used a curvilinear coordinate system, where $\xi(q_x, q_y) = \hbar v q_x$ and $q_y$ is the deviation from the nodal point along the constant $\xi(\mathbf{k})$ curves, so that $\xi(q_x, q_y)$ is independent of $q_y$. Therefore, these nodal points generically are described by a linear dispersion in the $q_x$-direction and a parabolic dispersion along $q_y$ i.e. they realize the semi-Dirac points described by Eq. (2). (We have included also an additional term proportional to $\mathscr{C}_2$, but this term is unimportant for qualitative considerations.)

In contrast to the simple model considered in Sec. 2 there can now be a varying number of semi-Dirac points present in the different bands. This gives rise to a rich phase diagram as a function of the tunneling amplitudes and spin-orbit coupling strength as shown in Fig. 8. Similar phase diagrams are expected as a function of arbitrary parameters of the model. Therefore, we expect that it is possible to tune the system through the phase transitions with the help of externally controllable parameters such as gate voltages and strain [85]. At the phase transitions between topologically distinct phases semi-Dirac points with opposite topological charges are nucleated/annihilated in pairwise manner similarly as discussed in Sec. 2.

In this section we have found semi-Dirac points, which are topologically stable in the presence of time-reversal symmetry, two-fold rotational symmetry, inversion symmetries within the layers, metallic $t_{2g}$-ES and a single active band in the superconductor. In the following sections we consider two possible ways for breaking the protection of the semi-Dirac points in a controlled manner. First we show that each semi-Dirac point can become gapped or split into two Dirac points if the metallic layer is tunnel coupled to two separate superconducting layers in a three-layer-structure. In this case there are two distinct superconducting bands so that the system is effectively coupled to a multiband superconductor. Secondly, we show that these transitions can occur also if the metallic layer is replaced with a superconductor supporting an intrinsic superconducting order parameter. These transitions demonstrate the nature of the semi-Dirac points as critical points of topological phase transitions in the unconstrained parameter space.

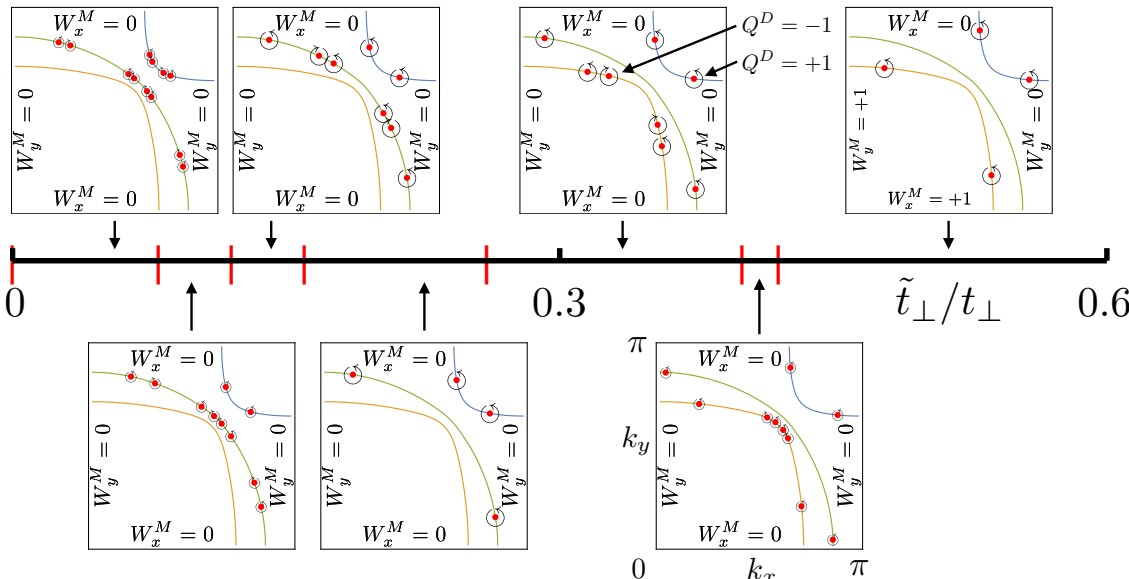

Figure 9: Phase diagram for the three layer system as a function of the tunneling strength $\tilde{t}_\perp$ to the upper $Sr_2IrO_4$ for the phase difference between the superconductors $\varphi = \pi$. As $\tilde{t}_\perp$ is increased the system goes through several phase transitions as Dirac nodes are nucleated and annihilated close to the Fermi lines of the $\alpha$, $\beta$ and $\gamma$ bands. In the last transition shown with increasing $\tilde{t}_\perp$ the system becomes a topological mirror superconductor with $W_x^{M,0} = 1$ and $W_y^{M,0} = 1$ as pairs of Dirac nodes on the Fermi line of the $\gamma$ band annihilate at the $k_x = 0$ and $k_y = 0$ lines. The tight-binding parameters are described in the text. For the tunneling amplitudes we have used the values $\tilde{A} = A = 0.68$ and $\tilde{B} = B = 0.44$.

## 6 Topological phase diagram for the three layer heterostructure

We now consider a three-layer-structure, where there are superconducting $Sr_2IrO_4$ layers both below and above the $t_{2g}$-ES layer. We assume that the lower superconducting $Sr_2IrO_4$ layer has the order parameter $\Delta_I(\mathbf{k})$ and it is tunnel coupled to the $t_{2g}$-ES via the tunneling matrices (52). Moreover, we assume that the upper superconducting $Sr_2IrO_4$ layer is identical to the lower one except that we allow the possibility of applying a phase-bias between the superconductors so that the order parameter in the upper layer is $\Delta_I(\mathbf{k})e^{i\varphi}$. In order to preserve the time-reversal symmetry we only consider the two possibilities $\varphi = 0, \pi$. Due to the nature of the orbitals the tunneling matrices between the upper layer and the $t_{2g}$-ES layer can be written as the following[9]

$$
\tilde{T}_{f,xy}(\mathbf{k}) = \frac{\tilde{A}\tilde{t}_\perp}{\sqrt{3}}\sigma_0 - \frac{2\tilde{B}\tilde{t}_\perp}{\sqrt{3}}(\sigma_x \sin k_y - \sigma_y \sin k_x),
$$

$$
\tilde{T}_{f,xz}(\mathbf{k}) = -\frac{2i\tilde{B}\tilde{t}_\perp}{\sqrt{3}}\sigma_0 \sin k_y + \frac{i\tilde{t}_\perp}{\sqrt{3}}\sigma_x,
$$

$$
\tilde{T}_{f,yz}(\mathbf{k}) = -\frac{2i\tilde{B}\tilde{t}_\perp}{\sqrt{3}}\sigma_0 \sin k_x - \frac{i\tilde{t}_\perp}{\sqrt{3}}\sigma_y. \tag{63}
$$

If the two interfaces are equivalent $\tilde{A} = A$, $\tilde{B} = B$ and $\tilde{t}_\perp = t_\perp$, the system obeys an inversion symmetry.

We assume that $\tilde{A} = A$, $\tilde{B} = B$ and consider $\tilde{t}_\perp$ as a parameter which can be controlled with the help of a tunnel barrier. (We have checked that introducing small asymmetries $\tilde{A} \neq A$ and $\tilde{B} \neq B$ does not change the results qualitatively.) Under these conditions we find that

if $\varphi = \pi$ the two induced order parameters satisfy the relations (26), whereas the signs in these relations are changed for $\varphi = 0$. As a result, we find that for $\varphi = 0$ the introduction of $\tilde{t}_\perp \neq 0$ always leads to an opening of a gap at the semi-Dirac points, and for $\varphi = \pi$ we generically obtain similar splitting transitions and phase diagrams as a function of $\tilde{t}_\perp$ as discussed in Sec. 2. The only difference is that splitting transitions can now occur in all three bands ($\alpha, \beta, \gamma$) leading to much richer phase diagrams. A phase diagram for particular values of the tunneling amplitudes as a function of $\tilde{t}_\perp$ is shown in Fig. 9 demonstrating the appearance of different types of Dirac phases and a topologically nontrivial mirror superconducting phase. We point out that the fully gapped topologically nontrivial phase cannot be realized in this system because nodal points always exist at the lines $k_x = \pm k_y$ due to the $d$-wave nature of the order parameter $\Delta_I(\mathbf{k})$ in $Sr_2IrO_4$.

## 7 Intrinsic order parameter in $Sr_2RuO_4$ in the presence of induced superconductivity from $Sr_2IrO_4$

The superconductivity in $Sr_2RuO_4$ has been studied extensively [3], because $Sr_2RuO_4$ is often considered to be a candidate material for supporting chiral $p$-wave superconductivity. However, in reality the order parameter is of multi-orbital nature and not yet fully understood [43–46], and $Sr_2RuO_4$ is likely to be characterized by a subtle competition between different phases. The important consequence of this is that there exists many nearly degenerate solutions for the superconducting order parameter making the system sensitive to all kinds of perturbations. In our system, the proximity induced superconducting order parameter acts as a strong perturbation causing a Josephson coupling between order parameters in $Sr_2IrO_4$ and $Sr_2RuO_4$. Therefore, we may expect that this coupling selects a particular order parameter in $Sr_2RuO_4$, which obeys the same symmetries as the proximity induced order parameter. Below we demonstrate that this is indeed the case by utilizing a similar approach as used in Ref. [35]. In that reference it was studied how a Zeeman field can lead to a reconstruction of the order parameter when several different superconducting order parameters are nearly degenerate, which is practically always the case in triplet superconductors. Here, the physics is very similar but the reconstruction just appears because of the Josephson coupling to the superconducting order parameter in $Sr_2IrO_4$.

The intrinsic singlet and triplet order parameters in $Sr_2RuO_4$, $\Delta_{nR}(\mathbf{k})$ and $\mathbf{d}_{nR}(\mathbf{k})$ (for each band $n = \alpha, \beta, \gamma$), can be expressed with the help of basis functions $\Delta_m^{(n)}(\mathbf{k})$, $\mathbf{d}_m^{(n)}(\mathbf{k})$ ($m = 1, 2, 3...$) as

$$\Delta_{nR}(\mathbf{k}) = \sum_m \psi_{nm} \Delta_m^{(n)}(\mathbf{k}) \tag{64}$$

and

$$\mathbf{d}_{nR}(\mathbf{k}) = \sum_m \eta_{nm} \mathbf{d}_m^{(n)}(\mathbf{k}), \tag{65}$$

where $\psi_{nm}$ and $\eta_{nm}$ are complex coefficients. We express them as $\psi_{nm} = \psi_{nm}^{\mathscr{T}} + i\psi_{nm}^{\mathscr{N}}$ ($\psi_{nm}^{\mathscr{T}}, \psi_{nm}^{\mathscr{N}} \in \mathbb{R}$) and $\eta_{nm} = \eta_{nm}^{\mathscr{T}} + i\eta_{nm}^{\mathscr{N}}$ ($\eta_{nm}^{\mathscr{T}}, \eta_{nm}^{\mathscr{N}} \in \mathbb{R}$). The superscripts refer to the parts of the order parameter obeying $\mathscr{T}$ and not obeying $\mathscr{N}$ time-reversal symmetry. The basis functions $\Delta_m^{(n)}(\mathbf{k})$, $\mathbf{d}_m^{(n)}(\mathbf{k})$ can be obtained by projecting the most general singlet and triplet order parameters into the irreducible representations of the symmetry group $G$ of the model [2, 86].

They can be written as

$$\Delta_1^{(n)}(\mathbf{k}) = a_{11}^{(n)}(\cos k_x - \cos k_y) + a_{12}^{(n)}[\cos(2k_x) - \cos(2k_y)] + a_{13}^{(n)}[\cos(2k_x)\cos k_y - \cos(2k_y)\cos k_x] + ..,$$

$$\Delta_2^{(n)}(\mathbf{k}) = a_{21}^{(n)} + .., \quad \Delta_3^{(n)}(\mathbf{k}) = a_{31}^{(n)}\sin k_x \sin k_y + .., \quad \Delta_4^{(n)}(\mathbf{k}) = a_{41}^{(n)}\sin k_x \sin k_y(\cos k_x - \cos k_y) + ..,$$

$$\mathbf{d}_1^{(n)}(\mathbf{k}) = b_{11}^{(n)}(\mathbf{e}_x \sin k_y + \mathbf{e}_y \sin k_x) + b_{12}^{(n)}(\cos k_x - \cos k_y)(\mathbf{e}_x \sin k_y - \mathbf{e}_y \sin k_x) + ..,$$

$$\mathbf{d}_2^{(n)}(\mathbf{k}) = b_{21}^{(n)}(\mathbf{e}_x \sin k_y - \mathbf{e}_y \sin k_x) + .., \mathbf{d}_3^{(n)}(\mathbf{k}) = b_{31}^{(n)}(\mathbf{e}_x \sin k_x - \mathbf{e}_y \sin k_y) + ..,$$

$$\mathbf{d}_4^{(n)}(\mathbf{k}) = b_{41}^{(n)}(\mathbf{e}_x \sin k_x + \mathbf{e}_y \sin k_y) + .., \quad \mathbf{d}_{5A}^{(n)}(\mathbf{k}) = b_{5A1}^{(n)}\mathbf{e}_z \sin k_x + .., \quad \mathbf{d}_{5B}^{(n)}(\mathbf{k}) = b_{5B1}^{(n)}\mathbf{e}_z \sin k_y + ... \tag{66}$$

These basis functions are a natural starting point for the development of the free energy expansion because the linearized gap equation is an eigenvalue problem, where the eigenvalue determines the critical transition temperature, and therefore the possible superconducting order parameters corresponding to the largest transition temperature must form a basis of an irreducible representation of the symmetry group of the model [2]. Each of these basis functions contains in general an infinite number of terms and transforms in a specific way in the symmetry transformations $g\mathbf{k}$ ($g \in G$). For singlet basis functions there exists four different possibilites distinguished by the different signs in the transformations $\Delta_m^{(n)}(-k_x, k_y) = \pm\Delta_m^{(n)}(k_x, k_y)$ and $\Delta_m^{(n)}(k_y, k_x) = \pm\Delta_m^{(n)}(k_x, k_y)$. All of these irreducible representations are one-dimensional. For the triplet order parameters there exists similarly four one-dimensional irreducible representations where the basis functions are distinguished by the different signs in the transformations $d_{m,x}^{(n)}(-k_x, k_y) = \pm d_{m,x}^{(n)}(k_x, k_y)$ and $d_{m,x}^{(n)}(k_y, k_x) = \pm d_{m,y}^{(n)}(k_x, k_y)$. Additionally, there exists one two-dimensional irreducible representation where the basis functions $\mathbf{d}_{5A}^{(n)}(\mathbf{k})$ and $\mathbf{d}_{5B}^{(n)}(\mathbf{k})$ are parallel to $\mathbf{e}_z$ and transform as $d_{5A,z}^{(n)}(-k_x, k_y) = -d_{5A,z}^{(n)}(k_x, k_y)$ and $d_{5B,z}^{(n)}(k_x, -k_y) = -d_{5B,z}^{(n)}(k_x, k_y)$. For the singlet basis functions the full expressions consistent with these transformations are given in Ref. [86] and for the triplet basis functions they can be obtained in a similar manner. The coefficients $a_{mk}^{(n)} \in \mathbb{R}$ and $b_{mk}^{(n)} \in \mathbb{R}$ are in principle variational parameters (for each band $n$ independently of the others), and they should be chosen so that the free energy is minimized. Therefore, we have included a superscript $(n)$ in all basis functions indicating that these coefficients can be different in each band $n = \alpha, \beta, \gamma$. As explained below the exact values of these coefficients are not important for our qualitative results, and therefore in the end we will select the relevant coefficients phenomenologically for each band so that the overlap between intrinsic and induced order parameters is maximized.

In the discussion above, we have assumed that the order parameter for each band can be solved independently of the other bands, and we have neglected the possibility of interband order parameters. However, due to similar arguments as presented below, we expect that the intrinsic interband order parameters do not spontaneously break any of the symmetries of the model, and therefore we do not expect them to change our results qualitatively.

The intrinsic order parameter can then be obtained by minimizing the free energy of the system with respect to $\psi_{nm}$ and $\eta_{nm}$. It is possible to show that due to symmetry reasons the order parameter pairs $(\psi_{nm}, \psi_{nm'})$, $(\eta_{nm}, \eta_{nm'})$ and $(\psi_{nm}, \eta_{nm'})$ will couple to each other in the quadratic order in the expansion of the free energy only if $m = m'$.[10] We assume that the temperature is above the critical temperature of the superconducting instability of any of the eigenmodes obtained by diagonalizing the quadratic terms in the expansion of the free energy. There exists a large range of temperatures where this assumption is valid because the critical temperature in $Sr_2RuO_4$ is expected to be much lower than the critical temperature in $Sr_2IrO_4$. Furthermore, this assumption is made in order to simplify the technical calculations and to make them analytically tractable, but in fact we expect that the same intrinsic superconducting order parameter is realized also at low temperatures because it is always favored

---

[10]See the Appendices for more details.

by the Josephson coupling, and in the absence of Josephson coupling the different options for the order parameter are nearly degenerate. With this assumption the only nonzero order parameters are the ones which obey the same symmetries as the proximity induced order parameters.[11] Therefore, $\psi_{nm} = 0$ and $\eta_{nm} = 0$ for $m \geq 2$. Moreover, the proximity induced order parameter couples only to the part of the $\psi_{n1}$ and $\eta_{n1}$ obeying time-reversal symmetry, so that also $\psi_{n1}^{\mathcal{N}} = 0$ and $\eta_{n1}^{\mathcal{N}} = 0$.[12] Therefore the free energy $F_n$ corresponding to each band $n = \alpha, \beta, \gamma$ can be written as the following[13]

$$F_n = F_{n0} - r_1^{ns}\psi_{n1}^{\mathcal{T}} - r_1^{nt}\eta_{n1}^{\mathcal{T}} + m^{ns}|\psi_{n1}^{\mathcal{T}}|^2 + m^{nt}|\eta_{n1}^{\mathcal{T}}|^2 - \kappa_1^{n\mathcal{T}}\psi_{n1}^{\mathcal{T}}\eta_{n1}^{\mathcal{T}}, \tag{67}$$

where $F_{n0}$ is a constant, $r_1^{ns}$ and $r_1^{nt}$ arise due to the Josephson coupling, $m^{ns}$ and $m^{nt}$ are the Ginzburg-Landau coefficients for singlet and triplet order parameters (renormalized due to the presence of induced order parameter), and $\kappa_1^{n\mathcal{T}}$ is the coupling between the singlet and triplet order parameters which appears only because of the presence of the induced order parameter. By minimizing the free energy we obtain

$$\psi_{n1}^{\mathcal{T}} = \frac{\kappa_1^{n\mathcal{T}} r_1^{nt} + 2m^{nt} r_1^{ns}}{4m^{ns}m^{nt} - |\kappa_1^{n\mathcal{T}}|^2} \tag{68}$$

and

$$\eta_{n1}^{\mathcal{T}} = \frac{\kappa_1^{n\mathcal{T}} r_1^{ns} + 2m^{ns} r_1^{nt}}{4m^{ns}m^{nt} - |\kappa_1^{n\mathcal{T}}|^2}. \tag{69}$$

Because the magnitudes of the order parameters depend only on the ratios of the Ginzburg-Landau coefficients, we expect that we can roughly estimate them with the following Fermi surface integrals[14]

$$r_1^{ns} = 2\int_{FS} dk \big[a_{n1}(\mathbf{k})\Delta_n(\mathbf{k}) + a_{n2}(\mathbf{k})|\mathbf{d}_n(\mathbf{k})|\big]\Delta_1^{(n)}(\mathbf{k}), \quad r_1^{nt} = 2\int_{FS} dk \big[a_{n1}(\mathbf{k})|\mathbf{d}_n(\mathbf{k})| + a_{n2}(\mathbf{k})\Delta_n(\mathbf{k})\big]\frac{\mathbf{d}_1^{(n)}(\mathbf{k})\cdot\mathbf{d}_n(\mathbf{k})}{|\mathbf{d}_n(\mathbf{k})|},$$

$$m^{ns} = \int_{FS} dk \Big[\frac{1}{4k_B T_{cs}} - a_{n1}(\mathbf{k}) - 4b_{n1}(\mathbf{k})\Big]\Delta_1^{(n)}(\mathbf{k})^2, \quad \kappa_1^{n\mathcal{T}} = 2\int_{FS} dk \big[a_{n2}(\mathbf{k}) + 4b_{n2}(\mathbf{k})\big]\Delta_1^{(n)}(\mathbf{k})\frac{\mathbf{d}_1^{(n)}(\mathbf{k})\cdot\mathbf{d}_n(\mathbf{k})}{|\mathbf{d}_n(\mathbf{k})|},$$

$$m^{nt} = \int_{FS} dk \bigg\{\bigg[\frac{1}{4k_B T_{ct}} - a_{n1}(\mathbf{k}) - a_{n2}(\mathbf{k})\frac{\Delta_n^2(\mathbf{k}) + |\mathbf{h}_n(\mathbf{k})|^2}{h_{n0}(\mathbf{k})\frac{\mathbf{h}_n(\mathbf{k})\cdot\mathbf{d}_n(\mathbf{k})}{|\mathbf{d}_n(\mathbf{k})|} + \Delta_n(\mathbf{k})|\mathbf{d}_n(\mathbf{k})|}\bigg]|\mathbf{d}_1^{(n)}(\mathbf{k})|^2$$

$$-\bigg[4b_{n1}(\mathbf{k}) - a_{n2}(\mathbf{k})\frac{\Delta_n^2(\mathbf{k}) + |\mathbf{h}_n(\mathbf{k})|^2}{h_{n0}(\mathbf{k})\frac{\mathbf{h}_n(\mathbf{k})\cdot\mathbf{d}_n(\mathbf{k})}{|\mathbf{d}_n(\mathbf{k})|} + \Delta_n(\mathbf{k})|\mathbf{d}_n(\mathbf{k})|}\bigg]\frac{[\mathbf{d}_1^{(n)}(\mathbf{k})\cdot\mathbf{d}_n(\mathbf{k})]^2}{|\mathbf{d}_n(\mathbf{k})|^2}\bigg\},$$

$$a_{n1}(\mathbf{k}) = a_{n+}(\mathbf{k}) + a_{n-}(\mathbf{k}), \quad b_{n1}(\mathbf{k}) = b_{n+}(\mathbf{k})\big[\Delta_n(\mathbf{k}) + |\mathbf{d}_n(\mathbf{k})|\big]^2 + b_{n-}(\mathbf{k})\big[\Delta_n(\mathbf{k}) - |\mathbf{d}_n(\mathbf{k})|\big]^2,$$

$$a_{n2}(\mathbf{k}) = a_{n+}(\mathbf{k}) - a_{n-}(\mathbf{k}), \quad b_{n2}(\mathbf{k}) = b_{n+}(\mathbf{k})\big[\Delta_n(\mathbf{k}) + |\mathbf{d}_n(\mathbf{k})|\big]^2 - b_{n-}(\mathbf{k})\big[\Delta_n(\mathbf{k}) - |\mathbf{d}_n(\mathbf{k})|\big]^2,$$

$$a_{n\pm}(\mathbf{k}) = \frac{\tanh\big[\beta|E_{n\pm}^0(\mathbf{k})|/2\big]}{4|E_{n\pm}^0(\mathbf{k})|}, \quad b_{n\pm}(\mathbf{k}) = \frac{\beta|E_{n\pm}^0(\mathbf{k})| - 2\tanh\big[\beta|E_{n\pm}^0(\mathbf{k})|/2\big] - \beta|E_{n\pm}^0(\mathbf{k})|\tanh^2\big[\beta|E_{n\pm}^0(\mathbf{k})|/2\big]}{32|E_{n\pm}^0(\mathbf{k})|^3},$$

$$|E_{n\pm}^0(\mathbf{k})| = \sqrt{\bigg[h_{n0}(\mathbf{k}) \pm \frac{\mathbf{h}_n(\mathbf{k})\cdot\mathbf{d}_n(\mathbf{k})}{|\mathbf{d}_n(\mathbf{k})|}\bigg]^2 + \big[\Delta_n(\mathbf{k}) \pm |\mathbf{d}_n(\mathbf{k})|\big]^2}. \tag{70}$$

Here the subscript $FS$ indicates that the integrals are computed over the Fermi surfaces $\xi_n(\mathbf{k}) = 0$ of each band $n$ ($n = \alpha, \beta, \gamma$), $T_{cs}$ and $T_{ct}$ are the native critical temperatures of singlet and triplet superconductivity in $Sr_2RuO_4$ (for simplicity we assume that they are independent on the band index $n$ but this is not essential for our qualitative results), $E_{n\pm}^0(\mathbf{k})$ are the quasiparticle energies at the Fermi surfaces, and $h_{n0}(\mathbf{k})$, $\mathbf{h}_n(\mathbf{k})$ and $\Delta_n(\mathbf{k})$, $\mathbf{d}_n(\mathbf{k})$ are the normal

---

[11] See the Appendices for more details.

[12] See footnote 11.

[13] See footnote 11.

[14] See footnote 11.

and superconducting parts of the self-energy induced by the $Sr_2IrO_4$ to the different bands discussed in Sec. 5. In order to be able to compute the coefficients $\psi_{n1}^{\mathcal{T}}$ and $\eta_{n1}^{\mathcal{T}}$, we need to fix also the coefficients $a_{1k}^{(n)}$ and $b_{1k}^{(n)}$ in the expressions of $\Delta_1^{(n)}(\mathbf{k})$ and $\mathbf{d}_1^{(n)}(\mathbf{k})$ for each band $n = \alpha, \beta, \gamma$. In principle they should be fixed so that the free energy is minimized. Thus, we expect that the momentum dependence of $\Delta_1^{(n)}(\mathbf{k})$ and $\mathbf{d}_1^{(n)}(\mathbf{k})$ in each band $n$ is determined by the competition between the type of order parameter favoured by the intrinsic interactions and the type of order parameter which has maximum overlap with the induced order parameter. Since the full interacting model is not available, we cannot use these coefficients as variational parameters but we have to fix them phenomenologically. Therefore, we will fix them so that the overlap with the induced order parameter is maximized. Since the singlet [triplet] order parameters $\Delta_1^{(n)}(\mathbf{k})$ and $\Delta_n(\mathbf{k})$ [$\mathbf{d}_1^{(n)}(\mathbf{k})$ and $\mathbf{d}_n(\mathbf{k})$] transform in the same way in the symmetry transformations $g\mathbf{k}$ ($g \in G$), we can simply choose

$$\Delta_1^{(n)}(\mathbf{k}) = \frac{\Delta_n(\mathbf{k})}{\max\{|\Delta_n(\mathbf{k})|\}}, \ \mathbf{d}_1^{(n)}(\mathbf{k}) = \frac{\mathbf{d}_n(\mathbf{k})}{\max\{|\mathbf{d}_n(\mathbf{k})|\}}, \tag{71}$$

where we have normalized the basis functions by dividing the induced order parameters $\Delta_n(\mathbf{k})$ and $\mathbf{d}_n(\mathbf{k})$ with their maximum values at the Fermi surfaces $\max\{|\Delta_n(\mathbf{k})|\}$ and $\max\{|\mathbf{d}_n(\mathbf{k})|\}$, respectively. We point out that the normalization of the basis functions can be chosen arbitrarily because it will be compensated in the values of $\psi_{n1}^{\mathcal{T}}$ and $\eta_{n1}^{\mathcal{T}}$ calculated from Eqs. (68) and (69). The advantage of our convention is that the parameters $\psi_{n1}^{\mathcal{T}}$ and $\eta_{n1}^{\mathcal{T}}$ directly describe the relevant energy scales of the intrinsic singlet and triplet order parameters. We also stress that the exact expressions for the basis functions are not important as long as they will have sufficiently large overlap with the induced order parameters.

We point out that although the absolute values of the Ginzburg-Landau coefficients are incorrect due to the fact that we have computed the integrals only over the fermi surface, we expect that their ratios will be approximately correct in the vicinity of the critical temperatures. For example in the case of the standard BCS superconductivity using a free energy calculated around the Fermi surface gives rise to a reasonably good agreement with the results obtained with the full free energy if the temperature is reasonably close to the critical temperature. Therefore, we expect that for our *qualitative* analysis (see below), where the exact quantitative values of $\psi_{n1}^{\mathcal{T}}$ and $\eta_{n1}^{\mathcal{T}}$ are unimportant, this approach should be sufficient.

## 8 Topological phase diagram for the $Sr_2RuO_4/Sr_2IrO_4$ heterostructure

We can now follow a similar approach as used above to study the phase diagram of the $Sr_2RuO_4/Sr_2IrO_4$ heterostructure. The basic idea is that once the temperature is lowered so that it approaches the critical temperature of $Sr_2RuO_4$ an intrinsic order parameter appears in the $Sr_2RuO_4$ layer breaking the protection of the semi-Dirac points. Therefore, this intrinsic order parameter is expected to cause an opening of a gap at the semi-Dirac point or a splitting transition similar to the ones studied in sections 2 and 6 depending on the nature of the intrinsic order parameter. We compute the intrinsic order parameter using the Ginzburg-Landau theory derived in Sec. 7. Since this theory is valid only at finite temperatures we have to fix the temperature so that $T > T_{cs}, T_{ct}$. In order to study the more relevant $T \rightarrow 0$ limit we would need to specify the full microscopic theory. However, we emphasize that the nature of the order parameter is not expected to change dramatically as one lowers the temperature. Therefore the intrinsic order parameter obtained at the finite temperature is representative for all temperatures and the phase diagram which we obtain using this order parameter is expected to qualitatively represent the phase diagram at $T = 0$ limit. Due to the approximations

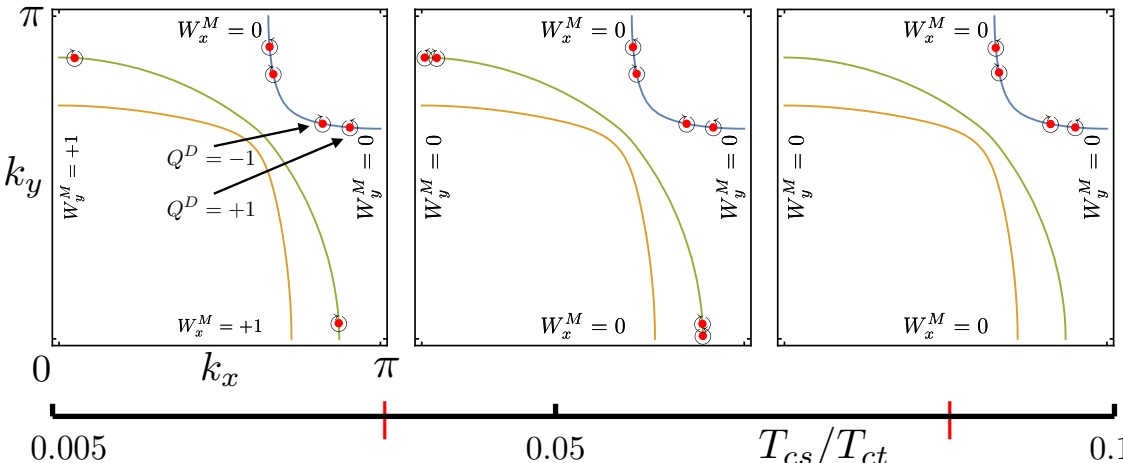

Figure 10: Phase diagram for the two layer system in the presence of intrinsic order parameters in the $t_{2g}$-ES as a function of $T_{cs}/T_{ct}$. The tight-binding parameters are described in the text and the tunneling amplitudes are chosen to be $A = B = -0.7$, so that in the absence of intrinsic order parameters the system supports semi-Dirac points in $\alpha$ and $\gamma$ bands. For $0.035 < T_{cs}/T_{ct} < 0.085$ the system supports a Dirac phase, where each semi-Dirac point has splitted into two Dirac points. For $T_{cs} \ll T_{ct}$ (the leftmost phase) the system supports topologically nontrivial mirror superconductivity with $W_x^{M,0} = 1$ and $W_y^{M,0} = 1$. For $T_{cs}/T_{ct} > 0.085$ the semi-Dirac point in the $\gamma$ band becomes gapped but Dirac points still exist in the $\alpha$ band. If $T_{cs}/T_{ct}$ would be further increased also the Dirac points shown in the $\alpha$ band would eventually become gapped. Because the Ginzburg-Landau theory is valid only for $T > T_{cs}, T_{ct}$ we have computed the intrinsic order parameters at finite temperature $T$. However, we expect that the nature of the intrinsic order parameters does not change dramatically as a function of temperature. Therefore, we expect that the phase diagram will stay qualitatively similar if the intrinsic order parameter is calculated in the limit $T \to 0$, but the locations of the phase boundaries will change quantitatively. In the calculation of the intrinsic order parameter the temperature is chosen to be $k_B T = 5 \cdot 10^{-4} t_L$ and the triplet critical temperature $T_{ct} = 0.95T$.

used in the derivation of the Ginzburg-Landau theory the predictions concerning the intrinsic order parameter are not expected to be quantitatively correct anyway.

On the level of the approximations discussed above the phase diagram of the $Sr_2RuO_4/Sr_2IrO_4$ heterostructure therefore depends only on a single parameter $T_{cs}/T_{ct}$ in addition to the parameters of the non-interacting model. We have extensively studied the phase diagrams as a function $T_{cs}/T_{ct}$ by fixing the other parameters to different values. This way we generically find that if the triplet instability is the dominating one $T_{ct} \gg T_{cs}$ we find splitting transitions of semi-Dirac points, whereas for dominating singlet instability $T_{cs} \gg T_{ct}$ the semi-Dirac points become generically gapped by lowering the temperature. Additionally, if $T_{ct} \gg T_{cs}$ it is possible to find situations where additional merging transitions lead to an appearance of topologically nontrivial mirror superconductivity similarly as discussed in sections 2 and 6. An example of a phase diagram as a function of $T_{cs}/T_{ct}$ demonstrating these different possibilities is shown in Fig. 10 for a particular choice of other parameters.

# 9 Summary and discussion

In the absence of constraints the semi-Dirac point can always be considered as a critical point of a topological phase transition where two Dirac points meet and merge in the momentum space. However, we have found that in a specific type of superconductor-metal heterostructures there naturally exists constraints which guarantee the topological stability of the semi-Dirac points. Furthermore, we have proposed that this kind of stable semi-Dirac phases can be realized if one of the layers in the heterostructure supports a large intraionic spin-orbit coupling. These systems can also support topologically distinct semi-Dirac phases with varying number of semi-Dirac points in the system. Therefore, it may be possible also to experimentally observe merging transitions of semi-Dirac points in these systems. Finally, we have shown that the protection of the semi-Dirac points can be broken in a controllable manner in a three-layer heterostructure and alternatively the protection can also become intrinsically broken if the metallic layer undergoes a transition to a superconducting state supporting an intrinsic order parameter. In the unconstrained parameter space where the protection of the semi-Dirac points is broken, these systems can support topologically distinct phases with various number of Dirac points in the different bands. The merging transition of Dirac points at the mirror lines can also lead to appearance of a topologically nontrivial mirror superconducting phase. If the superconducting layers support a fully gapped superconducting phase, it is possible also to obtain fully gapped topologically nontrivial phases in these heterostructures.

There are various experimental signatures of the topologically distinct phases and phase transitions discussed in this manuscript. The existence of semi-Dirac points and the various merging transitions shows up for example in the density of states of the system. In particular the density of states $D(E) \propto \sqrt{E}$ in the presence of semi-Dirac points whereas the Dirac points lead to $D(E) \propto E$. The density of states shows up in thermodynamic observables such as heat capacity, and it can also be studied with the help of tunneling voltage-current characteristics and ARPES. The different topological phases can also be probed with the help of surface states. In particular, the Dirac points give rise to Majorana flat bands at the edge and the topologically nontrivial mirror superconducting phases support helical Majorana edge modes.

There are also interesting directions for future research. The splitting-gapping transitions appearing in the three layer structure can be induced dynamically by applying a voltage between the superconductors. We expect that this will lead to interesting signatures in the ac Josephson effect of this system. On the other hand, the fact that the variation of the different parameters of the model only moves the semi-Dirac points in the momentum space allows for a possibility to design artificial gauge fields similarly as in the case of Weyl points [4]. In this manuscript we have studied clean systems. However, we point out that the effects of disorder may be different in the system with protected semi-Dirac points in comparison to those appearing at the critical points of the merging transitions [87,88]. From the viewpoint of phenomenology we expect that the systems supporting semi-Dirac points will also share common features also with their three dimensional analogues. A three dimensional analog of the semi-Dirac dispersion (masless relativistic particle in one direction and nonrelativistic dispersion in the other two directions) can take place in double Majorana-Weyl superconductors [89] and double Weyl semimetals [90,91]. Furthermore, they can be stable in condensed matter systems because of the symmetries of the system [91]. The existence of such kind of fermions has been speculated also in the particle physics context based on the assumption that special and general relativity are emergent properties of the quantum vacuum and the Lorentz invariance may be violated at very low energy [89]. Another 3D analog (massless relativistic particle in two directions and nonrelativistic parabolic dispersion in the third direction) is the merging point of two Weyl points with opposite chiralities [92–94].

## Acknowledgements

We thank Tero Heikkilä and Alessandra Cagnazzo for helpful discussions. The work is supported by the Academy of Finland Center of Excellence program (Project No. 284594) and the Research Council of Norway.

## A  Detailed description of the symmetries of the model

The symmetries of the system put strong constraints on the Hamiltonian. We will here consider the invariance of the system under mirroring about the $x$-axis, $\mathcal{M}_x : (x, y) \to (x, -y)$, mirroring about the $y$-axis, $\mathcal{M}_y : (x, y) \to (-x, y)$, 90° rotation about the $z$-axis, $\mathcal{R} : (x, y) \to (y, -x)$, and time reversal, $\mathcal{T}$.

In the basis $C_{i,\mathbf{k}} = (c_{yz,i,\mathbf{k},\uparrow}\, c_{yz,i,\mathbf{k},\downarrow}\, c_{xz,i,\mathbf{k},\uparrow}\, c_{xz,i,\mathbf{k},\downarrow}\, c_{xy,i,\mathbf{k},\uparrow}\, c_{xy,i,\mathbf{k},\downarrow})^T$, where $i$ is a layer index ($i = I$ for iridate and $i = M$ for $t_{2g}$ electron system), an operator $\widehat{O}$ in the Hamiltonian can be expressed in terms of a matrix $O$ as

$$\widehat{O} = \sum_{\mathbf{k}} C_{i,\mathbf{k}}^{\dagger} O_{ij}(\mathbf{k}) C_{j,\mathbf{k}}. \tag{72}$$

Invariance of the Hamiltonian under $\mathcal{M}_x$, $\mathcal{M}_y$, $\mathcal{R}_x$ and $\mathcal{T}$ means that

$$O_{ij}(k_x, k_y) = M_x O_{ij}(k_x, -k_y) M_x^{\dagger},$$
$$O_{ij}(k_x, k_y) = M_y O_{ij}(-k_x, k_y) M_y^{\dagger},$$
$$O_{ij}(k_x, k_y) = R O_{ij}(k_y, -k_x) R^{\dagger},$$
$$O_{ij}(k_x, k_y) = \Theta O_{ij}^{*}(-k_x, -k_y) \Theta^{\dagger}, \tag{73}$$

respectively, where

$$M_x = \begin{pmatrix} -i\sigma_y & 0 & 0 \\ 0 & i\sigma_y & 0 \\ 0 & 0 & -i\sigma_y \end{pmatrix}, \tag{74}$$

$$M_y = \begin{pmatrix} i\sigma_x & 0 & 0 \\ 0 & -i\sigma_x & 0 \\ 0 & 0 & -i\sigma_x \end{pmatrix}, \tag{75}$$

$$R = \begin{pmatrix} 0 & e^{-i\pi\sigma_z/4} & 0 \\ -e^{-i\pi\sigma_z/4} & 0 & 0 \\ 0 & 0 & -e^{-i\pi\sigma_z/4} \end{pmatrix}, \tag{76}$$

$$\Theta = \begin{pmatrix} i\sigma_y & 0 & 0 \\ 0 & i\sigma_y & 0 \\ 0 & 0 & i\sigma_y \end{pmatrix}. \tag{77}$$

A tunneling Hamiltonian between all states in the iridate layer and the $t_{2g}$ metal can be expressed as

$$\widehat{H}_{IM} = \sum_{\mathbf{k}} \tilde{C}_{I,\mathbf{k}}^{\dagger} T_{IM}(\mathbf{k}) C_{M,\mathbf{k}} + h.c., \tag{78}$$

where $\tilde{C}_{I,\mathbf{k}} \equiv (f_{\mathbf{k},\Uparrow}\, f_{\mathbf{k},\Downarrow}\, g_{\mathbf{k},\Uparrow}\, g_{\mathbf{k},\Downarrow}\, h_{\mathbf{k},\Uparrow}\, h_{\mathbf{k},\Downarrow})^T = U_{SO} C_{I,\mathbf{k}}$. Here

$$U_{SO} = \begin{pmatrix} \frac{i}{\sqrt{3}}\sigma_y & -\frac{i}{\sqrt{3}}\sigma_x & \frac{1}{\sqrt{3}}\sigma_0 \\ \frac{i}{\sqrt{2}}\sigma_z & \frac{1}{\sqrt{2}}\sigma_0 & 0 \\ \frac{i}{\sqrt{6}}\sigma_y & -\frac{i}{\sqrt{6}}\sigma_x & -\sqrt{\frac{2}{3}}\sigma_0 \end{pmatrix} \tag{79}$$

diagonalizes the atomic spin-orbit coupling term in the Hamiltonian. Because the atomic spin-orbit coupling in the iridate layer dominates all other terms, $f$, $g$ and $h$ orbitals describe the eigenstates in the iridate layer. Moreover, $g$ and $h$ orbitals are fully occupied and the low-energy theory for the iridate layer can be constructed using only the $f$ orbitals. We can now apply the constraints (73) to $U_{SO}^{\dagger} T_{IM}(\mathbf{k})$, which is expressed in the $t_{2g}$ basis. We find

$$
\begin{aligned}
T_{IM}(k_x, k_y) &= U_{SO} M_x U_{SO}^{\dagger} T_{IM}(k_x, -k_y) M_x^{\dagger}, \\
T_{IM}(k_x, k_y) &= U_{SO} M_y U_{SO}^{\dagger} T_{IM}(-k_x, k_y) M_y^{\dagger}, \\
T_{IM}(k_x, k_y) &= U_{SO} R U_{SO}^{\dagger} T_{IM}(k_y, -k_x) R^{\dagger}, \\
T_{IM}(k_x, k_y) &= U_{SO} \Theta U_{SO}^{T} T_{IM}^{*}(-k_x, -k_y) \Theta^{\dagger}.
\end{aligned}
\tag{80}
$$

The tunneling matrices $T_{f,yz}$, $T_{f,xz}$, $T_{f,xy}$, corresponds to the first row of $2 \times 2$ matrices of $T_{IM}$. If we express the tunneling matrices as

$$
T_{f,n}(\mathbf{k}) = T_{f,n,0}(\mathbf{k}) \sigma_0 + \mathbf{T}_{f,n}(\mathbf{k}) \cdot \boldsymbol{\sigma} \quad (n = yz, xz, xy),
\tag{81}
$$

it follows from the constraints (80) that

$$
\begin{array}{ll}
T_{f,yz,0}(\mathbf{k}) = i J_{AS}(k_x, k_y) & T_{f,yz,x}(\mathbf{k}) = i J_{AA}(k_x, k_y) \\
T_{f,yz,y}(\mathbf{k}) = i J_{SS}(k_x, k_y) & T_{f,yz,z}(\mathbf{k}) = J_{SA}(k_x, k_y) \\
T_{f,xz,0}(\mathbf{k}) = i J_{AS}(k_y, k_x) & T_{f,xz,x}(\mathbf{k}) = -i J_{SS}(k_y, k_x) \\
T_{f,xz,y}(\mathbf{k}) = -i J_{AA}(k_y, k_x) & T_{f,xz,z}(\mathbf{k}) = -J_{SA}(k_y, k_x) \\
T_{f,xy,0}(\mathbf{k}) = K_{SS}(k_x, k_y) & T_{f,xy,x}(\mathbf{k}) = K_{SA}(k_x, k_y) \\
T_{f,xy,y}(\mathbf{k}) = -K_{SA}(k_y, k_x) & T_{f,xy,z}(\mathbf{k}) = i K_{AA}(k_x, k_y).
\end{array}
\tag{82}
$$

Here the functions $K$ and $J$ are real and the subscript $S$ ($A$) designates that the function is symmetric (anti-symmetric) with respect to the corresponding momentum component. For example $K_{SA}(k_x, k_y)$ means that $K_{SA}(-k_x, k_y) = K_{SA}(k_x, k_y)$ and $K_{SA}(k_x, -k_y) = -K_{SA}(k_x, k_y)$.

We also need the tunneling matrices $T_n(\mathbf{k})$ from the iridate $f$-orbitals to the $n =: \alpha, \beta, \gamma$ bands in the $t_{2g}$-ES. Therefore, we rewrite the tunneling Hamiltonian as

$$
\widehat{H}_{IM} = \sum_{\mathbf{k}} \tilde{C}_{I,\mathbf{k}}^{\dagger} \bar{T}_{IM}(\mathbf{k}) \bar{C}_{M,\mathbf{k}} + h.c.,
\tag{83}
$$

where $\bar{C}_{M,\mathbf{k}} \equiv (c_{\alpha,\mathbf{k},\uparrow} c_{\alpha,\mathbf{k},\downarrow} c_{\beta,\mathbf{k},\uparrow} c_{\beta,\mathbf{k},\downarrow} c_{\gamma,\mathbf{k},\uparrow} c_{\gamma,\mathbf{k},\downarrow})^T = U_0(\mathbf{k}) C_{M,\mathbf{k}}$, $\bar{T}_{IM}(\mathbf{k}) = T_{IM}(\mathbf{k}) U_0^{\dagger}(\mathbf{k})$, and $U_0(\mathbf{k})$ diagonalises the Hamiltonian of the $t_{2g}$-ES,

$$
U_0(\mathbf{k}) h_0(\mathbf{k}) U_0^{\dagger}(\mathbf{k}) = \begin{pmatrix} \xi_\alpha(\mathbf{k}) \sigma_0 & 0 & 0 \\ 0 & \xi_\beta(\mathbf{k}) \sigma_0 & 0 \\ 0 & 0 & \xi_\gamma(\mathbf{k}) \sigma_0 \end{pmatrix}.
\tag{84}
$$

$U_0(\mathbf{k})$ is in general not unique. However, the derivations can be simplified considerably if we fix a particular convention for it. Therefore we fix the structure of $U_0(\mathbf{k})$ to be

$$
U_0^{\dagger}(\mathbf{k}) = \begin{pmatrix} 0 & u_{\alpha,yz}(\mathbf{k}) & 0 & u_{\beta,yz}(\mathbf{k}) & 0 & u_{\gamma,yz}(\mathbf{k}) \\ -u_{\alpha,yz}^{*}(\mathbf{k}) & 0 & -u_{\beta,yz}^{*}(\mathbf{k}) & 0 & -u_{\gamma,yz}^{*}(\mathbf{k}) & 0 \\ 0 & u_{\alpha,xz}(\mathbf{k}) & 0 & u_{\beta,xz}(\mathbf{k}) & 0 & u_{\gamma,xz}(\mathbf{k}) \\ -u_{\alpha,xz}^{*}(\mathbf{k}) & 0 & -u_{\beta,xz}^{*}(\mathbf{k}) & 0 & -u_{\gamma,xz}^{*}(\mathbf{k}) & 0 \\ u_{\alpha,xy}(\mathbf{k}) & 0 & u_{\beta,xy}(\mathbf{k}) & 0 & u_{\gamma,xy}(\mathbf{k}) & 0 \\ 0 & u_{\alpha,xy}(\mathbf{k}) & 0 & u_{\beta,xy}(\mathbf{k}) & 0 & u_{\gamma,xy}(\mathbf{k}) \end{pmatrix}.
\tag{85}
$$

Here

$$(0 \quad -u_{n,yz}^*(\mathbf{k}) \quad 0 \quad -u_{n,xz}^*(\mathbf{k}) \quad u_{n,xy}(\mathbf{k}) \quad 0)^T$$

and

$$(u_{n,yz}(\mathbf{k}) \quad 0 \quad u_{n,xz}(\mathbf{k}) \quad 0 \quad 0 \quad u_{n,xy}(\mathbf{k}))^T$$

for $n = \alpha, \beta, \gamma$ are normalized eigenvectors of $h_0(\mathbf{k})$ with eigenvalue $\xi_n(\mathbf{k})$. Notice that we have used here the fact that in an individual layer there exists both time-reversal and inversion symmetries which guarantee that the eigenvalues are doubly degenerate. Moreover, we fix $u_{n,xy}(\mathbf{k})$ to be real and positive. With these conventions $U_0(\mathbf{k})$ is unique. (This procedure is not well-defined if $u_{n,xy}(\mathbf{k}) = 0$, but in these cases $U_0(\mathbf{k})$ can be constructed using analytic continuation.) By using that $h_0(\mathbf{k})$ satisfies (73) we find that

$$
\begin{aligned}
U_0^\dagger(k_x, -k_y) &= M_x^\dagger U_0^\dagger(k_x, k_y)\tilde{M}_x, \\
U_0^\dagger(-k_x, k_y) &= M_y^\dagger U_0^\dagger(k_x, k_y)\tilde{M}_y, \\
U_0^\dagger(k_y, -k_x) &= R^\dagger U_0^\dagger(k_x, k_y)\tilde{R}, \\
U_0^\dagger(-k_x, -k_y) &= \Theta^T U_0^T(k_x, k_y)\tilde{\Theta}^*.
\end{aligned}
\tag{86}
$$

Here we have introduced matrices $\tilde{M}_x$, $\tilde{M}_y$, $\tilde{R}$ and $\tilde{\Theta}$ in order to guarantee that the structure of $U_0(\mathbf{k})$ [Eq. (85)] stays invariant in the transformations. We find that these matrices can be arbitrary block-diagonal matrices and $U_0(\mathbf{k})$ still diagonalizes the Hamiltonian $h_0(\mathbf{k})$, but the structure described in Eq. (85) is obeyed with a specific choice

$$
\tilde{M}_x^\dagger = i \begin{pmatrix} \sigma_y & 0 & 0 \\ 0 & \sigma_y & 0 \\ 0 & 0 & \sigma_y \end{pmatrix}, \quad
\tilde{M}_y^\dagger = i \begin{pmatrix} \sigma_x & 0 & 0 \\ 0 & \sigma_x & 0 \\ 0 & 0 & \sigma_x \end{pmatrix},
$$

$$
\tilde{R}^\dagger = \frac{-1}{\sqrt{2}} \begin{pmatrix} \sigma_0 + i\sigma_z & 0 & 0 \\ 0 & \sigma_0 + i\sigma_z & 0 \\ 0 & 0 & \sigma_0 + i\sigma_z \end{pmatrix}, \quad
\tilde{\Theta}_x^\dagger = -i \begin{pmatrix} \sigma_y & 0 & 0 \\ 0 & \sigma_y & 0 \\ 0 & 0 & \sigma_y \end{pmatrix}.
\tag{87}
$$

By using Eq. (86) the invariance of $\bar{T}_{IM}$ under $\mathcal{M}_x$, $\mathcal{M}_y$, $\mathcal{R}_x$ and $\mathcal{T}$ leads to

$$
\begin{aligned}
\bar{T}_{IM}(k_x, k_y) &= U_{SO} M_x U_{SO}^\dagger \bar{T}_{IM}(k_x, -k_y)\tilde{M}_x^\dagger, \\
\bar{T}_{IM}(k_x, k_y) &= U_{SO} M_y U_{SO}^\dagger \bar{T}_{IM}(-k_x, k_y)\tilde{M}_y^\dagger, \\
\bar{T}_{IM}(k_x, k_y) &= U_{SO} R U_{SO}^\dagger \bar{T}_{IM}(k_y, -k_x)\tilde{R}^\dagger, \\
\bar{T}_{IM}(k_x, k_y) &= U_{SO} \Theta U_{SO}^T \bar{T}_{IM}^*(-k_x, -k_y)\tilde{\Theta}^\dagger.
\end{aligned}
\tag{88}
$$

By considering only the tunneling between the $f$-band and the $\alpha$, $\beta$ and $\gamma$-bands in the $t_{2g}$-ES, Eq. (88) reduces to

$$T_n(k_x, k_y) = \sigma_y T_n(k_x, -k_y)\sigma_y, \tag{89}$$

$$T_n(k_x, k_y) = \sigma_x T_n(-k_x, k_y)\sigma_x, \tag{90}$$

$$T_n(k_x, k_y) = \frac{\sigma_0 - i\sigma_z}{\sqrt{2}} T_n(k_y, -k_x)\frac{\sigma_0 + i\sigma_z}{\sqrt{2}}, \tag{91}$$

$$T_n(k_x, k_y) = \sigma_y T_n^*(-k_x, -k_y)\sigma_y, \tag{92}$$

where $n = \alpha, \beta, \gamma$.

From Eqs. (89) and (90) it follows that $T_n$ is invariant under 2D inversion if

$$T_n(k_x, k_y) = \sigma_z T_n(-k_x, -k_y)\sigma_z. \tag{93}$$

Similarly it follows from Eqs. (89) and (91) that $T_n$ is invariant under mirroring about the diagonal if

$$T_n(k_x, k_y) = M_d T_n(k_y, k_x) M_d, \tag{94}$$

where $M_d = \frac{1}{\sqrt{2}}(\sigma_x - \sigma_y)$.

The symmetries (92) and (93) are used in the main text to guarantee the stability of the semi-Dirac points. The mirror symmetries (89) and (90) are needed to prove the existence of topological mirror invariants. All these symmetries are also used to simplify the expressions in the Ginzburg-Landau theory derived in Sec. C of this supplementary material.

## B  Microscopic analysis of the tunneling matrices

An explicit form for the tunneling matrices can be found by considering the dominating tunneling paths between two $t_{2g}$ layers on top of each other. The orientation and shape of the $t_{2g}$ orbitals are shown in Fig. 11(a). We assume lattice matched square lattices for the layers. The hoppings between a $yz$ orbital in the bottom layer to the $t_{2g}$ orbitals in the layer above are shown in the left column of Fig. 11(b). The system is seen from above. Due to the orientation of the lobes we see that the tunneling matrices from the $yz$ orbital located at the two dimensional position $\mathbf{r}$ in the bottom layer to the $yz$ orbitals in the neighbouring sites in the top layer are given by

$$\langle yz, t, \mathbf{r} | H | yz, b, \mathbf{r} \rangle = -t_{yz,yz}^{tb,0} \,, \tag{95}$$

$$\langle yz, t, \mathbf{r} \pm d\mathbf{e}_x | H | yz, b, \mathbf{r} \rangle = -t_{yz,yz}^{tb,x} \,, \tag{96}$$

$$\langle yz, t, \mathbf{r} \pm d\mathbf{e}_y | H | yz, b, \mathbf{r} \rangle = t_{yz,yz}^{tb,y} \,, \tag{97}$$

where $t$ and $b$ designates the top and bottom layer, respectively. We assume that the spin is conserved in these hopping processes. Similar considerations show that there will be no coupling between an $yz$ orbital in the bottom layer to the nearest neighbour $xy$ orbital in the top layer, as well as to the next nearest neighbour $xy$ orbitals along the $y$-direction in the top layer. However, there will be a coupling to the next nearest $xy$ orbitals above along the $x$-direction, but these matrix elements come with opposite sign

$$\langle xy, t, \mathbf{r} | H | yz, b, \mathbf{r} \rangle = 0 \,, \tag{98}$$

$$\langle xy, t, \mathbf{r} \pm d\mathbf{e}_x | H | yz, b, \mathbf{r} \rangle = \mp t_{xy,yz}^{tb,x} \,, \tag{99}$$

$$\langle xy, t, \mathbf{r} \pm d\mathbf{e}_y | H | yz, b, \mathbf{r} \rangle = 0. \tag{100}$$

In a similar way we find that there are no coupling between a $yz$ orbital in the bottom layer and $xz$ orbitals in the top layer up to next nearest neighbours.

To reduce the number of free parameters we will only keep the matrix elements of *dominating* order for each pair of orbitals in the two layers. By similarly considering the $xz$ and $xy$ orbitals in the bottom layer and the $t_{2g}$ orbitals in the top layer the tunneling part of the Hamiltonian $\hat{H}_{tb}$ takes the form

$$\begin{aligned}
\hat{H}_{tb} = \sum_{\mathbf{r}} \Big[ &-t_{yz,yz}^{tb,0} c_{yz,t,\mathbf{r}}^{\dagger} c_{yz,b,\mathbf{r}} - t_{xy,yz}^{tb,x} \left( c_{xy,t,\mathbf{r}+d\mathbf{e}_x}^{\dagger} - c_{xy,t,\mathbf{r}-d\mathbf{e}_x}^{\dagger} \right) c_{yz,b,\mathbf{r}} \\
&-t_{xz,xz}^{tb,0} c_{xz,t,\mathbf{r}}^{\dagger} c_{xz,b,\mathbf{r}} - t_{xy,xz}^{tb,y} \left( c_{xy,t,\mathbf{r}+d\mathbf{e}_y}^{\dagger} - c_{xy,t,\mathbf{r}-d\mathbf{e}_y}^{\dagger} \right) c_{xz,b,\mathbf{r}} \\
&-t_{yz,xy}^{tb,x} \left( c_{yz,t,\mathbf{r}+d\mathbf{e}_x}^{\dagger} - c_{yz,t,\mathbf{r}-d\mathbf{e}_x}^{\dagger} \right) c_{xy,b,\mathbf{r}} \\
&-t_{xz,xy}^{tb,y} \left( c_{xz,t,\mathbf{r}+d\mathbf{e}_y}^{\dagger} - c_{xz,t,\mathbf{r}-d\mathbf{e}_y}^{\dagger} \right) c_{xy,b,\mathbf{r}} + t_{xy,xy}^{tb,0} c_{xy,t,\mathbf{r}}^{\dagger} c_{xy,b,\mathbf{r}} \Big] + h.c. \tag{101}
\end{aligned}$$

$(a)$

$(b)$

Figure 11: (a) The different $t_{2g}$ orbitals and their orientation. (b) The possible tunneling paths between an orbital (thin lines) in the bottom layer to the nearest and and next nearest orbitals (thick lines) in the layer above. Also the corresponding matrix elements are shown. The orbitals are seen from above.

Due to the assumption of a square lattice it is from Fig. 11 apparent that $t_{xy,yz}^{tb,x} = t_{xy,xz}^{tb,y}$, $t_{yz,xy}^{tb,x} = t_{xz,xy}^{tb,y}$ and $t_{yz,yz}^{tb,0} = t_{xz,xz}^{tb,0}$. For simplicity we assume also that $t_{yz,xy}^{tb,x} = t_{xy,yz}^{tb,x}$, but this is not important for the qualitative results. After a Fourier transform the tunneling part of the

Hamiltonian takes the form

$$\hat{H}_{tb} = \sum_{\mathbf{k}} t_{\perp} \Big[ -c^{\dagger}_{yz,t,\mathbf{k}} c_{yz,b,\mathbf{k}} - i2B \sin k_x c^{\dagger}_{xy,t,\mathbf{k}} c_{yz,b,\mathbf{k}} - c^{\dagger}_{xz,t,\mathbf{k}} c_{xz,b,\mathbf{k}} - i2B \sin k_y c^{\dagger}_{xy,t,\mathbf{k}} c_{xz,b,\mathbf{k}}$$

$$+ A c^{\dagger}_{xy,t,\mathbf{k}} c_{xy,b,\mathbf{k}} - i2B \sin k_x c^{\dagger}_{yz,t,\mathbf{k}} c_{xy,b,\mathbf{k}} - i2B \sin k_y c^{\dagger}_{xz,t,\mathbf{k}} c_{xy,b,\mathbf{k}} \Big] + h.c., \quad (102)$$

where we have defined $t_{\perp} = t^{tb,0}_{yz,yz}$ and $A = t^{tb,0}_{xy,xy}/t^{tb,0}_{yz,yz}$ and $B = t^{tb,x}_{yz,xy}/t^{tb,0}_{yz,yz}$. Based on this kind of simple analysis we can not say anything definite about the magnitude and the sign of the parameters $t_{\perp}$, $A$ and $B$. However, since $t_{\perp} = t^{tb,0}_{yz,yz}$ corresponds to hopping between orbitals directly on top of each other with out of plane lobes, we expect this energy scale to be dominating so that $|A|, |B| < 1$. Moreover, based on the nature of the tunneling paths shown in Fig. 11 we expect $A$ and $B$ to be roughly of similar magnitude.

The tunneling matrices between the $f$ orbitals in the bottom iridate layer and the $t_{2g}$ orbitals in the top layer can be found by projecting the $t_{2g}$ operators for the bottom layer onto the $f$-band

$$c_{yz,b,\mathbf{k},\alpha} \to -\frac{i}{\sqrt{3}}\sigma_{y,\alpha\beta} f_{\mathbf{k},\beta}, \qquad c_{xz,b,\mathbf{k},\alpha} \to \frac{i}{\sqrt{3}}\sigma_{x,\alpha\beta} f_{\mathbf{k},\beta}, \qquad c_{xy,b,\mathbf{k},\alpha} \to \frac{1}{\sqrt{3}}\sigma_{0,\alpha\beta} f_{\mathbf{k},\beta}, \quad (103)$$

where $\alpha$ and $\beta$ are the spin and the pseudospin degree of freedom, respectively. The tunneling matrices are then easily found to be

$$T_{f,yz}(\mathbf{k}) = \frac{it_{\perp}}{\sqrt{3}} \Big[ 2B \sin k_x \sigma_0 - \sigma_y \Big], \quad (104)$$

$$T_{f,xz}(\mathbf{k}) = \frac{it_{\perp}}{\sqrt{3}} \Big[ 2B \sin k_y \sigma_0 + \sigma_x \Big], \quad (105)$$

$$T_{f,xy}(\mathbf{k}) = \frac{t_{\perp}}{\sqrt{3}} \Big[ A\sigma_0 + 2B(\sin k_y \sigma_x - \sin k_x \sigma_y) \Big]. \quad (106)$$

These tunneling matrices are used in the main text.

If the iridate layer is on the top of the $t_{2g}$ layer we use Eq. (102) and project the operators for the top layer onto the $f$-band. This yields

$$\tilde{T}_{f,yz}(\mathbf{k}) = \frac{i\tilde{t}_{\perp}}{\sqrt{3}} \Big[ -2\tilde{B} \sin k_x \sigma_0 - \sigma_y \Big], \quad (107)$$

$$\tilde{T}_{f,xz}(\mathbf{k}) = \frac{i\tilde{t}_{\perp}}{\sqrt{3}} \Big[ -2\tilde{B} \sin k_y \sigma_0 + \sigma_x \Big], \quad (108)$$

$$\tilde{T}_{f,xy}(\mathbf{k}) = \frac{\tilde{t}_{\perp}}{\sqrt{3}} \Big[ \tilde{A}\sigma_0 - 2\tilde{B}(\sin k_y \sigma_x - \sin k_x \sigma_y) \Big]. \quad (109)$$

These tunneling matrices are used in the main text in the case of the three-layer heterostructure, and we have denoted the parameters as $\tilde{t}_{\perp}$, $\tilde{A}$ and $\tilde{B}$ to allow the possibility of different magnitudes of the tunneling amplitudes to the top and bottom layers.

## C Derivation of Ginzburg Landau theory for $Sr_2RuO_4$ in the presence of induced order parameter

The intrinsic order parameters can be obtained by minimizing the free energy of the system with respect to them. The total free energy consist of the term $F_I$ arising from the decoupling interaction term (see below) and the free energy for the Bogoliubov quasiparticles

$$F_{\text{qp}} = \sum_{\mathbf{k}} F_{\text{qp}}(\mathbf{k}) = -\beta^{-1} \sum_{\mathbf{k}} \sum_m \ln[2\cosh(\beta E_m(\mathbf{k})/2)]. \quad (110)$$

Here $\beta$ is the inverse temperature, $E_m(\mathbf{k})$ are the positive quasiparticle energies, and the momentum summation should be calculated over the Brillouin zone. For the sake of analytical transparency, we will make several simplifying assumptions. First, we assume that the order parameters for the different bands $\alpha$, $\beta$ and $\gamma$ can be computed independently of each other i.e. we neglect the effects of the interband order parameters. Secondly, instead of computing the Free energy of the full Brillouin zone we will calculate it only over the Fermi surfaces for each of the bands i.e. over the lines $\xi_n(\mathbf{k}) = 0$ ($n = \alpha, \beta, \gamma$). This approximation is reasonable since we expect that the behavior of the free energy in the vicinity of the Fermi surface will be qualitatively similar as elsewhere in the Brillouin zone and the intrinsic order parameter will have the largest effect on $F_{qp}(\mathbf{k})$ in the vicinity of the Fermi surface. Finally, we assume that the overal magnitudes of the intrinsic order parameters in all bands satisfy $\beta|\mathbf{d}_{nR}| \ll 1$, $\beta|\Delta_{nR}| \ll 1$ ($n = \alpha, \beta, \gamma$), and the temperature is above the critical temperature of the superconducting instability in $Sr_2RuO_4$. We expect that these assumptions do not affect the results qualitatively, so that all the qualitative results presented in the manuscript will remain similar also in the parameter regimes where these assumptions are not satisfied.

In the vicinity of the Fermi surfaces $\xi_n(\mathbf{k}) = 0$ the BdG Hamiltonians can be written in the form

$$H_n(\mathbf{k}) = \begin{pmatrix} h_n(\mathbf{k}) & \tilde{\Delta}_n(\mathbf{k}) \\ \tilde{\Delta}_n^\dagger(\mathbf{k}) & -h_n^T(-\mathbf{k}) \end{pmatrix}, \tag{111}$$

where $h_n(\mathbf{k}) = h_{n0}(\mathbf{k})\sigma_0 + \mathbf{h}_n(\mathbf{k}) \cdot \vec{\sigma}$ is the induced normal part arising due to the coupling to the iridate, $\tilde{\Delta}_n(\mathbf{k}) = i\{[\Delta_n(\mathbf{k}) + \Delta_{nR}(\mathbf{k})]\sigma_0 + [\mathbf{d}_n(\mathbf{k}) + \mathbf{d}_{nR}(\mathbf{k})] \cdot \vec{\sigma}\}\sigma_y$, and $\Delta_n(\mathbf{k})$ $[\Delta_{nR}(\mathbf{k})]$ and $\mathbf{d}_n(\mathbf{k})$ $[\mathbf{d}_{nR}(\mathbf{k})]$ are the induced [intrinsic] singlet and triplet order parameters, respectively. Assuming that the self-energy arising due to the coupling to the iridate can be evaluated at zero energy (which should be a reasonably good approximation close to the Fermi surface), the induced terms in the Hamiltonian satisfy

$$h_{n0}(\mathbf{k})\sigma_0 + \mathbf{h}_n(\mathbf{k}) \cdot \vec{\sigma} = -\frac{\xi_I(\mathbf{k})}{\xi_I^2(\mathbf{k}) + \Delta_I^2(\mathbf{k})} T_n(\mathbf{k}) T_n^\dagger(\mathbf{k}),$$

$$\Delta_n(\mathbf{k})\sigma_0 + \mathbf{d}_n(\mathbf{k}) \cdot \vec{\sigma} = \frac{\Delta_I(\mathbf{k})}{\xi_I^2(\mathbf{k}) + \Delta_I^2(\mathbf{k})} T_n(\mathbf{k}) T_n^\dagger(\mathbf{k}), \tag{112}$$

where $T_n(\mathbf{k})$ is the tunneling matrix from iridate to the band $n = \alpha, \beta, \gamma$ in the ruthenate.

The free energy for the Bogoliubov quasiparticles over the Fermi surfaces ($n = \alpha, \beta, \gamma$) can be expressed as

$$F_{qp,n} = \int_{FS} dk \, F_{qp,n}(\mathbf{k}) = -\beta^{-1} \int_{FS} dk \sum_{\sigma=\pm} \ln[2\cosh(\beta E_{n\sigma}(\mathbf{k})/2)], \tag{113}$$

where

$$\int_{FS} dk F(\mathbf{k}) = \int_{\{\mathbf{k}|\xi_n(\mathbf{k})=0\}} dk F(\mathbf{k}),$$

$$E_{n\pm}(\mathbf{k}) = \sqrt{|E_{n\pm}^0(\mathbf{k})|^2 + \epsilon_{n\pm}(\mathbf{k})},$$

$$|E_{n\pm}^0(\mathbf{k})| = \sqrt{\left[h_{n0}(\mathbf{k}) \pm \frac{\mathbf{h}_n(\mathbf{k})\cdot\mathbf{d}_n(\mathbf{k})}{|\mathbf{d}_n(\mathbf{k})|}\right]^2 + \left[\Delta_n(\mathbf{k}) \pm |\mathbf{d}_n(\mathbf{k})|\right]^2},$$

$$\epsilon_{n\pm}(\mathbf{k}) = 2\left[\Delta_n(\mathbf{k}) \pm |\mathbf{d}_n(\mathbf{k})|\right]\Delta_{nR}^{\mathscr{T}}(\mathbf{k}) + 2\left[|\mathbf{d}_n(\mathbf{k})| \pm \Delta_n(\mathbf{k})\right]\frac{\mathbf{d}_{nR}^{\mathscr{T}}(\mathbf{k})\cdot\mathbf{d}_n(\mathbf{k})}{|\mathbf{d}_n(\mathbf{k})|}$$

$$+ |\Delta_{nR}^{\mathscr{T}}(\mathbf{k})|^2 \pm 2\Delta_{nR}^{\mathscr{T}}(\mathbf{k})\frac{\mathbf{d}_{nR}^{\mathscr{T}}(\mathbf{k})\cdot\mathbf{d}_n(\mathbf{k})}{|\mathbf{d}_n(\mathbf{k})|}$$

$$+ \left[1 \pm \frac{\Delta_n^2(\mathbf{k}) + |\mathbf{h}_n(\mathbf{k})|^2}{h_{n0}(\mathbf{k})\frac{\mathbf{h}_n(\mathbf{k})\cdot\mathbf{d}_n(\mathbf{k})}{|\mathbf{d}_n(\mathbf{k})|} + \Delta_n(\mathbf{k})|\mathbf{d}_n(\mathbf{k})|}\right]|\mathbf{d}_{nR}^{\mathscr{T}}(\mathbf{k})|^2$$

$$\mp \frac{\Delta_n^2(\mathbf{k}) + |\mathbf{h}_n(\mathbf{k})|^2}{h_{n0}(\mathbf{k})\frac{\mathbf{h}_n(\mathbf{k})\cdot\mathbf{d}_n(\mathbf{k})}{|\mathbf{d}_n(\mathbf{k})|} + \Delta_n(\mathbf{k})|\mathbf{d}_n(\mathbf{k})|}\frac{[\mathbf{d}_{nR}^{\mathscr{T}}(\mathbf{k})\cdot\mathbf{d}_n(\mathbf{k})]^2}{|\mathbf{d}_n(\mathbf{k})|^2}$$

$$+ |\Delta_{nR}^{\mathscr{N}}(\mathbf{k})|^2 \pm 2\Delta_{nR}^{\mathscr{N}}(\mathbf{k})\frac{\mathbf{d}_{nR}^{\mathscr{N}}(\mathbf{k})\cdot\mathbf{d}_n(\mathbf{k})}{|\mathbf{d}_n(\mathbf{k})|}$$

$$+ \left[1 \pm \frac{|\mathbf{d}_n(\mathbf{k})|}{\Delta_n(\mathbf{k})}\right]|\mathbf{d}_{nR}^{\mathscr{N}}(\mathbf{k})|^2 \mp \frac{|\mathbf{d}_n(\mathbf{k})|}{\Delta_n(\mathbf{k})}\frac{[\mathbf{d}_{nR}^{\mathscr{N}}(\mathbf{k})\cdot\mathbf{d}_n(\mathbf{k})]^2}{|\mathbf{d}_n(\mathbf{k})|^2}$$

$$\pm 2\frac{\Delta_n(\mathbf{k}) \pm |\mathbf{d}_n(\mathbf{k})|}{\Delta_n(\mathbf{k})}\frac{h_{n0}(\mathbf{k})}{\sqrt{h_{n0}^2(\mathbf{k}) + \Delta_n^2(\mathbf{k})}}\mathbf{d}_{nR}^{\mathscr{T}}(\mathbf{k})\cdot\frac{\mathbf{d}_{nR}^{\mathscr{N}}(\mathbf{k}) \times \mathbf{d}_n(\mathbf{k})}{|\mathbf{d}_n(\mathbf{k})|}. \qquad (114)$$

Here we have separated the intrinsic order parameters $\Delta_{nR}(\mathbf{k}) = \Delta_{nR}^{\mathscr{T}}(\mathbf{k}) + i\Delta_{nR}^{\mathscr{N}}(\mathbf{k})$ and $\mathbf{d}_{nR}(\mathbf{k}) = \mathbf{d}_{nR}^{\mathscr{T}}(\mathbf{k}) + i\mathbf{d}_{nR}^{\mathscr{N}}(\mathbf{k})$ to the contributions obeying $[\Delta_{nR}^{\mathscr{T}}(\mathbf{k}) \in \mathbb{R}, \mathbf{d}_{nR}^{\mathscr{T}}(\mathbf{k}) \in \mathbb{R}^3]$ and breaking $[\Delta_{nR}^{\mathscr{N}}(\mathbf{k}) \in \mathbb{R}, \mathbf{d}_{nR}^{\mathscr{N}}(\mathbf{k}) \in \mathbb{R}^3]$ the time-reversal symmetry. Using these expressions we obtain

$$F_{\text{qp},n} \approx F_{0,n} - \int_{FS} dk \sum_{\sigma=\pm} \left[ a_{n\sigma}(\mathbf{k})\epsilon_{n\sigma}(\mathbf{k}) + b_{n\sigma}(\mathbf{k})\epsilon_{n\sigma}^2(\mathbf{k}) \right]$$

$$\approx F_{0,n} - \int_{FS} dk \left\{ 2\left[ a_{n1}(\mathbf{k})\Delta_n(\mathbf{k}) + a_{n2}(\mathbf{k})|\mathbf{d}_n(\mathbf{k})| \right]\Delta_{nR}^{\mathcal{T}}(\mathbf{k}) \right.$$

$$+ 2\left[ a_{n1}(\mathbf{k})|\mathbf{d}_n(\mathbf{k})| + a_{n2}(\mathbf{k})\Delta_n(\mathbf{k}) \right]\frac{\mathbf{d}_{nR}^{\mathcal{T}}(\mathbf{k}) \cdot \mathbf{d}_n(\mathbf{k})}{|\mathbf{d}_n(\mathbf{k})|}$$

$$+ a_{n1}(\mathbf{k})|\Delta_{nR}^{\mathcal{T}}(\mathbf{k})|^2 + 2a_{n2}(\mathbf{k})\Delta_{nR}^{\mathcal{T}}(\mathbf{k})\frac{\mathbf{d}_{nR}^{\mathcal{T}}(\mathbf{k}) \cdot \mathbf{d}_n(\mathbf{k})}{|\mathbf{d}_n(\mathbf{k})|}$$

$$+ a_{n1}(\mathbf{k})|\Delta_{nR}^{\mathcal{N}}(\mathbf{k})|^2 + 2a_{n2}(\mathbf{k})\Delta_{nR}^{\mathcal{N}}(\mathbf{k})\frac{\mathbf{d}_{nR}^{\mathcal{N}}(\mathbf{k}) \cdot \mathbf{d}_n(\mathbf{k})}{|\mathbf{d}_n(\mathbf{k})|}$$

$$+ \left[ a_{n1}(\mathbf{k}) + a_{n2}(\mathbf{k})\frac{\Delta_n^2(\mathbf{k}) + |\mathbf{h}_n(\mathbf{k})|^2}{h_{n0}(\mathbf{k})\frac{\mathbf{h}_n(\mathbf{k})\cdot\mathbf{d}_n(\mathbf{k})}{|\mathbf{d}_n(\mathbf{k})|} + \Delta_n(\mathbf{k})|\mathbf{d}_n(\mathbf{k})|} \right]|\mathbf{d}_{nR}^{\mathcal{T}}(\mathbf{k})|^2$$

$$+ \left[ a_{n1}(\mathbf{k}) + a_{n2}(\mathbf{k})\frac{|\mathbf{d}_n(\mathbf{k})|}{\Delta_n(\mathbf{k})} \right]|\mathbf{d}_{nR}^{\mathcal{N}}(\mathbf{k})|^2$$

$$- a_{n2}(\mathbf{k})\frac{\Delta_n^2(\mathbf{k}) + |\mathbf{h}_n(\mathbf{k})|^2}{h_{n0}(\mathbf{k})\frac{\mathbf{h}_n(\mathbf{k})\cdot\mathbf{d}_n(\mathbf{k})}{|\mathbf{d}_n(\mathbf{k})|} + \Delta_n(\mathbf{k})|\mathbf{d}_n(\mathbf{k})|}\frac{[\mathbf{d}_{nR}^{\mathcal{T}}(\mathbf{k}) \cdot \mathbf{d}_n(\mathbf{k})]^2}{|\mathbf{d}_n(\mathbf{k})|^2}$$

$$- a_{n2}(\mathbf{k})\frac{|\mathbf{d}_n(\mathbf{k})|}{\Delta_n(\mathbf{k})}\frac{[\mathbf{d}_{nR}^{\mathcal{N}}(\mathbf{k}) \cdot \mathbf{d}_n(\mathbf{k})]^2}{|\mathbf{d}_n(\mathbf{k})|^2}$$

$$+ 2\frac{a_{n2}(\mathbf{k})\Delta_n(\mathbf{k}) + a_{n1}(\mathbf{k})|\mathbf{d}_n(\mathbf{k})|}{\Delta_n(\mathbf{k})}\frac{h_{n0}(\mathbf{k})}{\sqrt{h_{n0}^2(\mathbf{k}) + \Delta_n^2(\mathbf{k})}}\mathbf{d}_{nR}^{\mathcal{T}}(\mathbf{k}) \cdot \frac{\mathbf{d}_{nR}^{\mathcal{N}}(\mathbf{k}) \times \mathbf{d}_n(\mathbf{k})}{|\mathbf{d}_n(\mathbf{k})|}$$

$$\left. + 4b_{n1}(\mathbf{k})|\Delta_{nR}^{\mathcal{T}}(\mathbf{k})|^2 + 4b_{n1}(\mathbf{k})\frac{[\mathbf{d}_{nR}^{\mathcal{T}}(\mathbf{k}) \cdot \mathbf{d}_n(\mathbf{k})]^2}{|\mathbf{d}_n(\mathbf{k})|^2} + 8b_{n2}(\mathbf{k})\Delta_{nR}^{\mathcal{T}}(\mathbf{k})\frac{\mathbf{d}_{nR}^{\mathcal{T}}(\mathbf{k}) \cdot \mathbf{d}_n(\mathbf{k})}{|\mathbf{d}_n(\mathbf{k})|} \right\},$$

$$\tag{115}$$

where $F_{0,n}$ does not depend on the intrinsic order parameters, $a_{n1}(\mathbf{k}) = a_{n+}(\mathbf{k}) + a_{n-}(\mathbf{k})$, $a_{n2}(\mathbf{k}) = a_{n+}(\mathbf{k}) - a_{n-}(\mathbf{k})$, $b_{n1}(\mathbf{k}) = b_{n+}(\mathbf{k})\left[\Delta_n(\mathbf{k}) + |\mathbf{d}_n(\mathbf{k})|\right]^2 + b_{n-}(\mathbf{k})\left[\Delta_n(\mathbf{k}) - |\mathbf{d}_n(\mathbf{k})|\right]^2$, $b_{n2}(\mathbf{k}) = b_{n+}(\mathbf{k})\left[\Delta_n(\mathbf{k}) + |\mathbf{d}_n(\mathbf{k})|\right]^2 - b_{n-}(\mathbf{k})\left[\Delta_n(\mathbf{k}) - |\mathbf{d}_n(\mathbf{k})|\right]^2$ and

$$a_{n\pm}(\mathbf{k}) = \frac{\tanh\left[\beta|E_{n\pm}^0(\mathbf{k})|/2\right]}{4|E_{n\pm}^0(\mathbf{k})|},$$

$$b_{n\pm}(\mathbf{k}) = \frac{\beta|E_{n\pm}^0(\mathbf{k})| - 2\tanh\left[\beta|E_{n\pm}^0(\mathbf{k})|/2\right] - \beta|E_{n\pm}^0(\mathbf{k})|\tanh^2\left[\beta|E_{n\pm}^0(\mathbf{k})|/2\right]}{32|E_{n\pm}^0(\mathbf{k})|^3}. \tag{116}$$

We can express the intrinsic singlet [triplet] order parameter with the help of basis functions for irreducible representations $\Delta_m^{(n)}(\mathbf{k})\left[\mathbf{d}_m^{(n)}(\mathbf{k})\right]$ ($m = 1, 2, 3...$) as

$$\Delta_{nR}^{\mathcal{T}}(\mathbf{k}) = \sum_m \psi_{nm}^{\mathcal{T}}\Delta_m^{(n)}(\mathbf{k}), \quad \Delta_{nR}^{\mathcal{N}}(\mathbf{k}) = \sum_m \psi_{nm}^{\mathcal{N}}\Delta_m^{(n)}(\mathbf{k}),$$

$$\mathbf{d}_{nR}^{\mathcal{T}}(\mathbf{k}) = \sum_m \eta_{nm}^{\mathcal{T}}\mathbf{d}_m^{(n)}(\mathbf{k}), \quad \mathbf{d}_{nR}^{\mathcal{N}}(\mathbf{k}) = \sum_m \eta_{nm}^{\mathcal{N}}\mathbf{d}_m^{(n)}(\mathbf{k}), \tag{117}$$

where $\psi_{nm}^{\mathcal{T}}, \psi_{nm}^{\mathcal{N}}, \eta_{nm}^{\mathcal{T}}, \eta_{nm}^{\mathcal{N}} \in \mathbb{R}$. The basis functions $\Delta_m^{(n)}(\mathbf{k})$, $\mathbf{d}_m^{(n)}(\mathbf{k})$ can be obtained by projecting the most general singlet and triplet order parameters into the irreducible representa-

tions of the symmetry group $G$ of the model. They can be written as

$$\Delta_1^{(n)}(\mathbf{k}) = a_{11}^{(n)}(\cos k_x - \cos k_y) + a_{12}^{(n)}[\cos(2k_x) - \cos(2k_y)] + a_{13}^{(n)}[\cos(2k_x)\cos k_y - \cos(2k_y)\cos k_x] + ..,$$
$$\Delta_2^{(n)}(\mathbf{k}) = a_{21}^{(n)} + ..,$$
$$\Delta_3^{(n)}(\mathbf{k}) = a_{31}^{(n)} \sin k_x \sin k_y + ..,$$
$$\Delta_4^{(n)}(\mathbf{k}) = a_{41}^{(n)} \sin k_x \sin k_y (\cos k_x - \cos k_y) + ...,$$
$$\mathbf{d}_1^{(n)}(\mathbf{k}) = b_{11}^{(n)}(\mathbf{e}_x \sin k_y + \mathbf{e}_y \sin k_x) + b_{12}^{(n)}(\cos k_x - \cos k_y)(\mathbf{e}_x \sin k_y - \mathbf{e}_y \sin k_x) + ...,$$
$$\mathbf{d}_2^{(n)}(\mathbf{k}) = b_{21}^{(n)}(\mathbf{e}_x \sin k_y - \mathbf{e}_y \sin k_x) + ...,$$
$$\mathbf{d}_3^{(n)}(\mathbf{k}) = b_{31}^{(n)}(\mathbf{e}_x \sin k_x - \mathbf{e}_y \sin k_y) + ...,$$
$$\mathbf{d}_4^{(n)}(\mathbf{k}) = b_{41}^{(n)}(\mathbf{e}_x \sin k_x + \mathbf{e}_y \sin k_y) + ...,$$
$$\mathbf{d}_{5A}^{(n)}(\mathbf{k}) = b_{5A1}^{(n)}\mathbf{e}_z \sin k_x + ...,$$
$$\mathbf{d}_{5B}^{(n)}(\mathbf{k}) = b_{5B1}^{(n)}\mathbf{e}_z \sin k_y + ... . \tag{118}$$

Each of these basis functions contains in general an infinite number of terms and transforms in a specific way under the symmetry transformations $g\mathbf{k}$ ($g \in G$). The coefficients $a_{mk}^{(n)} \in \mathbb{R}$ and $b_{mk}^{(n)} \in \mathbb{R}$ are in principle variational parameters (for each band $n$ independently of the others), and they should be chosen so that the free energy is minimized. Therefore, we have included a superscript $(n)$ in all basis functions indicating that these coefficients can be different in each band $n = \alpha, \beta, \gamma$. As explained below the exact values of these coefficients are not important for our qualitative results, and therefore in the end we will select the relevant coefficients phenomenologically for each band so that the overlap between intrinsic and induced order parameters is maximized.

This way we get (notice that $n = \alpha, \beta, \gamma$ denotes the band index and $m = 1, 2, 3...$ describes the basis functions)

$$F_{\mathrm{qp},n} \approx F_{0,n} - \sum_m [r_m^{ns} \psi_{nm}^{\mathscr{T}} + r_m^{nt} \eta_{nm}^{\mathscr{T}}]$$
$$- \sum_{m,m'} \left[ f_{m,m'}^{ns\mathscr{T}} \psi_{nm}^{\mathscr{T}} \psi_{nm'}^{\mathscr{T}} + f_{m,m'}^{ns\mathscr{N}} \psi_{nm}^{\mathscr{N}} \psi_{nm'}^{\mathscr{N}} + f_{m,m'}^{nt\mathscr{T}} \eta_{nm}^{\mathscr{T}} \eta_{nm'}^{\mathscr{T}} + f_{m,m'}^{nt\mathscr{N}} \eta_{nm}^{\mathscr{N}} \eta_{nm'}^{\mathscr{N}} \right]$$
$$- \sum_{m,m'} \left[ \kappa_{m,m'}^{n\mathscr{T}} \psi_{nm}^{\mathscr{T}} \eta_{nm'}^{\mathscr{T}} + \kappa_{m,m'}^{n\mathscr{N}} \psi_{nm}^{\mathscr{N}} \eta_{nm'}^{\mathscr{N}} \right] - \sum_{m,m'} \kappa_{m,m'}^{n\mathscr{T}\mathscr{N}} \eta_{nm}^{\mathscr{T}} \eta_{nm'}^{\mathscr{N}}, \tag{119}$$

where

$$r_m^{ns} = 2 \int_{FS} dk \big[ a_{n1}(\mathbf{k}) \Delta_n(\mathbf{k}) + a_{n2}(\mathbf{k}) |\mathbf{d}_n(\mathbf{k})| \big] \Delta_m^{(n)}(\mathbf{k}),$$

$$r_m^{nt} = 2 \int_{FS} dk \big[ a_{n1}(\mathbf{k}) |\mathbf{d}_n(\mathbf{k})| + a_{n2}(\mathbf{k}) \Delta_n(\mathbf{k}) \big] \frac{\mathbf{d}_m^{(n)}(\mathbf{k}) \cdot \mathbf{d}_n(\mathbf{k})}{|\mathbf{d}_n(\mathbf{k})|},$$

$$f_{m,m'}^{ns\mathcal{T}} = \int_{FS} dk \big[ a_{n1}(\mathbf{k}) + 4 b_{n1}(\mathbf{k}) \big] \Delta_m^{(n)}(\mathbf{k}) \Delta_{m'}^{(n)}(\mathbf{k}),$$

$$f_{m,m'}^{ns\mathcal{N}} = \int_{FS} dk \, a_{n1}(\mathbf{k}) \Delta_m^{(n)}(\mathbf{k}) \Delta_{m'}^{(n)}(\mathbf{k}),$$

$$f_{m,m'}^{nt\mathcal{T}} = \int_{FS} dk \left\{ \left[ a_{n1}(\mathbf{k}) + a_{n2}(\mathbf{k}) \frac{\Delta_n^2(\mathbf{k}) + |\mathbf{h}_n(\mathbf{k})|^2}{h_{n0}(\mathbf{k}) \frac{\mathbf{h}_n(\mathbf{k}) \cdot \mathbf{d}_n(\mathbf{k})}{|\mathbf{d}_n(\mathbf{k})|} + \Delta_n(\mathbf{k}) |\mathbf{d}_n(\mathbf{k})|} \right] \mathbf{d}_m^{(n)}(\mathbf{k}) \cdot \mathbf{d}_{m'}^{(n)}(\mathbf{k}) \right.$$

$$\left. + \left[ 4 b_{n1}(\mathbf{k}) - a_{n2}(\mathbf{k}) \frac{\Delta_n^2(\mathbf{k}) + |\mathbf{h}_n(\mathbf{k})|^2}{h_{n0}(\mathbf{k}) \frac{\mathbf{h}_n(\mathbf{k}) \cdot \mathbf{d}_n(\mathbf{k})}{|\mathbf{d}_n(\mathbf{k})|} + \Delta_n(\mathbf{k}) |\mathbf{d}_n(\mathbf{k})|} \right] \frac{[\mathbf{d}_m^{(n)}(\mathbf{k}) \cdot \mathbf{d}_n(\mathbf{k})][\mathbf{d}_{m'}^{(n)}(\mathbf{k}) \cdot \mathbf{d}_n(\mathbf{k})]}{|\mathbf{d}_n(\mathbf{k})|^2} \right\},$$

$$f_{m,m'}^{nt\mathcal{N}} = \int_{FS} dk \left\{ \left[ a_{n1}(\mathbf{k}) + a_{n2}(\mathbf{k}) \frac{|\mathbf{d}_n(\mathbf{k})|}{\Delta_n(\mathbf{k})} \right] \mathbf{d}_m^{(n)}(\mathbf{k}) \cdot \mathbf{d}_{m'}^{(n)}(\mathbf{k}) \right.$$

$$\left. - a_{n2}(\mathbf{k}) \frac{|\mathbf{d}_n(\mathbf{k})|}{\Delta_n(\mathbf{k})} \frac{[\mathbf{d}_m^{(n)}(\mathbf{k}) \cdot \mathbf{d}_n(\mathbf{k})][\mathbf{d}_{m'}^{(n)}(\mathbf{k}) \cdot \mathbf{d}_n(\mathbf{k})]}{|\mathbf{d}_n(\mathbf{k})|^2} \right\},$$

$$\kappa_{m,m'}^{n\mathcal{T}} = 2 \int_{FS} dk \big[ a_{n2}(\mathbf{k}) + 4 b_{n2}(\mathbf{k}) \big] \Delta_m^{(n)}(\mathbf{k}) \frac{\mathbf{d}_{m'}^{(n)}(\mathbf{k}) \cdot \mathbf{d}_n(\mathbf{k})}{|\mathbf{d}_n(\mathbf{k})|},$$

$$\kappa_{m,m'}^{n\mathcal{N}} = 2 \int_{FS} dk \, a_{n2}(\mathbf{k}) \Delta_m^{(n)}(\mathbf{k}) \frac{\mathbf{d}_{m'}^{(n)}(\mathbf{k}) \cdot \mathbf{d}_n(\mathbf{k})}{|\mathbf{d}_n(\mathbf{k})|},$$

$$\kappa_{m,m'}^{n\mathcal{T}\mathcal{N}} = 2 \int_{FS} dk \, \frac{a_{n2}(\mathbf{k}) \Delta_n(\mathbf{k}) + a_{n1}(\mathbf{k}) |\mathbf{d}_n(\mathbf{k})|}{\Delta_n(\mathbf{k})} \frac{h_{n0}(\mathbf{k})}{\sqrt{h_{n0}^2(\mathbf{k}) + \Delta_n^2(\mathbf{k})}} \mathbf{d}_m^{(n)}(\mathbf{k}) \cdot \frac{\mathbf{d}_{m'}^{(n)}(\mathbf{k}) \times \mathbf{d}_n(\mathbf{k})}{|\mathbf{d}_n(\mathbf{k})|}.$$

$$\tag{120}$$

By utilizing the symmetries (89)-(94), we obtain $h_{nz}(\mathbf{k}) = d_{nz}(\mathbf{k}) = 0$,

$$\Delta_n(\pm k_x, k_y) = \Delta_n(k_x, k_y), \ \Delta_n(k_x, \pm k_y) = \Delta_n(k_x, k_y), \ \Delta_n(k_y, k_x) = -\Delta_n(k_x, k_y),$$
$$d_{nx}(\pm k_x, k_y) = d_{nx}(k_x, k_y), \ d_{nx}(k_x, \pm k_y) = \pm d_{nx}(k_x, k_y), \ d_{nx}(k_y, k_x) = d_{ny}(k_x, k_y),$$
$$d_{ny}(\pm k_x, k_y) = \pm d_{ny}(k_x, k_y), \ d_{ny}(k_x, \pm k_y) = d_{ny}(k_x, k_y), \ d_{ny}(k_y, k_x) = d_{nx}(k_x, k_y),$$
$$h_{n0}(\pm k_x, k_y) = h_{n0}(k_x, k_y), \ h_{n0}(k_x, \pm k_y) = h_{n0}(k_x, k_y), \ h_{n0}(k_y, k_x) = h_{n0}(k_x, k_y),$$
$$h_{nx}(\pm k_x, k_y) = h_{nx}(k_x, k_y), \ h_{nx}(k_x, \pm k_y) = \pm h_{nx}(k_x, k_y), \ h_{nx}(k_y, k_x) = -h_{ny}(k_x, k_y),$$
$$h_{ny}(\pm k_x, k_y) = \pm h_{ny}(k_x, k_y), \ h_{ny}(k_x, \pm k_y) = h_{ny}(k_x, k_y), \ h_{ny}(k_y, k_x) = -h_{nx}(k_x, k_y).$$

$$\tag{121}$$

The basis functions in Eqs. (118) have been chosen in such a way that they all behave differently in these transformations. Moreover, we have ordered them so that $\Delta_1(\mathbf{k})$ and $\mathbf{d}_1(\mathbf{k})$ behave similarly as $\Delta_n(\mathbf{k})$ and $\mathbf{d}_n(\mathbf{k})$, respectively. Therefore, we can straightforwardly show that

$$r_m^{ns} = \delta_{m,1} r_1^{ns}, \ r_m^{nt} = \delta_{m,1} r_1^{nt}. \tag{122}$$

By utilizing the transformations (121) and the corresponding transformations for the basis

functions, we obtain

$$f_{m,m'}^{ns\mathcal{T}} = \delta_{m,m'} f_{m,m}^{ns\mathcal{T}}, \; f_{m,m'}^{ns\mathcal{N}} = \delta_{m,m'} f_{m,m}^{ns\mathcal{N}}, \; f_{m,m'}^{nt\mathcal{T}} = \delta_{m,m'} f_{m,m}^{nt\mathcal{T}}, \; f_{m,m'}^{nt\mathcal{N}} = \delta_{m,m'} f_{m,m}^{nt\mathcal{N}},$$
$$\kappa_{m,m'}^{n\mathcal{T}} = \delta_{m,m'} \kappa_{m,m}^{n\mathcal{T}}, \; \kappa_{m,m'}^{n\mathcal{N}} = \delta_{m,m'} \kappa_{m,m}^{n\mathcal{N}}, \; \kappa_{m,m'}^{n\mathcal{T}\mathcal{N}} = 0. \tag{123}$$

The equation $\kappa_{m,m'}^{n\mathcal{T}\mathcal{N}} = 0$ also directly follows from the transformation of the integrand $\mathbf{k} \rightarrow -\mathbf{k}$ in Eq. (120). We have also fixed the relative orderings of the singlet and triplet basis functions in a specific way in order to make sure that $\kappa_{m,m'}^{n\mathcal{T}}$ and $\kappa_{m,m'}^{n\mathcal{N}}$ are non-zero only if $m = m'$. Since $\kappa_{m,m'}^{n\mathcal{T}}$ and $\kappa_{m,m'}^{n\mathcal{N}}$ are only nonvanishing for $m = m'$, we will in the following simplify notation by renaming $\kappa_{m,m'}^{n\mathcal{T}}$ and $\kappa_{m,m'}^{n\mathcal{N}}$ to $\kappa_m^{n\mathcal{T}}$ and $\kappa_m^{n\mathcal{N}}$, respectively.

Additionally the free energy contains also the term arising from the decoupling of the interaction term

$$F_{I,n} = \sum_m \left[ \frac{1}{g_{nms}} (\psi_{nm}^{\mathcal{T}} \psi_{nm}^{\mathcal{T}} + \psi_{nm}^{\mathcal{N}} \psi_{nm}^{\mathcal{N}}) + \frac{1}{g_{nmt}} (\eta_{nm}^{\mathcal{T}} \eta_{nm}^{\mathcal{T}} + \eta_{nm}^{\mathcal{N}} \eta_{nm}^{\mathcal{N}}) \right], \tag{124}$$

where $g_{nms}$ and $g_{nmt}$ describe the strengths of the effective attractive interactions in the singlet and triplet channels, respectively. Their calculation would require the specification of the full microscopic interaction model. However, the exact values of $g_{nms}$ and $g_{nmt}$ are not important in the following.

This way we obtain

$$F_n \approx F_{0,n} + \sum_m \left[ F_{n,m}^{\mathcal{T}} + F_{n,m}^{\mathcal{N}} \right],$$
$$F_{n,m}^{\mathcal{T}} = -\delta_{m,1} \left[ r_1^{ns} \psi_{n1}^{\mathcal{T}} + r_1^{nt} \eta_{n1}^{\mathcal{T}} \right] + \left( \psi_{nm}^{\mathcal{T}}, \eta_{nm}^{\mathcal{T}} \right) \bar{M}_{n,m}^{\mathcal{T}} \left( \psi_{nm}^{\mathcal{T}}, \eta_{nm}^{\mathcal{T}} \right)^T,$$
$$F_{n,m}^{\mathcal{N}} = \left( \psi_{nm}^{\mathcal{N}}, \eta_{nm}^{\mathcal{N}} \right) \bar{M}_{n,m}^{\mathcal{N}} \left( \psi_{nm}^{\mathcal{N}}, \eta_{nm}^{\mathcal{N}} \right)^T, \tag{125}$$

where the stability matrices $\bar{M}_{n,m}$ are

$$\bar{M}_{n,m}^{\mathcal{T}} = \begin{pmatrix} g_{nms}^{-1} - f_{m,m}^{ns\mathcal{T}} & -\kappa_m^{n\mathcal{T}}/2 \\ -\kappa_m^{n\mathcal{T}}/2 & g_{nmt}^{-1} - f_{m,m}^{nt\mathcal{T}} \end{pmatrix}, \; \bar{M}_{n,m}^{\mathcal{N}} = \begin{pmatrix} g_{nms}^{-1} - f_{m,m}^{ns\mathcal{N}} & -\kappa_m^{n\mathcal{N}}/2 \\ -\kappa_m^{n\mathcal{N}}/2 & g_{nmt}^{-1} - f_{m,m}^{nt\mathcal{N}} \end{pmatrix}. \tag{126}$$

Since the intrinsic order parameters are obtained by minimizing the free energy, it is clear from this expression that each pair of order parameters $\left( \psi_{nm}^{\mathcal{T}}, \eta_{nm}^{\mathcal{T}} \right)$ and $\left( \psi_{nm}^{\mathcal{N}}, \eta_{nm}^{\mathcal{N}} \right)$, respectively, can be solved independently of each other. Furthermore, we can diagonalize the stability matrices by introducing a change of variables

$$\psi_{nm}^{\mathcal{T}(\mathcal{N})} = \chi_{nm1}^{\mathcal{T}(\mathcal{N})} \sin \theta_{nm}^{\mathcal{T}(\mathcal{N})} - \chi_{nm2}^{\mathcal{T}(\mathcal{N})} \cos \theta_{nm}^{\mathcal{T}(\mathcal{N})},$$
$$\eta_{nm}^{\mathcal{T}(\mathcal{N})} = \chi_{nm1}^{\mathcal{T}(\mathcal{N})} \cos \theta_{nm}^{\mathcal{T}(\mathcal{N})} + \chi_{nm2}^{\mathcal{T}(\mathcal{N})} \sin \theta_{nm}^{\mathcal{T}(\mathcal{N})},$$
$$\sin \theta_{nm}^{\mathcal{T}(\mathcal{N})} = \frac{1}{\sqrt{2}} \sqrt{1 + \frac{g_{nmt}^{-1} - f_{m,m}^{nt\mathcal{T}(\mathcal{N})} - g_{nms}^{-1} + f_{m,m}^{ns\mathcal{T}(\mathcal{N})}}{\sqrt{(g_{nmt}^{-1} - f_{m,m}^{nt\mathcal{T}(\mathcal{N})} - g_{nms}^{-1} + f_{m,m}^{ns\mathcal{T}(\mathcal{N})})^2 + (\kappa_m^{n\mathcal{T}(\mathcal{N})})^2}}},$$
$$\cos \theta_{nm}^{\mathcal{T}(\mathcal{N})} = \frac{1}{\sqrt{2}} \frac{\kappa_m^{n\mathcal{T}(\mathcal{N})}}{|\kappa_m^{n\mathcal{T}(\mathcal{N})}|} \sqrt{1 - \frac{g_{nmt}^{-1} - f_{m,m}^{nt\mathcal{T}(\mathcal{N})} - g_{nms}^{-1} + f_{m,m}^{ns\mathcal{T}(\mathcal{N})}}{\sqrt{(g_{nmt}^{-1} - f_{m,m}^{nt\mathcal{T}(\mathcal{N})} - g_{nms}^{-1} + f_{m,m}^{ns\mathcal{T}(\mathcal{N})})^2 + (\kappa_m^{n\mathcal{T}(\mathcal{N})})^2}}}. \tag{127}$$

For these new variables $\chi_{nm1(2)}^{\mathcal{T}(\mathcal{N})}$, which are the eigenmodes of the linearized gap equations and describe specific linear combinations of singlet and triplet order parameters, the free energy becomes

$$F_{n,m}^{\mathcal{T}} = -\delta_{m,1} \left[ r_{11}^{n\chi} \chi_{n11}^{\mathcal{T}} + r_{12}^{n\chi} \chi_{n12}^{\mathcal{T}} \right] + \mathcal{M}_{nm1}^{\mathcal{T}} |\chi_{nm1}^{\mathcal{T}}|^2 + \mathcal{M}_{nm2}^{\mathcal{T}} |\chi_{nm2}^{\mathcal{T}}|^2,$$
$$F_{n,m}^{\mathcal{N}} = \mathcal{M}_{nm1}^{\mathcal{N}} |\chi_{nm1}^{\mathcal{N}}|^2 + \mathcal{M}_{nm2}^{\mathcal{N}} |\chi_{nm2}^{\mathcal{N}}|^2, \tag{128}$$

where the Josephson couplings $r_{11}^{n\chi}$ and $r_{12}^{n\chi}$ for the eigenmodes $\chi_{n11}^{\mathcal{T}}$ and $\chi_{n12}^{\mathcal{T}}$, respectively, are

$$r_{11}^{n\chi} = r_1^{ns} \sin\theta_{n1}^{\mathcal{T}} + r_1^{nt}\cos\theta_{n1}^{\mathcal{T}}, \ r_{12}^{n\chi} = -r_1^{ns}\cos\theta_{n1}^{\mathcal{T}} + r_1^{nt}\sin\theta_{n1}^{\mathcal{T}} \tag{129}$$

and

$$\mathcal{M}_{nm1}^{\mathcal{T}(\mathcal{N})} = \frac{g_{nms}^{-1} - f_{m,m}^{ns\mathcal{T}(\mathcal{N})} + g_{nmt}^{-1} - f_{m,m}^{nt\mathcal{T}(\mathcal{N})}}{2}$$
$$- \frac{\sqrt{\left(g_{nmt}^{-1} - f_{m,m}^{nt\mathcal{T}(\mathcal{N})} - g_{nms}^{-1} + f_{m,m}^{ns\mathcal{T}(\mathcal{N})}\right)^2 + \left(\kappa_m^{n\mathcal{T}(\mathcal{N})}\right)^2}}{2},$$

$$\mathcal{M}_{nm2}^{\mathcal{T}(\mathcal{N})} = \frac{g_{nms}^{-1} - f_{m,m}^{ns\mathcal{T}(\mathcal{N})} + g_{nmt}^{-1} - f_{m,m}^{nt\mathcal{T}(\mathcal{N})}}{2}$$
$$+ \frac{\sqrt{\left(g_{nmt}^{-1} - f_{m,m}^{nt\mathcal{T}(\mathcal{N})} - g_{nms}^{-1} + f_{m,m}^{ns\mathcal{T}(\mathcal{N})}\right)^2 + \left(\kappa_m^{n\mathcal{T}(\mathcal{N})}\right)^2}}{2}. \tag{130}$$

The instability for the different eigenmodes (corresponding to the appearance of superconducting order parameters) appears when $\mathcal{M}_{nm1(2)}^{\mathcal{T}(\mathcal{N})}$ becomes negative. In order to simplify the theory, we assume that the temperature $T$ is above all the critical temperatures for superconducting instability in $Sr_2RuO_4$. This means that $\chi_{nm1(2)}^{\mathcal{T}(\mathcal{N})} = 0$ for $m \geq 2$, $\chi_{n11}^{\mathcal{N}} = 0$ and $\chi_{n12}^{\mathcal{N}} = 0$ i.e.

$$\psi_{nm}^{\mathcal{T}(\mathcal{N})} = 0 \text{ for } m \geq 2, \ \eta_{nm}^{\mathcal{T}(\mathcal{N})} = 0 \text{ for } m \geq 2, \psi_{n1}^{\mathcal{N}} = 0, \ \eta_{n1}^{\mathcal{N}} = 0, \tag{131}$$

and thus we only need to solve $\chi_{n11}^{\mathcal{T}}$ and $\chi_{n12}^{\mathcal{T}}$ determining $\psi_{n1}^{\mathcal{T}}$ and $\eta_{n1}^{\mathcal{T}}$ for each band $n = \alpha, \beta, \gamma$. By minimizing the free energy, we obtain

$$\chi_{n11}^{\mathcal{T}} = \frac{r_{11}^{n\chi}}{2\mathcal{M}_{n11}^{\mathcal{T}}}, \ \chi_{n12}^{\mathcal{T}} = \frac{r_{12}^{n\chi}}{2\mathcal{M}_{n12}^{\mathcal{T}}}. \tag{132}$$

Thus

$$\psi_{n1}^{\mathcal{T}} = \chi_{n11}^{\mathcal{T}}\sin\theta_{n1}^{\mathcal{T}} - \chi_{n12}^{\mathcal{T}}\cos\theta_{n1}^{\mathcal{T}} = \frac{\kappa_1^{n\mathcal{T}}r_1^{nt} + 2\left(g_{n1t}^{-1} - f_{1,1}^{nt\mathcal{T}}\right)r_1^{ns}}{4\left(g_{n1s}^{-1} - f_{1,1}^{ns\mathcal{T}}\right)\left(g_{n1t}^{-1} - f_{1,1}^{nt\mathcal{T}}\right) - |\kappa_1^{n\mathcal{T}}|^2},$$

$$\eta_{n1}^{\mathcal{T}} = \chi_{n11}^{\mathcal{T}}\cos\theta_{n1}^{\mathcal{T}} + \chi_{n12}^{\mathcal{T}}\sin\theta_{n1}^{\mathcal{T}} = \frac{\kappa_1^{n\mathcal{T}}r_1^{ns} + 2\left(g_{n1s}^{-1} - f_{1,1}^{ns\mathcal{T}}\right)r_1^{nt}}{4\left(g_{n1s}^{-1} - f_{1,1}^{ns\mathcal{T}}\right)\left(g_{n1t}^{-1} - f_{1,1}^{nt\mathcal{T}}\right) - |\kappa_1^{n\mathcal{T}}|^2}. \tag{133}$$

The effective interactions $g_{n1s}$ and $g_{n1t}$ are related to the native critical temperatures for singlet $T_{cs}$ and triplet $T_{ct}$ superconductivity (observed in the absence of induced superconductivity) in $Sr_2RuO_4$, respectively. In the framework of our approximations (where all the coefficients are computed as if the contribution would only come from the Fermi surface) we can express these relations as

$$g_{n1s}^{-1} = \frac{1}{4k_B T_{cs}}\int_{FS} dk \ \Delta_1^{(n)}(\mathbf{k})^2,$$

$$g_{n1t}^{-1} = \frac{1}{4k_B T_{ct}}\int_{FS} dk \ |\mathbf{d}_1^{(n)}(\mathbf{k})|^2. \tag{134}$$

By denoting

$$m^{ns} = g_{n1s}^{-1} - f_{1,1}^{ns\mathcal{T}}, \ \ m^{nt} = g_{n1t}^{-1} - f_{1,1}^{nt\mathcal{T}}, \tag{135}$$

we arrive to the expressions used in the main text.

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
