# Peer review of "Robust semi-Dirac points and unconventional topological phase transitions in doped superconducting Sr2IrO4 tunnel coupled to t2g electron systems"

_SciPost Physics, doi:SciPost Phys. 3, 041 (2017)_

## Round 2 · Referee Report · Anonymous (Referee 1) · 2017-10-2

Strengths

1- Careful and extensive analysis of the topic 2- The models include enough microscopic details that they are actually relevant for experiments

Weaknesses

1- Too many different models and Hamiltonians, it's hard to keep track

Report

The authors provide an in-depth analysis of the physics of a doped Sr2IrO4 layer on top of a t2g electron gas, like Sr2RuO4. They find multiple interesting phases, including semi-Dirac points. The phase diagram becomes even richer when the t2g electron gas becomes itself superconducting.

The paper is well written and provides a lot of useful information for theorists and experimentalists studying this kind of heterostructure. I therefore recommend publication.

Requested changes

1- It might help to have a Table listing all the different models (bilayer or trilayer, simplified or microscopically accurate,...) used in the paper and the conclusion reached for each of them. The paper is quite long and contains a lot of information, so it might be worth trying to summarize the main points even more.

2- The Fermi surfaces of bulk Sr2RuO4 actually have some non-trivial dependence on the out-of-plane momentum k_z if one looks at their spin-orbital content [arXiv:1303.5444]. It might therefore be an oversimplification to take the kz=0 Fermi surface to model a few layer system. Furthermore, the only monolayers of Sr2RuO4 that were made [ 1605.05669] showed some reconstruction and therefore have Fermi surfaces that are different from the bulk ones. In particular, the gamma pocket seemed to have gone through a Lifshitz transition and to be on the other side of the van Hove singularity. It might be good to check the results for a tight-binding model specifically fitted to the few-layer physics and not the bulk one.

---

## Round 3 · Author Response

Dear Editor,

we would like to thank you for communicating us the referee report. We thank the referee for her/his positive report and careful reading of the manuscript.

The referee has requested two changes on the manuscript.

  1. Referee: It might help to have a Table listing all the different models (bilayer or trilayer, simplified or microscopically accurate,...) used in the paper and the conclusion reached for each of them. The paper is quite long and contains a lot of information, so it might be worth trying to summarize the main points even more.

Response: We agree with the referee, and we have included a Table listing the different models and the conclusions reached for each of them in the last paragraph of the introduction.

  1. Referee: The Fermi surfaces of bulk Sr2RuO4 actually have some non-trivial dependence on the out-of-plane momentum k_z if one looks at their spin-orbital content [arXiv:1303.5444]. It might therefore be an oversimplification to take the kz=0 Fermi surface to model a few layer system. Furthermore, the only monolayers of Sr2RuO4 that were made [ 1605.05669] showed some reconstruction and therefore have Fermi surfaces that are different from the bulk ones. In particular, the gamma pocket seemed to have gone through a Lifshitz transition and to be on the other side of the van Hove singularity. It might be good to check the results for a tight-binding model specifically fitted to the few-layer physics and not the bulk one.

Response: The use of the bulk tight-binding parameters for Sr2RuO4 is a simplification. As mentioned by the referee the Fermi surfaces have a weak dependence on the out-of-plane momentum and in thin layers the gamma band can be closer to a Lifshitz transition than in the bulk system. Moreover, when heterostructures are created the band structure parameters will again be modified because of the interface reconstructions (as already pointed out in the manuscript). Predicting the correct band structure parameters for the heterostructures requires density functional theory calculations and even then there are large uncertainties in these predictions. The important point in our paper is however that the band structure parameters are not important for our qualitative conclusions. Similar kind of transitions will exist in the phase diagram as long as the symmetries are present and the tunneling matrices are topologically nontrivial. The effect of the band structure parameters is only a renormalization of the critical points of phase transitions in the phase diagrams shown in the paper. We also point out that based on the reference mentioned by the referee the gamma band can be on either side of the Lifshitz transition depending on whether the t2g material is chosen to be Sr2RuO4 or Ba2RuO4 and the choice of the substrate material. Therefore, our current choice of using the bulk Sr2RuO4 tight binding parameters is well justified. Moreover, the possibility of changing the tight binding parameters for example by strain engineering opens new interesting possibilities because it may allow to drive the system through topological phase transitions in a controlled way. We have discussed this in the revised version of the manuscript on p11.

We have addressed all the comments of the referee and we believe that the paper is now suitable for publication in SciPost.

Yours sincerely, Mats Horsdal and Timo Hyart

---

## Round 3 · List of Changes

- Added a short summary of the models studied in the paper at the end of Section I

- Added Table I

- Added discussion about bulk vs single-layer Sr2RuO4 on p11.

-Added references [18], [86] and [87].

- Added sentence below eq. (52) in supplementary material:
Since $\kappa^{n\cal{T}}_{m,m'}$ and $\kappa^{n\cal{N}}_{m,m'}$ are only nonvanishing for $m=m'$, we will in the following simplify notation by renaming $\kappa^{n\cal{T}}_{m,m'}$ and $\kappa^{n\cal{N}}_{m,m'}$ to $\kappa^{n\cal{T}}_{m}$ and $\kappa^{n\cal{N}}_{m}$, respectively.

- Changed $\kappa^{n\cal{N}}_{m,m'}$ to $\kappa^{n\cal{N}}_{m}$ in (55) in the Supplementary material.

---

## Editorial Decision

published